# LENGTH GENERALIZATION WITH LOG-DEPTH RECURRENT UNITS

## ABSTRACT

Length generalization remains a persistent challenge for neural networks: recurrent models tend to suffer from positional biases, while Transformers are constrained by fixed computational depth. Regular languages provide a frequently used testbed for evaluating length generalization, as any sequence can be exactly verified to determine its label. We propose the Log-Depth Recurrent Unit (LDRU), which composes token embeddings through a learned pairwise operator inspired by monoid composition, yielding uniform logarithmic depth across tokens. On 21 regular tasks, consisting of standard benchmarks and new prefix languages, the LDRU achieves 100% out-of-distribution accuracy on 18 tasks and at least 96% on the remaining 3, consistently outperforming recurrent and attention-based models. These results establish the LDRU as an effective architecture for length generalization on regular languages and a promising direction for compositional sequence modeling.

## 1 INTRODUCTION

Generalization to out-of-distribution (OOD) sequence lengths remains a central open challenge in neural modeling. As defining OOD sequences is difficult in natural language contexts, regular languages have become a popular testbed for studying this problem. Regular languages are simple yet diverse, connect directly to automata theory, and allow exact verification of correctness at any sequence length. Evaluating architectures on these tasks enables precise measurement of whether a model has learned a task's underlying rules (Strobl et al., 2024). Prior work (Delétang et al., 2023; Chi et al., 2023; Butoi et al., 2025) shows that networks with recurrence generalize more reliably to regular languages than Transformers (Vaswani et al., 2017), which operate at a fixed depth regardless of sequence length. However, recurrent neural networks (RNNs; Elman, 1990) still suffer from positional bias: early tokens must traverse longer computational paths than later ones, leading to long-range memory problems (Bengio et al., 1994). It is therefore desirable to design an architecture that demonstrates reliable generalization to regular tasks as a first step toward architectures that reliably generalize to more complex language tasks.

To address this gap, we propose the *Log-Depth Recurrent Unit* (LDRU), an architecture that composes token embeddings through a learned pairwise operator applied in a balanced reduction tree. This design avoids the positional bias of RNNs by giving every token the same computational depth, while still allowing information to propagate across the entire sequence. We hypothesize that when trained on regular tasks, the LDRU will generalize as it is designed to induce a monoid representing the underlying task, ideally enabling the same expressiveness as RNNs, with improved parallelizability across the sequence and the ability to handle long-range memory problems more effectively.

RNNs are often associated with deterministic finite automata (DFAs) because of their state-based inductive bias (Weiss et al., 2018; Merrill, 2019; Michalenko et al., 2019). The LDRU, in contrast, draws on another algebraic structure with equivalent power over regular languages: the *monoid* (Sakarovitch, 2009). In this setting, a monoid consists of equivalence classes capturing the sequences that induce the same behavior (state-mapping) within its associated DFA. These classes are composed with an associative operator, enabling the use of the *reduction algorithm* (Ladner & Fischer, 1980; Blelloch, 1990) for efficient sequence processing. The LDRU is designed to learn such an operator, which we hypothesize enables generalization through composition to regular tasks. We discuss the connection to automata theory more technically in Appendix A. The reduction algo-

rithm takes an associative binary operator $\odot$ to reduce a sequence of elements $e_1, \ldots, e_n$ into a single result $e_1 \odot \cdots \odot e_n$. This result can be computed efficiently using a balanced binary tree (Blelloch, 1990) (see Fig. 2). Processing sequences with the reduction directly motivates the LDRU's design: we parameterize a binary operator $\odot_\theta$ that learns to approximate a reduction operator $\odot$.

We evaluate the LDRU's ability to generalize to regular language transduction tasks (regular tasks for conciseness). To specifically test long-range dependency handling, we propose prefix languages, regular tasks that depend only on a fixed-length prefix of a sequence to determine its output symbol. We compare the LDRU to established architectures: the RNN, the LSTM (Hochreiter & Schmidhuber, 1997), the Transformer, and RegularGPT (Chi et al., 2023), a recurrent Transformer with adaptive depth that approximates a reduction-like operation with attention masks.

The LDRU consistently outperforms all baselines, achieving 100% generalization on 18 out of 21 tasks when trained on sequences up to length 40, and near-perfect on the remaining 3. Our experiments show that the LDRU's generalization improves when trained on longer sequences; monoid analysis reveals that the LDRU's generalization failures arise from insufficient coverage of monoid data rather than an architectural limitation.

Although our experiments focus on regular languages, this is a deliberate choice: they offer a rigorous testbed where generalization can be measured exactly. Demonstrating systematic length generalization here establishes a clear proof of concept for the LDRU's inductive bias. We view these results as a foundation for extending reduction-based architectures to broader compositional domains like algorithmic reasoning, where formal verification may be infeasible.

This paper makes the following contributions:

- We propose the *Log-Depth Recurrent Unit* (LDRU), a neural architecture with logarithmic computational depth with an associative inductive bias to approximate monoid operators.

- We introduce *prefix languages*, a new class of regular tasks designed to test long-range dependency handling.

- We demonstrate the LDRU's *superior length generalization* across 21 regular tasks, including established benchmarks and our prefix languages. We further evaluate the LDRU on ListOps (Nangia & Bowman, 2018), a hierarchical task requiring compositional generalization, and 8 natural language tasks: classification tasks from GLUE (Wang et al., 2019), AG's News, and DBPedia (Zhang et al., 2015), where the LDRU shows competitive performance compared to Transformer baselines.

- We examine training data requirements for length generalization, relating generalization failures to the lack of equivalence class compositions in short sequences, providing insights that may inform future applications beyond regular tasks.

The rest of this paper is organized as follows. Section 2 introduces the prefix languages. Section 3 details the method. Section 4 explains our experimental set-up, and we provide the results in Section 5. Related work is discussed in Section 7. We discuss the implications of our findings in Section 6. Finally, we conclude in Section 8.

## 2 PREFIX LANGUAGES

We introduce prefix languages, denoted $P_{p,q}$, regular tasks for testing long-range dependency handling. The first $p$ symbols determine the machine's final state, requiring a model to retain memory of its current state while processing the remainder of the sequence. We also parameterize $q$, the number of symbols in the alphabet. See Fig. 1 for a graphical representation of $P_{2,2}$. We give a mathematical definition of prefix languages in Appendix B.

## 3 METHOD

The main component of the LDRU is a parameterized binary operator $\odot_\theta$. The LDRU processes sequences by repeatedly applying $\odot_\theta$ to adjacent pairs of embeddings, inducing a binary tree structure (see Fig. 2 for an illustration of the process on a length 8 sequence). For a sequence of length $n$,

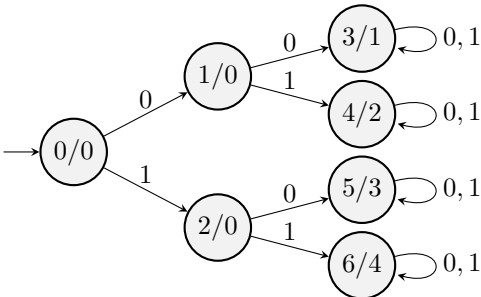

Figure 1: The prefix language $P_{2,2}$. Nodes are states, and edges are transitions. Labels within nodes $X/Y$ indicate the state $X$ and output symbol $Y$ if a sequence's run ends in $X$.

processing requires $\lceil \log_2 n \rceil$ steps. Each step consists of applying the operator on all pairs, followed by a residual feedforward network, then layer normalization. The weights of these components are also shared across steps. We pad odd-length sequences with a zero embedding, which acts as a neutral placeholder.

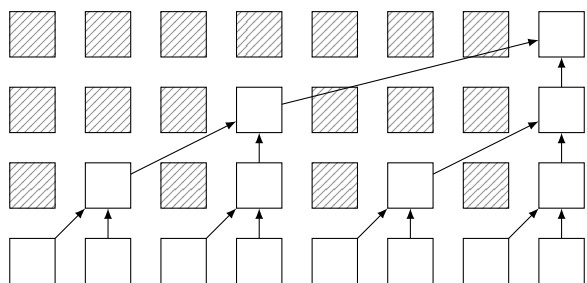

Figure 2: LDRU processing of a length 8 sequence. Each step of $\odot_\theta$ (Eqs. 1–3) halves the number of embeddings, and all steps form a binary tree reduction. After $\log_2(8) = 3$ steps, the final embedding represents the sequence, demonstrating the $O(\log n)$ depth of the LDRU.

The core of $\odot_\theta$ is an element-wise sum (an associative operation) with learned gating weights to control information flow. We produce the gating weights using a multi-layer perceptron (MLP). The operator's gating design draws from the success of gating information flow in recurrent architectures (Hochreiter & Schmidhuber, 1997; Chung et al., 2014; Jozefowicz et al., 2015). We define the operator $\odot_\theta : \mathbb{R}^d \times \mathbb{R}^d \to \mathbb{R}^d$ on inputs $\mathbf{h}_i, \mathbf{h}_j \in \mathbb{R}^d$ as follows:

$$[\mathbf{g}_i; \mathbf{g}_j] = \text{MLP}([\mathbf{h}_i; \mathbf{h}_j]) \text{ where } \mathbf{g}_i, \mathbf{g}_j \in \mathbb{R}^d \tag{1}$$

$$\mathbf{f}_i = \mathbf{V}_i(\mathbf{g}_i \circ \mathbf{h}_i) + \mathbf{b}_i, \mathbf{f}_j = \mathbf{V}_j(\mathbf{g}_j \circ \mathbf{h}_j) + \mathbf{b}_j \tag{2}$$

$$\odot_\theta(\mathbf{h}_i, \mathbf{h}_j) = \mathbf{W}_{\text{out}}(\mathbf{f}_i + \mathbf{f}_j) + \mathbf{b}_{\text{out}} \in \mathbb{R}^d, \tag{3}$$

where $[\cdot; \cdot]$ denotes concatenation, $\circ$ is element-wise multiplication, $\mathbf{V}_i, \mathbf{V}_j, \mathbf{W}_{\text{out}} \in \mathbb{R}^{d \times d}$ are learned projection matrices, and $\mathbf{b}_i, \mathbf{b}_j, \mathbf{b}_{\text{out}} \in \mathbb{R}^d$ are learned bias vectors. The MLP has three layers with ReLU activation and expansion factors $(1, 2, 1)$. $\mathbf{V}_i$, $\mathbf{V}_j$, and $\mathbf{W}_{out}$ are all initialized with the identity, and the corresponding biases are initialized to $\mathbf{0}$. The inherently associative element-wise sum combining token embeddings and the combination of these initializations biases $\odot_\theta$ toward approximately associative behavior.

We briefly describe how a sequence is processed by the whole model. The sequence is first embedded using a learned embedding layer, followed by layer normalization and a residual feedforward network. We apply dropout after each $\odot_\theta$ step, which encourages more robust generalization (see Appendix E.1). The embedded sequence is then processed by $\odot_\theta$, as detailed above. The output, $\mathbf{h}_n$, is passed to a linear classifier to produce class logits. Complete architectural diagrams and initialization schemes are provided in Appendix C.

## 3.1 COMPLEXITY ANALYSIS

The LDRU achieves $O(nd^2)$ work complexity with $O(\log n)$ computational depth (referred to as depth), combining RNN-like linear scaling (of sequence length) with logarithmic-time paralleliza-tion. This contrasts with RNNs' $O(n)$ sequential depth and Transformers' $O(n^2d + nd^2)$ quadratic sequence length complexity. Further discussion on complexity is provided in Appendix D, which also includes an empirical runtime analysis whose scaling reflects the $\lceil \log_2(n) \rceil$ step structure of the LDRU.

Table 1: Computational complexity comparison. The LDRU achieves $\log n$ depth and avoids quadratic length scaling.

| Architecture | Work Complexity | Depth |
|---|:---:|:---:|
| RNN | $O(nd^2)$ | $O(n)$ |
| Transformer | $O(n^2d + nd^2)$ | $O(1)$ |
| **LDRU (Ours)** | $\mathbf{O(nd^2)}$ | $\mathbf{O(\log n)}$ |

We present the practical trade-offs between the LDRU, the RNN, and the Transformer in Fig. 3. We profile the forward and backward passes of each architecture on sequences of varying lengths with a batch size of 32. While the LDRU costs more FLOPs and reaches higher peak memory usage than the RNN, it's increased parallelism on the GPU results in higher throughput. The Transformer is significantly slower and costs more FLOPs due to its quadratic length complexity.

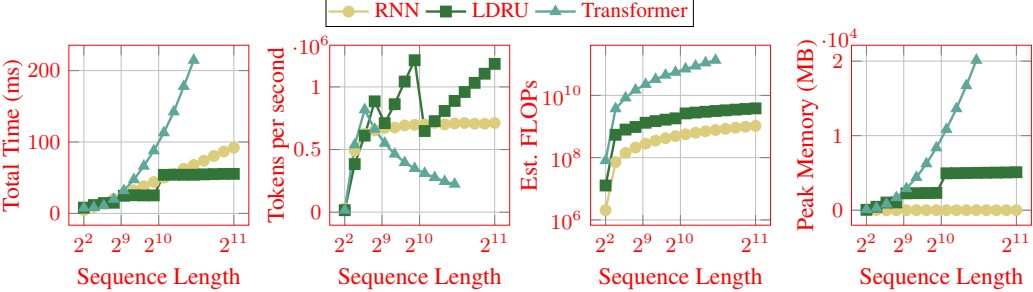

Figure 3: Empirical runtime analysis of the LDRU, RNN, and Transformer. We note that the LDRU's throughput reaches maxima at powers of 2 as these are the optimal lengths for reduction, this is followed by a drop in throughput once it exceeds the power of 2.

## 4 EXPERIMENTS

We evaluate the LDRU using sequence classification tasks. In the regular task setting, this corre-sponds to predicting the single output symbol produced by the task's corresponding Moore machine. We illustrate the evaluated machines in Appendix E. The regular tasks we examine are a combina-tion of those previously studied by Delétang et al. (2023) and Bhattamishra et al. (2020), and 6 parameterizations of prefix languages. We also study ListOps (Nangia & Bowman, 2018) to assess the LDRU's ability to hierarchical tasks beyond regular languages. We aim to answer the following research questions:

**RQ1** Does the LDRU's generalization ability outperform established state-of-the-art architectures?

**RQ2** Does increasing the maximum training sequence length improve OOD performance?

**RQ3** Does the choice of $\odot_\theta$ impact OOD performance?

All models were trained under identical conditions (same training lengths, optimizer, regularization, and number of steps), with full hyperparameters detailed in Appendix E. We train each model by minimizing the cross-entropy loss between the class logits and the ground-truth labels. We use the AMSGrad (Reddi et al., 2018) optimizer for regular task experiments, as we empirically observed

improved generalization (see Appendix E.1 for details). We use the Adam (Kingma & Ba, 2015) optimizer for the non-regular tasks. We use 3 seeds per experiment to evaluate performance robustness to weight initialization (unless otherwise stated). We now describe the experiments designed to answer each research question.

**RQ1** We train the LDRU on 21 regular tasks on sequences of variable length, from 1 to 40. We evaluate generalization using OOD accuracy, i.e., the mean accuracy over 512 sampled sequences for each length from 41 to 500. This setup is standard in length generalization studies on regular tasks (Delétang et al., 2023; Liu et al., 2023; Ruoss et al., 2023; Chi et al., 2023). We compare the LDRU to the classic RNN, the LSTM, the Transformer, and RegularGPT (Chi et al., 2023) (a recurrent Transformer designed for generalization to regular tasks) architectures. We train for 100k or 1M steps, depending on the task. We use 3 positional encodings (PEs) for the Transformer: NoPE (Kazemnejad et al., 2023), ALiBi (Press et al., 2022), and randomized RoPE (Ruoss et al., 2023), denoted as $\sim$RoPE. We assess the statistical significance of performance differences between the LDRU and baseline architectures across tasks using Student's t-tests.

We additionally train the LDRU on ListOps sequences of length 5 to 40 with a maximum nesting of 3 and at most 9 arguments per list. We train models using 3 dataset sizes: 100k, 500k, and 1M sequences. We evaluate on a range of sequences to study length, depth and argument generalization. For evaluation, we consider length buckets: $[5, 20]$, $[21, 40]$, $[41, 60]$, $[61, 80]$, $[81, 100]$, $[101, 200]$, max depths: 3, 5, max arguments: 9, 14. We were able to generate 10k samples per combination of these variables except for lengths $[81, 100]$, $[101, 200]$ for max depth 3, max arguments 9. We use 5 seeds per experiment. We compare the LDRU to the LSTM, Transformer with NoPE, ALiBi, and standard Sinusoidal PE (Vaswani et al., 2017), and BBT-GRC (Shi et al., 2018), a balanced binary tree recursive neural network equipped with a Gated Recursive Cell (Shen et al., 2019). Further detail can be found in Appendix H.1.

**RQ2** We train LDRUs and RNNs on the $D_4$, $D_6$, $D_8$, and $D_{12}$ tasks (where $D_n$ is the recognition of the Dyck-1 language up to $n$ stack depth), using maximum lengths of the training sequences: 60, 100, and 150. We use these tasks because neither the RNN nor the LDRU generalizes with 100% accuracy to these tasks when trained with a maximum training length of 40 (except for the LDRU on $D_4$). We report the OOD accuracy, e.g., for a maximum training length of 60, the evaluation is on sequences of length 61 to 500.

**RQ3** We compare four architectural choices of the pairwise operator $\odot_\theta$ on the regular tasks from Delétang et al. (2023). All other settings are the same as for **RQ1**. We test: (1) our MLP implementation (Eqs. 1–3), (2) element-wise sum, (3) a linear projection of the concatenation of the two embeddings without activation, and (4) a simple gated combination of the two embeddings. We report mean OOD accuracy for each operator-task combination.

## 5 RESULTS

We present the results of the **RQ1** regular experiments in Table 2. The LDRU demonstrates superior length generalization to all baselines, achieving 100% OOD accuracy on 18 tasks and near-perfect performance on the remaining 3 tasks. The high performance of the LDRU outperforms all 3 Transformer (TF) PE schemes except $\sim$RoPE on $D_8$ and $D_{12}$. $\sim$RoPE has the best performance on these tasks, indicating that while it is not as consistent as the LDRU, it is more powerful than NoPE and ALiBi on $D_n$. However, this PE does not provide consistent gains as its performance on Modular Arithmetic is considerably lower (31.4%) compared to the other PEs (76.0%, 54.4% for NoPE and ALiBi, respectively). The LDRU outperforms RegularGPT and matches or exceeds the capabilities of RNNs and LSTMs (except the RNN on $D_8$). We also evaluated two state space models (SSMs), S5 (Smith et al., 2023) and SD-SSM (Terzić et al., 2025), and found that they both performed worse than the LDRU on all tasks. We compared the LDRU and RNN performance on $D_8$ and determined that the performance gap is not significant. Full results tables are given in Appendix F.

In Fig. 4, we show ListOps results on the 1M dataset (additional results provided in Appendix F). BBT-GRC achieves the highest accuracy across test sequences for all dataset sizes, followed by the LDRU, then Transformer (ALiBi) and LSTM, then the Transformer with NoPE and Sinusoidal PEs. All models show increased performance when trained on larger datasets, but the relative gap between

Table 2: OOD accuracy results across 21 regular tasks. All architectures are trained on sequences with lengths 1–40 and evaluated on lengths 41–500. Results show mean $\pm$ standard deviation across seeds. A $^\dagger$ indicates statistical significance of the LDRU's improvements over baselines.

| Task | RNN | LSTM | TF (NoPE) | TF (ALiBi) | TF ($\sim$RoPE) | RegularGPT | LDRU |
|---|---|---|---|---|---|---|---|
| Even Pairs | $77.4 \pm 12.2$ | $100.0 \pm 0.0$ | $50.3 \pm 0.0^\dagger$ | $61.1 \pm 5.1^\dagger$ | $91.5 \pm 7.9$ | $91.9 \pm 0.7^\dagger$ | $\mathbf{100.0 \pm 0.0}$ |
| Modular Arithmetic | $100.0 \pm 0.0$ | $100.0 \pm 0.0$ | $76.0 \pm 0.0^\dagger$ | $54.4 \pm 4.5^\dagger$ | $31.4 \pm 5.5^\dagger$ | $99.1 \pm 1.3$ | $\mathbf{100.0 \pm 0.0}$ |
| Parity Check | $100.0 \pm 0.0$ | $100.0 \pm 0.0$ | $50.1 \pm 0.0^\dagger$ | $49.9 \pm 0.0^\dagger$ | $49.9 \pm 0.0^\dagger$ | $99.8 \pm 0.7$ | $\mathbf{100.0 \pm 0.0}$ |
| Cycle Navigation | $100.0 \pm 0.0$ | $60.3 \pm 2.4^\dagger$ | $20.0 \pm 0.0^\dagger$ | $21.6 \pm 0.7^\dagger$ | $23.3 \pm 1.5^\dagger$ | $99.9 \pm 0.2$ | $\mathbf{100.0 \pm 0.0}$ |
| $D_2$ | $100.0 \pm 0.0$ | $100.0 \pm 0.0$ | $100.0 \pm 0.0^\dagger$ | $100.0 \pm 0.0$ | $98.2 \pm 1.7$ | $93.7 \pm 5.5$ | $\mathbf{100.0 \pm 0.0}$ |
| $D_3$ | $100.0 \pm 0.0$ | $97.7 \pm 4.0$ | $99.9 \pm 0.0^\dagger$ | $98.3 \pm 2.6$ | $92.5 \pm 10.2$ | $87.6 \pm 4.6^\dagger$ | $\mathbf{100.0 \pm 0.0}$ |
| $D_4$ | $95.0 \pm 8.7$ | $100.0 \pm 0.0$ | $99.5 \pm 0.0^\dagger$ | $96.0 \pm 4.7$ | $97.6 \pm 0.4^\dagger$ | $94.1 \pm 4.7$ | $\mathbf{100.0 \pm 0.0}$ |
| $D_6$ | $97.1 \pm 4.7$ | $98.4 \pm 1.6$ | $96.5 \pm 0.0$ | $92.3 \pm 6.8$ | $98.4 \pm 1.0$ | $87.9 \pm 3.7^\dagger$ | $\mathbf{98.5 \pm 2.4}$ |
| $D_8$ | $\mathbf{99.2 \pm 1.3}$ | $89.0 \pm 4.3$ | $89.0 \pm 0.2^\dagger$ | $89.7 \pm 7.7$ | $98.7 \pm 0.8$ | $91.0 \pm 2.8^\dagger$ | $98.1 \pm 1.6$ |
| $D_{12}$ | $92.7 \pm 9.5$ | $82.1 \pm 1.7^\dagger$ | $81.3 \pm 0.3^\dagger$ | $84.4 \pm 9.6$ | $\mathbf{98.7 \pm 1.0}$ | $89.8 \pm 4.1^\dagger$ | $96.0 \pm 2.1^\dagger$ |
| Tomita 3 | $100.0 \pm 0.0$ | $100.0 \pm 0.0$ | $98.7 \pm 0.0^\dagger$ | $100.0 \pm 0.0^\dagger$ | $70.7 \pm 8.9^\dagger$ | $93.7 \pm 3.3$ | $\mathbf{100.0 \pm 0.0}$ |
| Tomita 4 | $100.0 \pm 0.0$ | $100.0 \pm 0.0$ | $96.1 \pm 0.0^\dagger$ | $100.0 \pm 0.0$ | $85.5 \pm 2.1^\dagger$ | $98.4 \pm 1.5$ | $\mathbf{100.0 \pm 0.0}$ |
| Tomita 5 | $100.0 \pm 0.0$ | $100.0 \pm 0.0$ | $74.3 \pm 0.0^\dagger$ | $74.3 \pm 0.0^\dagger$ | $74.3 \pm 0.0^\dagger$ | $97.7 \pm 2.2$ | $\mathbf{100.0 \pm 0.0}$ |
| Tomita 6 | $100.0 \pm 0.0$ | $100.0 \pm 0.0$ | $50.2 \pm 0.0^\dagger$ | $50.0 \pm 0.0^\dagger$ | $50.0 \pm 0.0^\dagger$ | $92.2 \pm 4.7$ | $\mathbf{100.0 \pm 0.0}$ |
| Tomita 7 | $100.0 \pm 0.0$ | $100.0 \pm 0.0$ | $100.0 \pm 0.0$ | $100.0 \pm 0.0$ | $100.0 \pm 0.0$ | $100.0 \pm 0.0$ | $\mathbf{100.0 \pm 0.0}$ |
| $P_{1,2}$ | $82.2 \pm 3.3^\dagger$ | $100.0 \pm 0.0$ | $50.3 \pm 0.0^\dagger$ | $56.0 \pm 2.3^\dagger$ | $89.2 \pm 12.7$ | $100.0 \pm 0.0$ | $\mathbf{100.0 \pm 0.0}$ |
| $P_{2,2}$ | $65.4 \pm 24.0$ | $100.0 \pm 0.0$ | $25.2 \pm 0.0^\dagger$ | $93.0 \pm 5.6$ | $86.4 \pm 5.0^\dagger$ | $99.6 \pm 0.3$ | $\mathbf{100.0 \pm 0.0}$ |
| $P_{4,2}$ | $99.4 \pm 1.0$ | $100.0 \pm 0.0$ | $6.3 \pm 0.0^\dagger$ | $90.0 \pm 10.3$ | $58.5 \pm 11.1^\dagger$ | $91.5 \pm 2.1^\dagger$ | $\mathbf{100.0 \pm 0.0}$ |
| $P_{1,4}$ | $61.4 \pm 15.7$ | $100.0 \pm 0.0$ | $25.2 \pm 0.0^\dagger$ | $40.3 \pm 2.4^\dagger$ | $79.5 \pm 7.0^\dagger$ | $100.0 \pm 0.0$ | $\mathbf{100.0 \pm 0.0}$ |
| $P_{2,4}$ | $96.9 \pm 2.5$ | $100.0 \pm 0.0$ | $6.3 \pm 0.0^\dagger$ | $31.0 \pm 8.1^\dagger$ | $51.6 \pm 7.6^\dagger$ | $99.7 \pm 0.2$ | $\mathbf{100.0 \pm 0.0}$ |
| $P_{4,4}$ | $85.0 \pm 15.0$ | $97.3 \pm 2.2$ | $0.4 \pm 0.0^\dagger$ | $21.3 \pm 7.6^\dagger$ | $58.2 \pm 2.7^\dagger$ | $95.3 \pm 0.6^\dagger$ | $\mathbf{100.0 \pm 0.0}$ |

BBT-GRC and the LDRU increases from 100k to 500k. However, the LDRU remains competitive, slightly outperforming the Transformer (ALiBi) and LSTM, particularly in the longest sequences.

The hyperparameter sweep on the LDRU showed that its performance improves when associativity regularization is applied to $\odot_\theta$, minimizing the cosine distance between $(\mathbf{h}_a \odot_\theta \mathbf{h}_b) \odot_\theta \mathbf{h}_c$ and $\mathbf{h}_a \odot_\theta (\mathbf{h}_b \odot_\theta \mathbf{h}_c)$. This observation indicates that, despite not achieving the highest performance, the LDRU learns approximately associative composition, consistent with its intended inductive bias toward compositional generalization. We detail this additional loss term in Appendix G.

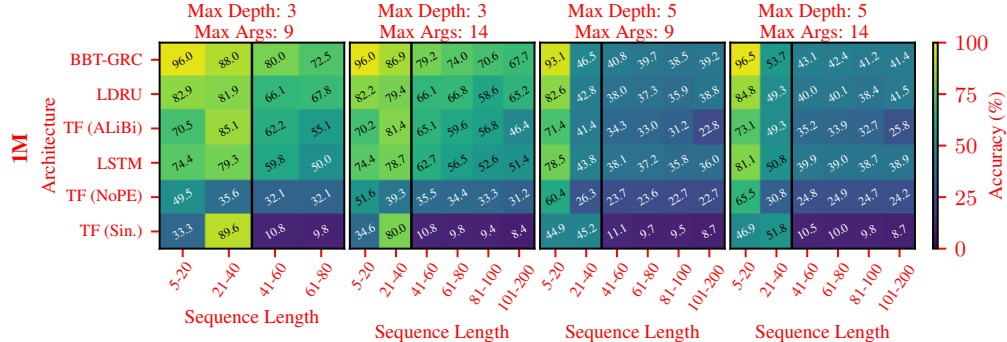

Figure 4: Heatmaps presenting the results of the ListOps experiments. Each cell presents the accuracy of 10k generated sequences for that particular combination of sequence length, maximum depth, and maximum number of arguments.

We present the results for **RQ2** in Fig. 5. The OOD accuracy of both the LDRU and RNN tends to improve as the maximum training length increases. The LDRU's performance on $D_{12}$ is consistent with its performance on the other $D_n$ tasks, whereas the RNN's performance on $D_{12}$ indicates that it is highly sensitive to weight initialization. This observation suggests that it may be easier to induce a monoid-like structure rather than a DFA-like structure for complex languages.

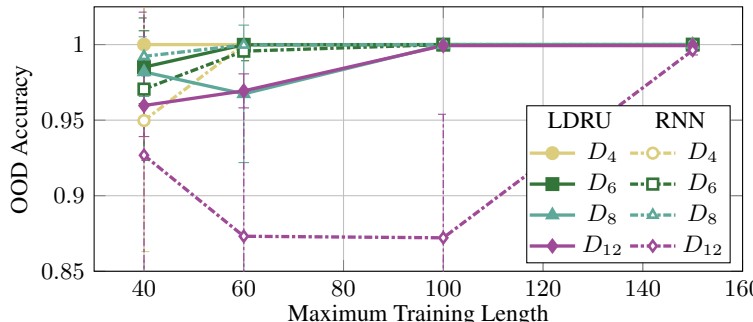

Figure 5: Trained LDRU and RNN architectures tend to achieve higher OOD accuracy on the $D_n$ tasks as maximum training length increases.

We present the results of **RQ3** in Table 3. Our MLP operator achieves 100% OOD accuracy on the four tasks. The other three choices fail on Modular Arithmetic, with element-wise sum also failing on Even Pairs. Performance varies on Modular Arithmetic, with the element-wise sum achieving 32.6%, the linear operator achieving 60.8%, and the gated sum achieving 67.1% OOD accuracy. In contrast, Parity Check and Cycle Navigation maintain 100.0% accuracy across all operator variants. Even Pairs appears to be a task with intermediate sensitivity, with element-wise sum dropping to 51.9% while the linear operator and gated sum maintain 100.0% accuracy. These results highlight that the LDRU's ability to generalize depends critically on the capacity of the operator $\odot_\theta$.

Table 3: OOD accuracies across different architectural choices of $\odot_\theta$ and regular tasks.

| Operator | Even Pairs | Modular Arithmetic | Parity Check | Cycle Navigation |
|---|---|---|---|---|
| MLP | $100.0 \pm 0.0$ | $100.0 \pm 0.0$ | $100.0 \pm 0.0$ | $100.0 \pm 0.0$ |
| Elem. Sum | $51.9 \pm 0.1$ | $32.6 \pm 0.3$ | $100.0 \pm 0.0$ | $100.0 \pm 0.0$ |
| Linear | $100.0 \pm 0.0$ | $60.8 \pm 1.7$ | $100.0 \pm 0.0$ | $100.0 \pm 0.0$ |
| Gated Sum | $100.0 \pm 0.0$ | $67.1 \pm 2.9$ | $100.0 \pm 0.0$ | $100.0 \pm 0.0$ |

## 6 DISCUSSION

Maximum training length is the primary driver of OOD performance in the regular task setting. As the maximum training length increases, the LDRU reaches 100% OOD accuracy on $D_6$ by length 60 and 99.9–100.0% on $D_8$ and $D_{12}$ by length 100 (Fig. 5). The RNN shows a similar trend but with higher variability on $D_{12}$ (standard deviations 8.2–9.5% up to length 100). These observations are consistent with the positional bias of recurrent networks, where earlier tokens must pass through longer computational paths. Learning $D_{12}$ requires the model to handle a greater number of possible state runs compared to the other $D_n$ tasks, which may increasingly challenge its capacity for long-term dependencies. We do not claim a definitive causal explanation here, but emphasize the LDRU's more stable behavior under increased training lengths.

Fig. 6 provides insight into why the LDRU does not reach 100.0% OOD accuracy on the $D_n$ languages when trained with insufficient sequence lengths. The heatmaps show the empirical probability of witnessing equivalence class (EC) compositions in the underlying monoid for even-length sequences from $D_6$. This task provides a useful compromise: it is among the harder cases where models do not immediately generalize, while remaining tractable to visualize. The leftmost heatmap reflects training sequences of length 10–40 and shows that many complex compositions are rarely observed, whereas the rightmost heatmap, taken from the longest OOD sequences we evaluate, displays a denser pattern. This sparsity gap is consistent with the empirical observations provided in Fig. 5, where $D_6$ achieves 98.5% OOD accuracy when trained on sequences up to length 40 but reaches 100% once trained on sequences of length 60 and above. These findings suggest that limited training lengths mean the model is not sufficiently exposed to rare EC compositions, leading to imperfect generalization.

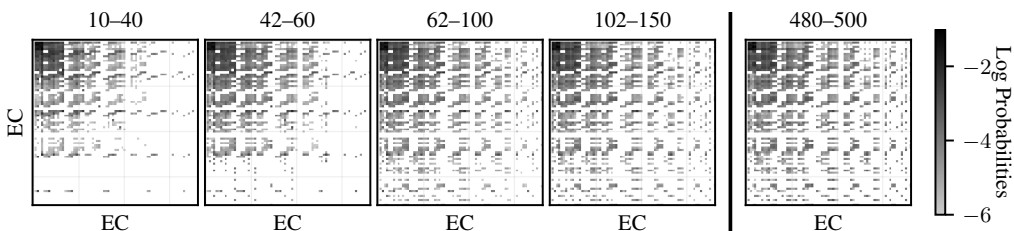

Figure 6: EC composition analysis for the $D_6$ syntactic monoid across different sequence length ranges. Each heatmap shows the log probability of compositions between ECs in the monoid. The sparse composition patterns in shorter sequences (lengths 10–40) explain why training with insufficient length leads to imperfect generalization, as it does not cover all compositions in the longer sequences (lengths 480–500).

To examine whether the LDRU's representations reflect monoid structure, we analyze the embedding space of $D_6$. We present the embeddings of all sequences up to length 12 using t-SNE (van der Maaten & Hinton, 2008) in Fig. 7a; the upper and lower plots show embeddings of the LDRU trained up to lengths 40 and 150, respectively. The embeddings display clustering into ECs, supporting our expectation that the LDRU's induced representation aligns with monoid structures. If the LDRU had learned to behave as a DFA, we would expect fewer clusters (8 states) compared to the syntactic monoid (141 classes).

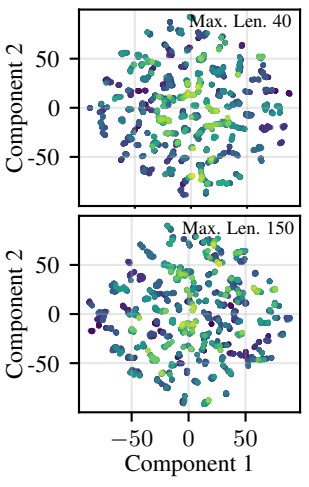

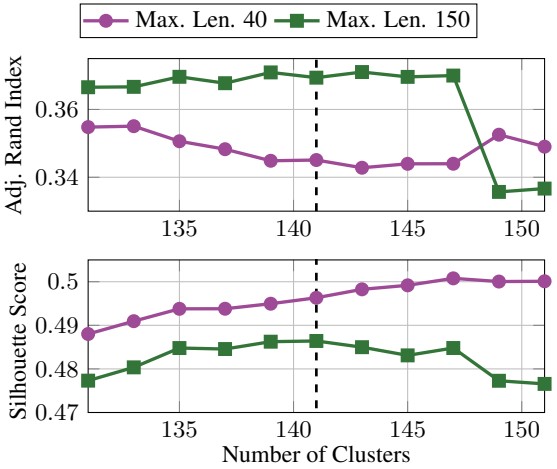

(a) t-SNE plot of $D_6$ embeddings colored by EC, showing distinct clusters learned by the LDRU.

(b) Clustering analysis of $D_6$ embeddings. The model with higher maximum training length displays improved alignment with the monoid structure, shown by higher ARI (top) and Silhouette Score peak (bottom).

We further analyze the embeddings using $k$-means clustering, varying $k$ around 141, the number of ECs in the $D_6$ syntactic monoid. We report the Adjusted Rand Index (ARI; Hubert & Arabie, 1985) and Silhouette Score (Rousseeuw, 1987) in Fig. 7b. The ARI measures the similarity between fitted clusters and the true ECs, while the Silhouette Score quantifies the separation between clusters. The length 150 model attains a higher ARI than the length 40 model, indicating improved alignment with ECs. The Silhouette Score peaks at $k = 141$ for the length 150 model, consistent with clustering at the expected granularity of the monoid. In contrast, the length 40 model has no clear peak, reflecting weaker alignment to the monoid.

The LDRU's design reflects two key principles for length generalization. First, the reduction's structure gives all tokens equal computational depth, mitigating depth-related positional bias. Second, in our experiments, only the MLP operator achieves 100% OOD accuracy on Modular Arithmetic, whereas simpler operators such as element-wise sum can still solve tasks like Parity Check and

Cycle Navigation. Residual connections and layer normalization after each $\odot_\theta$ step help stabilize optimization across logarithmic depth. Together, these elements balance structural constraints that encourage more general compositional learning, suggesting possible applicability to more complex cases. To substantiate this claim, we conduct preliminary experiments on natural language sequence classification tasks from GLUE (Wang et al., 2019), AG's News, and DBPedia (Zhang et al., 2015). Experimental details can be found in Appendix H.2. We present the results of these experiments in Table 4. We observe that the LDRU performs competitively with Transformers on these tasks when trained from scratch, achieving the best performance on QQP, MNLI (matched and mismatched), and DBPedia. These results indicate the LDRU's potential applicability beyond regular languages.

Table 4: Performance across GLUE benchmarks and two additional text classification datasets. Metrics follow GLUE conventions: CoLA (Matthew's Correlation Coefficient), SST-2 / MNLI / QNLI (%, accuracy), QQP / MRPC (%, F1/accuracy). Subscript M indicates matched and MM mismatched accuracy for MNLI. AG's News and DBPedia report classification accuracy on test data. Values are means across 5 seeds.

| Architecture | GLUE Benchmarks | | | | | | | Additional Tasks | |
| | CoLA | SST-2 | MRPC | QQP | $\text{MNLI}_\text{M}$ | $\text{MNLI}_\text{MM}$ | QNLI | AG's News | DBPedia |
|---|---|---|---|---|---|---|---|---|---|
| TF (ALiBi) | 0.096 | 79.3 | 56.2/62.6 | 74.0/78.3 | 53.9 | 52.8 | 58.1 | **89.8** | 98.2 |
| TF (Sin.) | **0.124** | **81.1** | **59.1/67.5** | 70.7/75.8 | 49.0 | 50.5 | 56.9 | 89.1 | 97.8 |
| TF (NoPE) | 0.097 | 79.1 | 59.0/64.0 | 72.8/77.8 | 50.5 | 51.1 | **58.8** | 89.6 | 97.8 |
| LDRU | 0.085 | 80.9 | 57.0/63.2 | **76.9/79.1** | **58.3** | **57.9** | 56.1 | 89.0 | **98.6** |

## 7 RELATED WORK

We position the LDRU within the context of related work.

**Tree-Based Recursive Neural Networks** Balanced binary tree recursive neural networks (Munkhdalai & Yu, 2017, BBT-RvNNs) form an essential precursor to the LDRU, but they differ significantly in their algorithmic constraints and computational objectives. Classic RvNNs (Socher et al., 2013; Tai et al., 2015; Yu & Liu, 2018; Shi et al., 2018; Shen et al., 2019) use a recursive cell to process sequences according to a binary tree that is either fixed, heuristically chosen, or induced from a linguistic prior. In contrast, the LDRU operator is explicitly designed to approximate a compositional algebra. This inductive bias is absent in standard recursive cells, which makes them less suitable for learning stable or repeatable composition rules.

More recent works, such as Recursion-in-Recursion (RIR) (Ray Chowdhury & Caragea, 2023), also leverage log-depth recursion to achieve strong length generalization on complex algorithmic tasks, including ListOps. RIR is a framework for trading off speed with RvNN expressivity, via a two-level recursion using an RvNN (inner recursion) within an $k$-ary tree structure (outer recursion). The LDRU is distinct from these methods because its operator is biased toward learning associative structures. Approximate associativity allows the reduction to behave like a recurrence over the input sequence. This distinction matters for generalization: RvNNs without an associativity bias may inherit fragile arbitrary bracketing, whereas the LDRU's operator must serve as a consistent partial evaluator for every subsequence. These algorithmic constraints align with our equivalence class analysis explaining why the LDRU did not generalize to $D_8$ and $D_{12}$.

**Length Generalization** Length generalization on formal languages is a key test of systematic reasoning in sequence models (Delétang et al., 2023; Butoi et al., 2025). Classic RNNs and LSTMs tend to show alignment with regular languages (Merrill et al., 2020), but using a state-based inductive bias limits efficient parallelization and introduces challenges with long-range dependencies (Bengio et al., 1993; 1994). Our evaluation of the prefix languages highlights the RNN's difficulty in modeling long-range dependencies. Transformers have well-documented length generalization problems (Anil et al., 2022; Liu et al., 2023; Hahn & Rofin, 2024; Zhou et al., 2024a;b; Huang et al., 2025b;a), and state space models (Gu et al., 2022) exhibit similar limitations (Fan et al., 2024; Sarrof et al., 2024; Terzić et al., 2025). Our empirical evaluation confirms these limitations, with both SSMs underperforming the LDRU on all 21 regular tasks. Theory has shown that Transformers can solve specific regular tasks like Parity Check (Chiang & Cholak, 2022), but they are generally not suited to the structure of finite-state automata (Hahn, 2020). Recurrent Transformers trade

parallelism for improved length generalization (Soulos et al., 2024; Fan et al., 2025). Of these architectures, RegularGPT (Chi et al., 2023) represents the closest approach to our work through adaptive weight sharing and sliding-window-dilated attention, which together implement the scan. However, layer norm parameters are not shared between adaptive layers, and empirically, RegularGPT's performance underperforms the LDRU.

**Architecture Modifications** One line of research toward length generalization explores augmentations to existing architectures for improving length generalization. Approaches for improving generalization in Transformers include scratchpad methods (Nye et al., 2022; Kazemnejad et al., 2023) that use intermediate reasoning steps, position coupling (Cho et al., 2024; McLeish et al., 2024; Cho et al., 2025) that assign structure to positional encodings, and modifying positional encodings (Press et al., 2022; Ruoss et al., 2023). In this work, our experiments focus on sequence classification or single-token prediction, where scratchpad methods and position coupling are not directly applicable, as they require autoregressive decoding. Prior evidence on randomized positional encodings shows that such modifications can yield task improvements, but they do not enable reliable generalization across regular tasks (Ruoss et al., 2023), consistent with our evaluation on ∼RoPE. Other work has considered additional loss terms to encourage length generalization (Butoi et al., 2025), observing that, again, its improvements are not robust across all tasks. This contrasts with the LDRU, which achieves reliable generalization across all 21 regular tasks we evaluate.

**Reduction Applications** The reduction (and its more general algorithm, the scan (Blelloch, 1990)) has been used to accelerate existing machine learning methods, including RNNs (Martin & Cundy, 2018) and backpropagation (Wang et al., 2020). More recently, it has been used to more efficiently compute linear state updates in SSMs like S4 (Gu et al., 2022), Mamba (Gu & Dao, 2023), and S5 (Smith et al., 2023). These approaches have enabled fast, state-of-the-art performance in reinforcement learning contexts (Lu et al., 2023). Parallel DeltaNet (Yang et al., 2024) likewise accelerates linear recurrent updates but stays within a linear state-space formulation, and log-linear attention (Guo et al., 2025) achieves logarithmic depth by hierarchically expanding a linear RNN's state. Prefix-Scannable Models (Yau et al., 2025) permit arbitrary (even non-associative) aggregation under a fixed Blelloch parenthesization, whereas LDRU aims at the opposite regime: it learns an operator that is stable across parenthesizations, enabling it to act as a true reduction rather than a tree-specific aggregator. Learnable monoids have also been proposed as aggregation functions over nodes in graphs (Ong & Veličković, 2022).

## 8 CONCLUSION

In this work, we have presented three principal contributions that advance the understanding of length generalization in neural networks. We proposed the *Log-Depth Recurrent Unit* (LDRU), a neural architecture that implements the reduction algorithm to process sequences with logarithmic depth, intended to induce compositional structure when trained to solve regular tasks. We introduced the *prefix languages*, a novel class of regular tasks specifically designed to test long-range dependency handling, which highlighted a weakness in the classic RNNs' ability to generalize. We conducted a comprehensive empirical evaluation demonstrating that the LDRU achieves superior length generalization compared to standard architectures, attaining 100% accuracy on 18 of 21 regular tasks and near-perfect performance on the remaining tasks: the remaining 3 reached an OOD accuracy of at least 99.9% when increasing the maximum training sequence length. Our claims are restricted to regular tasks, where the algebraic structure provides clear ground truth and rigorous evaluation, and we leave extensions to broader domains for future work.

Several research directions emerge from our findings. First, *establishing formal guarantees for enabling generalization* presents an immediate challenge. Our results on the $D_n$ tasks suggest that generalization with the LDRU requires exposure to all equivalence class compositions, but we lack formal bounds on the minimum training sequence lengths needed to achieve this coverage. Second, *benchmarking the LDRU's computational efficiency* requires systematic evaluation against established baselines on throughput and memory usage across varying sequence lengths. While our analysis demonstrates $O(\log n)$ depth in principle, it remains to be shown whether this advantage translates into practical efficiency gains on modern hardware. Finally, *extending to more complicated tasks* would test how the LDRU behaves in more difficult settings, where we hypothesize it can maintain length generalization advantages on tasks with compositional structure.

ETHICS STATEMENT

This work uses synthetic formal language benchmarks and does not involve any sensitive data. We do not foresee any immediate ethical concerns.

**Use of Large Language Models** We used large language models as assistive tools to help rewrite and improve the clarity and readability of the paper. They were not used for research ideation, experimental design, or data analysis. All scientific content, results, and conclusions are our own, and we take full responsibility for the final manuscript.

REPRODUCIBILITY STATEMENT

We provide full implementation details in Appendix E, including hyperparameters, extended methods for data generation, and complete results tables. An anonymous repository with the source code, data scripts, and step-by-step instructions for installation, toy experiments, and full reproduction of all results will be shared with reviewers during the review process. If the paper is accepted, we will de-anonymize and publicly release the repository.

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

# A  CONNECTION TO AUTOMATA THEORY

This section provides the necessary automata theory to discuss the *syntactic monoid*; Sakarovitch (2009) provides a thorough introduction to this area.

An alphabet $\Sigma$ is a finite set of symbols. A sequence is a finite concatenation of symbols from $\Sigma$, and $\Sigma^*$ denotes the set of all sequences. A language is a subset of $\Sigma^*$.

A *deterministic finite automaton* (DFA) is defined as $\mathcal{A} = (Q, \Sigma, \delta, q_I, F)$, where $Q$ is a finite set of states, $\delta$ is a transition function $\delta : Q \times \Sigma \to Q$, $q_I \in Q$ is an initial state, and $F \subseteq Q$ is a set of accepting states. $\delta(q, s) = r$ denotes a transition from $q$ to $r$ labeled with $s$. The $(Q, \Sigma, \delta)$ component of a DFA is a *semiautomaton*. A run is a sequence of states $q_0, q_1, \ldots, q_n$ where there is a transition between each pair of consecutive states. A sequence is accepted by a DFA if and only if it induces a run from $q_I$ to a state in $F$ using $\delta$; otherwise, the sequence is rejected. The language a DFA recognizes is the sequences it accepts, and DFAs are equivalent when they recognize the same language. A DFA is minimal if no equivalent DFA has fewer states. A language is regular when it can be recognized by a DFA (Sakarovitch, 2009).

Transduction tasks, like evaluating Modular Arithmetic expressions, require replacing $F$ in the DFA with an output alphabet $S$ and a function mapping states to output symbols $g : Q \to S$. This replacement transforms a DFA into a *Moore machine*, i.e., $(Q, \Sigma, S, \delta, q_I, g)$ (Sakarovitch, 2009). The final state of the run induced by processing a sequence is passed to $g$ to produce that sequence's output symbol. For consistency, we treat all tasks as transduction tasks by converting recognition tasks into Moore machines with $S = \{0, 1\}$ and $g(q) = 1$ if $q \in F$ and $g(q) = 0$ otherwise.

A *monoid* is $(M, \odot, \epsilon)$, where $M$ is a set, $\odot : M \times M \to M$ is an associative binary operator and $\epsilon$ is a neutral element for $\odot$ (Sakarovitch, 2009). Let $\mathcal{A}$ be a DFA that recognizes language $L$. Every sequence $w \in \Sigma^*$ induces a function $e : Q \to Q$ in $\mathcal{A}$. The *transition monoid* $(E, \odot, \epsilon)$ of $\mathcal{A}$ is the monoid of $e$ functions induced by $\Sigma^*$ in $\mathcal{A}$. The monoid operator works by composing two state mappings into a new state mapping. We refer to $e$ functions as *equivalence classes* (ECs)

as there exists a canonical morphism $\varphi : \Sigma^* \to E$ (Holzer & König, 2004). We can process a sequence $w \in \Sigma^*$ by computing $\varphi(w)(q_I) \in F$. Transition monoids are equivalent representations of semiautomata. The *syntactic monoid* is the transition monoid of the minimal DFA that recognizes $L$.

It is not practically feasible to process a sequence $w \in \Sigma^*$ with $\varphi(w)$ directly as its domain is infinite, but we can derive its EC compositionally. We compose ECs of subsections of the sequence using $\odot$. As $\odot$ is associative, any composition order of ECs will result in a correct evaluation of $\varphi(w)$.

For example, let $w = w_1 \ldots w_n$ be a sequence and assume our task is to decide if $w$ is accepted by the regular language $L$ with minimal DFA $(Q, \Sigma, \delta, q_I, F)$ and syntactic monoid $(E, \odot, \epsilon)$. Suppose instead of $\varphi$ we have $\varphi' : \Sigma \to E$. If we apply $\varphi'$ to each $w_i$, then we obtain a sequence of monoid elements $\varphi(w) = \varphi'(w_1) \odot \cdots \odot \varphi'(w_n) = e_1 \odot \cdots \odot e_n$ with $e_i \in E$. We can efficiently leverage the associativity to process regular languages using the reduction (Hillis & Steele, 1986).

The LDRU is designed to learn an approximation to the syntactic monoid of the target language. We parameterize the monoid operator $\odot_\theta$ using a neural network to approximate the true operator $\odot$. The LDRU's reduction procedure allows it to compose the learned ECs over input subsequences in a parallelizable manner, enabling efficient processing of long sequences. By training the LDRU on examples from the target language, it learns to approximate the syntactic monoid's structure and behavior, facilitating accurate sequence evaluation even for lengths beyond those seen during training.

# B  PREFIX LANGUAGES DEFINITION

The prefix language, $P_{p,q}$, with a prefix length of $p$ over $q$ symbols is a Moore machine defined as:

$$P_{p,q} = (Q = \left\{0, 1, 2, \ldots, \frac{q^{p+1} - 1}{q - 1} - 1\right\},$$
$$\Sigma = \{0, \ldots, q - 1\},$$
$$S = \{0, \ldots, q^p\},$$
$$\delta = \delta_{p,q},$$
$$I = 0,$$
$$g = g_{p,q}),$$

where $g_{p,q}(i) = 0$ when $i \in \left\{0, \ldots, \frac{q^p - 1}{q - 1} - 1\right\}$ and $i - (\frac{q^p - 1}{q - 1} - 1)$ otherwise. The transition function $\delta_{p,q}$ is $\delta(i, j) = iq + 1 + j$ when $i \in \left\{0, \ldots, \frac{q^p - 1}{q - 1} - 1\right\}$. Otherwise, $\delta(i, j) = i \, \forall j \in \Sigma$. The output function $g_{p,q}$ ensures that the first $\frac{q^p - 1}{q - 1} - 1$ states output 0, and the remaining states output their state number minus $\frac{q^p - 1}{q - 1} - 1$.

**Worked Example** To illustrate how to use this definition to construct a prefix language, we consider the case of $P_{4,2}$, i.e., the prefix language with a prefix length of 4 over an alphabet of size 2. For these values, we calculate the associated $p, q$ based terms for $P_{4,2}$: $q - 1 = 1$, $q^p = 16$, $\frac{q^{p+1} - 1}{q - 1} - 1 = 30$ and $\frac{q^p - 1}{q - 1} - 1 = 14$. To provide clarity on these terms:

1. the first is the maximum integer present in the alphabet, $\Sigma = \{0, 1\}$;

2. the second is the maximum integer present in the output alphabet, $S = \{0, \ldots, 16\}$ (we do not subtract 1 here because 0 is the invalid prefix symbol and the remainder are the possible different prefixes);

3. the third is the maximum state, $Q = \{0, 1, \ldots, 30\}$ (for a total of 31 states, $\sum_{i=0}^{4} q^i = 31$);

4. the fourth is the term controlling the output function, ensuring that the output is 0 for the first 15 states and the output is the state number minus 14 for the remaining states. It also controls the transition function, ensuring that the first 15 states can transition to the following states while reading the first $p$ symbols, and defines self-transitions when reading any further symbols.

These terms lead us to the Moore machine for $P_{4,2}$ as follows:

$$
\begin{aligned}
P_{4,2} = (&Q = \{0, \ldots, 14\}, \\
&\Sigma = \{0, 1\}, \\
&S = \{0, \ldots, 16\}, \\
&\delta = \delta_{4,2}, \\
&I = 0, \\
&g = g_{4,2}),
\end{aligned}
$$

where $g_{4,2}(i) = 0$ when $i \in \{0, \ldots, 14\}$ and $i - 14$ otherwise (i.e., when $i \in \{15, \ldots, 30\}$). The transition function $\delta_{4,2}$ is defined as $\delta(i, j) = 2i + 1 + j$ when $i \in \{0, \ldots, 14\}$ and $i$ otherwise. For example, $\delta_{4,2}(0, 0) = 1$ and $\delta_{4,2}(0, 1) = 2$. For state 1, $\delta_{4,2}(1, 0) = 3$ and $\delta_{4,2}(1, 1) = 4$.

## C  LDRU OPERATOR DETAILS

We detail the MLP parameterization of the binary operator $\odot_\theta$ as described in the main text. The operator is designed to compute a weighted combination of two input embeddings $\mathbf{h}_i, \mathbf{h}_j \in \mathbb{R}^d$ using a three-layer MLP with gating vectors. We provide reference diagrams in Fig. 8.

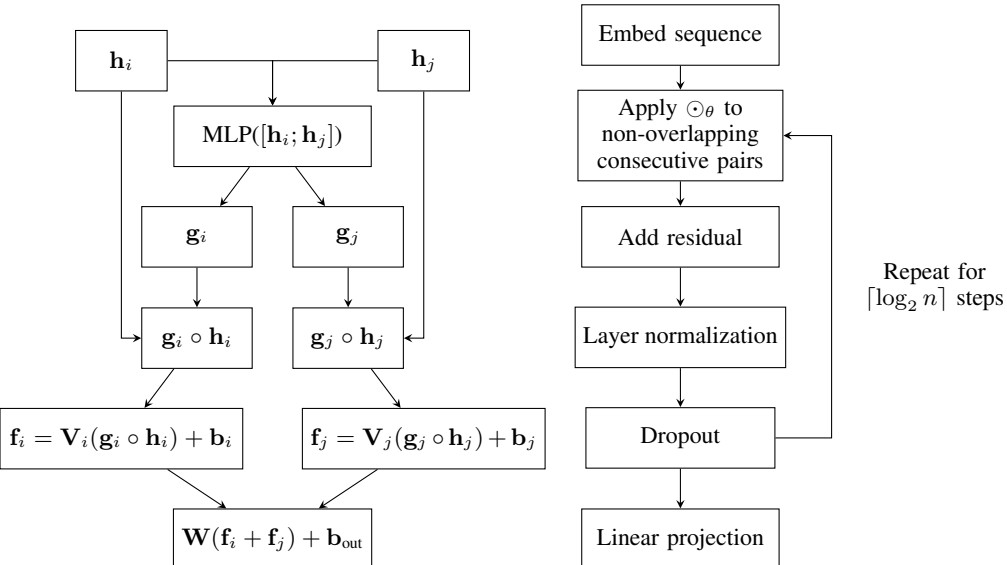

(a) Detailed implementation of the $\odot_\theta$ operator showing the MLP gating mechanism with element-wise multiplication and separate linear projections.

(b) Complete LDRU processing pipeline of an $n$-length sequence, showing the reduction implementation with residual connections and normalization.

Figure 8: LDRU architecture diagrams showing (a) the detailed $\odot_\theta$ operator implementation and (b) the complete processing pipeline.

The MLP used to produce the gating vectors from the concatenated embeddings $[\mathbf{h}_i; \mathbf{h}_j] \in \mathbb{R}^{2d}$ is structured as follows. The MLP consists of three linear layers with ReLU activations in between, and it outputs two gating vectors $\mathbf{g}_i, \mathbf{g}_j \in \mathbb{R}^d$, one for each input embedding. The hidden dimensions of the layers are $(2d, 4d, 2d)$. We expand the MLP function used in Eq. 1 for further clarity. The MLP

produces gating vectors given two input embeddings $\mathbf{h}_i, \mathbf{h}_j \in \mathbb{R}^d$ as follows:

$$\mathbf{x} = [\mathbf{h}_i; \mathbf{h}_j] \in \mathbb{R}^{2d} \tag{4}$$

$$\mathbf{z}_1 = \mathbf{W}_1 \mathbf{x} + \mathbf{b}_1 \in \mathbb{R}^{2d} \tag{5}$$

$$\mathbf{z}_2 = \mathrm{ReLU}(\mathbf{z}_1) \tag{6}$$

$$\mathbf{z}_3 = \mathbf{W}_2 \mathbf{z}_2 + \mathbf{b}_2 \in \mathbb{R}^{4d} \tag{7}$$

$$\mathbf{z}_4 = \mathrm{ReLU}(\mathbf{z}_3) \tag{8}$$

$$[\mathbf{g}_i; \mathbf{g}_j] = \mathbf{W}_3 \mathbf{z}_4 + \mathbf{b}_3 \in \mathbb{R}^{2d}. \tag{9}$$

We initialize the weights of the MLP and linear projections using standard techniques to ensure effective training. The MLP weights are initialized using Glorot normal initialization (Glorot & Bengio, 2010). The projections $\mathbf{V}_i, \mathbf{V}_j$ and the output projection $\mathbf{W}_{\text{out}}$ are initialized to the identity matrix. All biases are initialized to zero, enabling the model to behave as a gated element-wise sum initially.

## D  COMPUTATIONAL COMPLEXITY ANALYSIS

We provide a detailed computational complexity analysis of the LDRU compared to RNNs and Transformers. We analyze complexity in terms of:

**Work complexity**  Total number of operations required to process a sequence.

**Depth complexity**  Longest chain of sequential operations (determines parallelizability).

**Memory complexity**  Total memory required to store parameters and intermediate results.

We consider a sequence of length $n$ and embedding dimension $d$. The LDRU processes the sequence using $n-1$ $\odot_\theta$ operations, each requiring $O(d^2)$ work due to the MLP computation. The total work complexity is therefore $O(nd^2)$. The reduction processes sequences in exactly $\lceil \log_2 n \rceil$ steps, giving it depth $O(\log n)$ because $\lceil \log_2 n \rceil \leq \log_2(2n) = 1 + \log_2(n)$. The parameters of $\odot_\theta$ are $O(d^2)$, leading to a total memory complexity of $O(nd + d^2)$ when including the storing of intermediate embeddings. Note that complexities reflect standard implementations; attention variants beyond dot-product attention can improve Transformer scaling.

| Architecture | Work | Depth | Memory |
|---|---|---|---|
| RNN | $O(nd^2)$ | $O(n)$ | $O(d^2)$ |
| LSTM | $O(nd^2)$ | $O(n)$ | $O(d^2)$ |
| Transformer | $O(n^2 d + nd^2)$ | $O(1)$ | $O(n^2 + nd + d^2)$ |
| **LDRU** | $\mathbf{O(nd^2)}$ | $\mathbf{O(\log n)}$ | $\mathbf{O(nd + d^2)}$ |

Table 5: Complexity comparison across architectures.

We give the experimental setup to benchmark the practical scaling of the RNN, LDRU, and Transformer presented in Fig. 3. We used a batch size of 32, input size of 16, output size of 2 across all models. The RNN hidden size was set to 400. The LDRU used the same settings as in the regular task experiments, an embedding dim of 64. The Transformer used 3 layers, an embedding dimension of 64 and 8 attention heads. Setting these hyperparameters ensured that all models had a similar number of parameters (RNN: 168,002; LDRU: 162,498; Transformer: 150,338). We measured the wall-clock time for forward and backward passes across sequence lengths from 4 to 2048 on an NVIDIA RTX 6000 Ada Generation GPU. Runtimes exclude initialization overhead and were measured after warm-up. To obtain stable measurements, we averaged runtimes over 128 passes for each sequence length.

### D.1  PRACTICAL BATCH SCALING

To complement the theoretical complexity analysis and the additional computational experiments in the main text, we provide further benchmarks of the wall-clock runtime for the LDRU. We measured

both forward and backward pass times as a function of sequence length across a range of batch sizes to identify whether the empirical performance aligns with our theoretical predictions. We benchmarked the LDRU (with the same hyperparameters as the main experiments, see Appendix E) on an NVIDIA RTX 6000 Ada Generation GPU, measuring the wall-clock time for each pass while varying sequence length and batch size. Runtimes exclude initialization overhead and were measured after warm-up.

**Results** Fig. 9 shows that runtime increases stepwise with sequence length, reflecting the $\lceil \log_2(n) \rceil$ number of reduction steps required by the LDRU. This staircase pattern is apparent when testing at midpoints between the smaller powers of two, where the runtime remains flat until the next reduction step is introduced. At the largest batch sizes and sequence lengths, the GPU ran out of memory, preventing some measurements. The observed scaling is consistent with the LDRU's expected complexity, demonstrating efficient parallelization on GPUs.

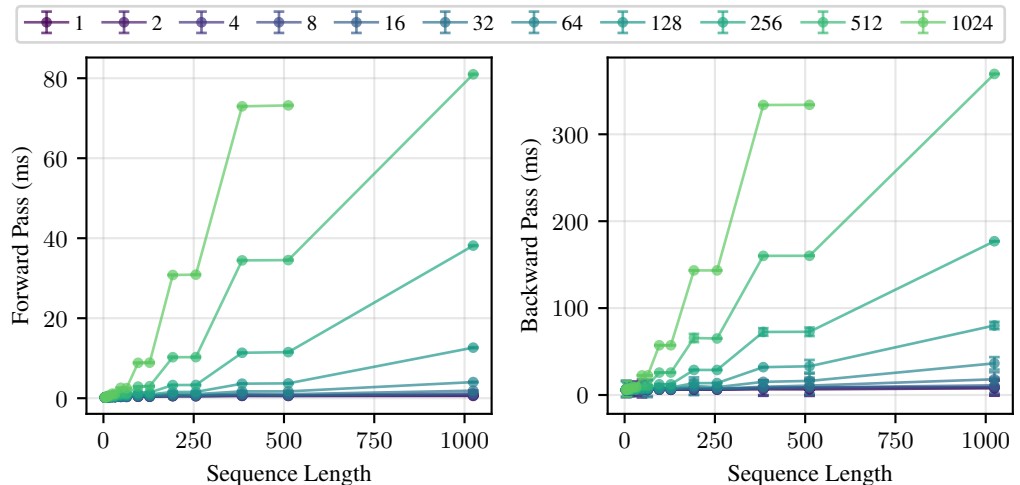

Figure 9: Forward (left) and backward (right) pass runtimes of the LDRU across varying sequence lengths and batch sizes. Points denote mean runtime over 64 passes; error bars indicate standard deviation. The legend indicates the batch size used to compute each point. The observed scaling is consistent with the theoretical $\lceil \log_2(n) \rceil$ complexity of the LDRU, demonstrating efficient parallelization on GPUs.

## E  REGULAR TASK EXPERIMENTAL DETAILS

This section provides detailed information about the experimental setup, including computing infrastructure, dataset generation procedures, and model hyperparameters to ensure reproducibility. Our codebase is fully implemented in JAX (Bradbury et al., 2018; DeepMind et al., 2020) in Haiku (Hennigan et al., 2020). The experiments were executed across different clusters consisting of NVIDIA RTX 6000 Ada Generation, NVIDIA L40S, and NVIDIA A100 80GB GPUs.

**Statistical Analysis** To assess the statistical significance of performance differences between the LDRU and baseline models across tasks, we performed single t-tests on baselines where the LDRU has 100.0% generalization with zero deviation; otherwise, we conducted a paired t-test comparing the OOD accuracies of both methods.

**Task Details** We present the task details and corresponding automata representations in Table 6 and Table 7. We present all tasks as Moore machines for consistency, as described in the main text. The tasks are selected from Delétang et al. (2023) (the regular tasks) and Bhattamishra et al. (2020), which are well-known benchmarks for evaluating length generalization in sequence processing models. The prefix languages evaluated can be constructed using the definition provided in Appendix B.

**Sequence Sampling** Parity Check sequences are sampled by generating binary sequences uniformly at random of length $n$ and determining the label using the sum of the sequence modulo 2. Likewise,

Table 6: Descriptions of the Even Pairs, Modular Arithmetic, Parity Check, and Cycle Navigation tasks with diagrams.

| Task | Description | Automaton |
|------|-------------|-----------|
| Even Pairs | Determine if input sequence has an even number of 01 and 10 pairs. | |
| Modular Arithmetic (mod 5) | Evaluate the input sequence modulo 5. | Presented separately in Fig. 10 due to its complexity. |
| Parity Check | Determine if input sequence has an even number of 1s. State 0 represents even parity, state 1 represents odd parity. | |
| Cycle Navigation | Navigate a cycle of 5 states based on the input sequence. | |

Even Pairs sequences are sampled in the same manner, but the label is determined by counting the number of 01 and 10 pairs in the sequence and checking if their sum is even. Cycle Navigation sequences are sampled with a uniform distribution over the alphabet $\{-1, 0, 1\}$, and the label is determined by summing the sequence modulo the cycle length (5). Sampling Modular Arithmetic sequences is different as we always sample valid arithmetic expressions where sequences at even index (so the first token, third token, etc.) are operands ($\{0, 1, 2, 3, 4\}$) and sequences at odd index are operators ($\{+, -, \times\}$). Determining the label is not as simple as evaluating the expression, as we cannot obey the usual orders of operations (a restriction of regular languages), so we evaluate the expression from left to right, ignoring operator precedence. The label is the result of the expression modulo 5.

We sample positive $D_n$ sequences using a modification to the algorithm presented in Arnold & Sleep (1980) that samples balanced parenthesis strings. We constrain the algorithm to obey the fixed depth $n$ and to close brackets at the end of the sequence to return to the initial state, ensuring that it is always a positive sequence. Within the training lengths, we further augment the algorithm with randomness to increase the diversity of the sequences. This randomness is introduced by modifying the probability of closing brackets at each step (when it is possible but not necessary to close a bracket) with noise sampled from $\mathcal{N}(0, 0.15)$ and a depth bias that decreases the probability of closing brackets as the depth increases. This ensures that the sequences are still positive while introducing variability. The depth bias is $0.1 \times \frac{\text{current\_depth}}{\text{max\_depth}}$. These augmentations only occur within

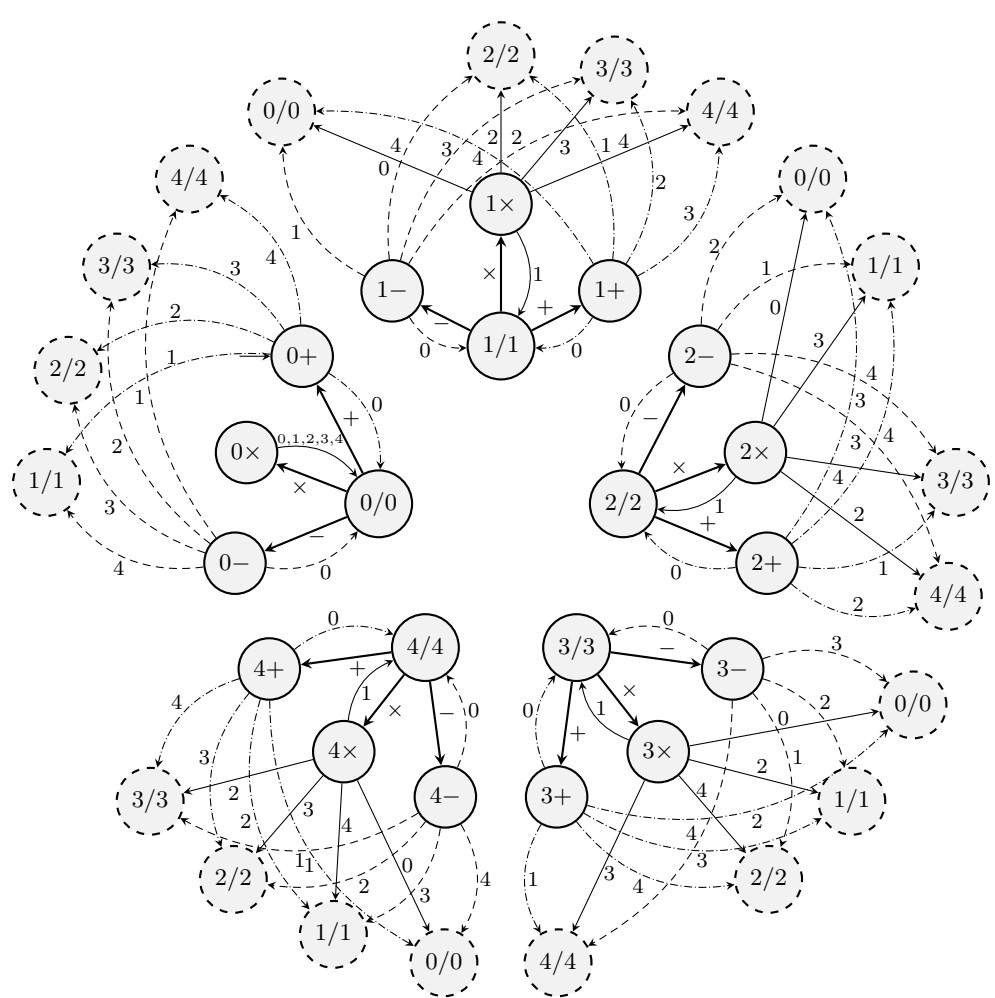

Figure 10: Moore machine for evaluating Modular Arithmetic modulo 5 expressions. The automaton's initial state is 0+. The sequences are constrained to valid arithmetic expressions, and the automaton handles multiplication, addition, and subtraction. Dashed states indicate symbolically linked states (i.e., the inner states) for presentation clarity.

the training lengths, so the test sequences are sampled using the constrained algorithm without randomness. Negative sequences are sampled by generating binary sequences uniformly at random and checking if the sequence is accepted. Therefore, the batches are not guaranteed to be balanced, but it is unlikely to be significantly unbalanced.

For the Tomita languages, we explicitly construct their DFAs (Table 7) and simulate their behavior to sample sequences. Again, we sample positive and negative sequences separately, but we oversample sequences when we cannot guarantee the acceptance (or rejection) of the sampled sequences. For Tomita 3, we constrained positive sequence sampling by disallowing the transition from state 3 to state 4, but this does not guarantee acceptance, so we oversampled by a factor of 2. When sampling negative sequences, we sampled symbols uniformly at random and checked if the sequence was accepted by the DFA (oversampling factor 2.5). For Tomita 4, we can guarantee the sampling of positive sequences by disallowing the transition from state 2 to state 3. However, we cannot guarantee the rejection of sequences, so we oversample negative sequences by a factor of 3. For Tomita 5, we oversample positive sequences by a factor of 5 and oversample negative sequences by a factor of 2. For Tomita 6, we oversample positive sequences by a factor of 4 and negative sequences by a factor of 2. For Tomita 7, we are able to guarantee the acceptance of positive sequences, but uniform sampling of the transitions would have caused low diversity in the sequences, so we bias

Table 7: Descriptions of the $D_n$ and Tomita tasks with diagrams. The Tomita languages are originally sourced from Tomita (1982) and their diagrams are adapted from Fdez. del Pozo Romero & Lago-Fernández (2023). All languages in this table are defined over the binary alphabet $\{0, 1\}$.

| Task | Description | Automaton |
|------|-------------|-----------|
| $D_n$ | Recognize sequences that belong to the regular expression $(0D_{n-1}^*1)^*$ where $D_1 = (01)^*$. The diagram recognizes $D_3$. |  |
| Tomita 3 | Recognize sequences where there is no odd-length 0 consecutive subsequence after an odd-length 1 consecutive subsequence. |  |
| Tomita 4 | Recognize sequences where 000 does not occur |  |
| Tomita 5 | Recognize sequences where there are an even number of 0s and 1s. |  |
| Tomita 6 | Recognize sequences where number of 1s$-$number of 0s mod $3 = 0$ |  |
| Tomita 7 | Recognize sequences that belong to the regular expression $0^*1^*0^*1^*$. |  |

the probabilities of taking transitions. We set the probability of a self-transition to $1 - \frac{4}{\max(\text{length},16)}$, where length is the sequence length. The probability to transitioning to the next state to $\frac{4}{\max(\text{length},16)}$. This ensures that the sequences are still positive while increasing diversity in the sequences. For negative sequences, we oversample by a factor of 5.

Sequences for $P_{p,q}$ tasks are sampled by generating a sequence of length $n$ with a uniform distribution over the alphabet $\{0, \ldots, q-1\}$. The label is determined by mapping the first $p$ symbols to the corresponding output class.

**Hyperparameters** We provide general training hyperparameters in Table 10 and task-specific training hyperparameters in Table 11. Model hyperparameters are detailed in Table 12. The model hyperparameters for the RNN, LSTM, and Transformer are the same as in Delétang et al. (2023) to take advantage of their extensive experimentation. However, their training hyperparameters differ from ours as we apply linear-warmup on the learning rate, $L_2$ regularization, and the AMSGrad op-

timizer instead of Adam. For RegularGPT hyperparameters, we ran a hyperparameter sweep using the Modular Arithmetic task. We used the range of hyperparameters given in Chi et al. (2023). The hyperparameter grid is shown in Table 8.

Tasks that did not converge by 100k steps for any architecture were trained for 1M steps for all architectures. Tasks that were trained for 1M steps were: Modular Arithmetic, $D_4$, $D_6$, $D_8$, and $D_{12}$. This allocation reflects the greater complexity of processing the underlying languages compared to simpler languages like the Tomita tasks and prefix languages.

For the LDRU, we maintained consistent hyperparameters within task families (e.g., one family is the $D_n$ tasks). We used a lower dropout (0.1) for the Delétang et al. (2023) family compared to the other tasks (0.25) because we hypothesize that the Modular Arithmetic task requires more capacity in $\odot_\theta$ than the other tasks.

Learning rates were assigned based on preliminary experiments and we typically applied a base learning rate of $1 \times 10^{-3}$ for tasks on 100k steps and $1 \times 10^{-4}$ for tasks on 1M steps (the only exceptions were: Modular Arithmetic with a learning rate of $1 \times 10^{-3}$ and $D_2$ and $D_3$ with a learning rate of $1 \times 10^{-4}$ to maintain learning rate consistency within task families). This was the case for the LDRU, RNN, LSTM, Transformer, and S5 architectures. However, we applied a learning rate of $1 \times 10^{-4}$ for all tasks for RegularGPT and SD-SSM because it improved convergence speed.

Table 8: RegularGPT hyperparameter sweep on Modular Arithmetic task. Grid search over optimizer, learning rate, and dropout probability with fixed architectural parameters.

| Parameter | Values |
|---|---|
| **Fixed Parameters** | |
| Embedding dimension | 256 |
| Number of heads | 8 |
| Chunk size | 2 |
| Shared weights | True |
| Thickness | 1 |
| **Varied Parameters** | |
| Optimizer | Adam, AMSGrad |
| Learning rate | $1 \times 10^{-4}, 3 \times 10^{-4}, 5 \times 10^{-4}$ |
| Dropout probability | 0.0, 0.1 |
| **Total configurations** | **12** |

Table 9: RegularGPT hyperparameter sweep results on Modular Arithmetic task. OOD accuracy for a single seed per configuration. Models trained for 250k steps on sequences up to length 40, evaluated on lengths 41–500. Note: main results use 1M steps. Best configuration highlighted in bold.

| Optimizer | Learning Rate | Dropout = 0.0 | Dropout = 0.1 |
|---|---|---|---|
| Adam | $1 \times 10^{-4}$ | 69.8 | 73.2 |
| Adam | $3 \times 10^{-4}$ | 72.2 | 49.2 |
| Adam | $5 \times 10^{-4}$ | 49.0 | 61.1 |
| AMSGrad | $1 \times 10^{-4}$ | **88.7** | 73.2 |
| AMSGrad | $3 \times 10^{-4}$ | 68.4 | 78.1 |
| AMSGrad | $5 \times 10^{-4}$ | 80.8 | 73.4 |

**Model Architecture Hyperparameters** We present the model hyperparameters for all baseline models and the LDRU in Table 12.

**Different Operator Choices** Here we describe the different operators we evaluate as a choice for $\odot_\theta$. We have described the MLP operator in the main text and Appendix C. Let $\mathbf{h}_i, \mathbf{h}_j$ be the embeddings of the two input tokens. We also evaluate the following operators:

Table 10: General training hyperparameters shared across all experiments.

| Parameter | Value |
|---|---|
| Optimizer | AMSGrad |
| Base learning rate | $1 \times 10^{-3}$ / $1 \times 10^{-4}$ ($D_n$ tasks; RegularGPT and SD-SSM) |
| Init learning rate | $1 \times 10^{-8}$ |
| Learning rate schedule | Linear warmup (20% of steps) |
| Weight decay | 0.0 |
| $L_2$ regularization | $5 \times 10^{-4}$ |
| Gradient clipping | 1.0 (global norm) |
| Centralized gradients | Yes |
| Batch size | 256 |
| Sequence length sampling | Uniform between 1 and max length |
| Max training length | 40 (standard) / 60, 100, 150 (length experiments) |
| Early stopping | None |
| Class balancing | Equal positive/negative examples per batch |
| Precision | Float32 |
| Seeds | 0, 1, 2 |

Table 11: Task-specific training hyperparameters showing training steps (all architectures) and dropout (LDRU only).

| Task | Training Steps | Dropout |
|---|---|---|
| *1) Delétang et al. (2023)* | | |
| Even Pairs | 100,000 | 0.1 |
| Modular Arithmetic | 1,000,000 | 0.1 |
| Parity Check | 100,000 | 0.1 |
| Cycle Navigation | 100,000 | 0.1 |
| *2) Bhattamishra et al. (2020)* | | |
| $D_2$ | 100,000 | 0.25 |
| $D_3$ | 100,000 | 0.25 |
| $D_4$ | 1,000,000 | 0.25 |
| $D_6$ | 1,000,000 | 0.25 |
| $D_8$ | 1,000,000 | 0.25 |
| $D_{12}$ | 1,000,000 | 0.25 |
| Tomita 3 | 100,000 | 0.25 |
| Tomita 4 | 100,000 | 0.25 |
| Tomita 5 | 100,000 | 0.25 |
| Tomita 6 | 100,000 | 0.25 |
| Tomita 7 | 100,000 | 0.25 |
| *3) Prefix languages (Ours)* | | |
| $P_{1,2}$ | 100,000 | 0.25 |
| $P_{2,2}$ | 100,000 | 0.25 |
| $P_{4,2}$ | 100,000 | 0.25 |
| $P_{1,4}$ | 100,000 | 0.25 |
| $P_{2,4}$ | 100,000 | 0.25 |
| $P_{4,4}$ | 100,000 | 0.25 |

Table 12: Model architecture hyperparameters for all baseline models and LDRU. Note that while dropout can be applied to RegularGPT, the hyperparameter sweep indicated that zero dropout is better for Modular Arithmetic, so we did not use it. Total parameters rounded to 2 significant figures.

| Component | Parameter | RNN | LSTM | Transformer | S5 | SD-SSM | RegularGPT | LDRU |
|---|---|---|---|---|---|---|---|---|
| Embedding | Embedding dim | None | None | 64 | None | None | 256 | 64 |
| | Vocab size | Task-dependent | Task-dependent | Task-dependent | Task-dependent | Task-dependent | Task-dependent | Task-dependent |
| | Initialization | – | – | $\mathcal{N}(0, 0.02)$ | – | – | $\mathcal{N}(0, 0.02)$ | $\mathcal{N}(0, 0.02)$ |
| Core Architecture | Layers/Blocks | 1 | 1 | 5 | 2 | 1 | 1 | 1 |
| | Hidden dim | 256 | 256 | 64 | 256 | 256 | 256 | 64 |
| Residual Connections | Dimension | – | – | 256 | – | – | 1024 | 256 |
| Normalization | Layer norm | No | No | Pre-norm & Post-norm | Post-norm | Post-norm | Pre-norm & Post-norm | Post-norm |
| LDRU Operator | MLP hidden dims | – | – | – | – | – | – | $128 \rightarrow 256 \rightarrow 64$ |
| | Activation | – | – | – | – | – | – | ReLU |
| | MLP initialization | – | – | – | – | – | – | Glorot (Glorot & Bengio, 2010) |
| | Projection initialization | – | – | – | – | – | – | Identity |
| S5 | Size base | – | – | – | 256 | – | – | – |
| | Num blocks | – | – | – | 8 | – | – | – |
| | Activation fn | – | – | – | GELU | – | – | – |
| | Mode | – | – | – | Mean pooling | – | – | – |
| | Clipped Eigs | – | – | – | Yes | – | – | – |
| | Discretization | – | – | – | ZOH | – | – | – |
| SD-SSM | Transition matrices | – | – | – | – | 8 | – | – |
| | $L_p$ norm | – | – | – | – | 1.2 | – | – |
| Transformer | Attention heads | – | – | 8 | – | – | 8 | – |
| | Head dimension | – | – | 8 | – | – | 32 | – |
| | Positional encoding | – | – | ALiBi/None/~RoPE | – | – | None | – |
| RegularGPT | Chunk size | – | – | – | – | – | 2 | – |
| | Shared weights | – | – | – | – | – | Yes | – |
| | Dilated attention | – | – | – | – | – | Yes | – |
| Output | Classifier type | Linear | Linear | Linear | Linear | Linear | Linear | Linear |
| | Output classes | Task-dependent | Task-dependent | Task-dependent | Task-dependent | Task-dependent | Task-dependent | Task-dependent |
| Regularization | Dropout locations | – | – | Attention & Residual | Activation | – | Attention & Residual | Post-norm |
| **Total Parameters** | **Approx.** | **67k** | **270k** | **250k** | **270k** | **530k** | **3200k** | **160k** |

**Element-wise Sum (Elem. Sum)** The element-wise sum of the two embeddings, $\mathbf{h}_i + \mathbf{h}_j$.

**Linear** A linear projection of the concatenated embeddings, i.e, $\mathbf{W}([\mathbf{h}_i; \mathbf{h}_j]) + \mathbf{b}$, where $\mathbf{W}$ is a learned weight matrix and $\mathbf{b}$ is a learned bias vector.

**Gated Sum** A gated sum of the two embeddings. We determine the gates using a linear projection of the concatenated embeddings, i.e., $\mathbf{g} = \sigma(\mathbf{W}_g([\mathbf{h}_i; \mathbf{h}_j]) + \mathbf{b}_g)$, where $\sigma$ is the sigmoid activation function. The output is then $\mathbf{g} \circ \mathbf{h}_i + (1 - \mathbf{g}) \circ \mathbf{h}_j$.

### E.1 OPTIMIZER ABLATION

**Experimental Setup** We use the Modular Arithmetic task to compare the OOD performance of the LDRU when training with the Adam and AMSGrad optimizers with and without dropout. The experiments were conducted with fixed hyperparameters (except for the optimizer algorithm and dropout rate) found in Table 10 and Table 12. We train the models for 1M steps on sequences up to length 40 and evaluate them on sequences of lengths 41–500. The results are averaged over 10 seeds.

**Results** We present the results of this ablation in Table 13 and Fig. 11. With both optimizers, dropout improves generalization, as the models trained without dropout (both Adam and AMSGrad) perform worse on the OOD test set. Adam without dropout demonstrated catastrophic forgetting on seed 9, indicating that it is more unstable compared to AMSGrad. AMSGrad with dropout achieves the best performance, indicating robustness when handling the Modular Arithmetic task.

While the experiment was training, we evaluated a validation batch of 1024 sequences of length 500 every 1000 steps: the performance on this batch is how we determined the validation loss (Val. Loss) and accuracy (Val. Acc.) presented in Fig. 11. This evaluation gave us an idea of how the model would perform on the most OOD sequences during training, and we found it to be a reliable indicator of the model's generalization across all the sequences up to length 500. The test accuracy (Test Acc.) is the performance on the test set of sampled 512 sequences per length, from 1 to 500. The $\Delta$Log Loss is the difference between the log of the validation loss and the log of the training loss, which indicates how well the model generalizes to OOD sequences and overfits to the training sequences. We also note that the validation loss is evaluated on the model parameters without dropout, while the training loss includes dropout. The $\Delta$Log Loss reveals an interesting difference between AMSGrad with and without dropout: with dropout indicates improving performance on the validation set during training, while without dropout indicates that the model overfits to the training sequences without improving on the validation set.

Table 13: OOD accuracies when training the LDRU with different optimizers and dropout combinations on the Modular Arithmetic task. Individual seed results with mean and standard deviation across 10 seeds. The results demonstrate that dropout is necessary for generalization, with AMSGrad showing superior performance when combined with dropout. We note that the AMSGrad optimizer without dropout achieves competitive OOD accuracy compared to the optimizers with dropout.

| Seed | (Adam, 0.0) | (Adam, 0.1) | (AMSGrad, 0.0) | (AMSGrad, 0.1) |
|---|---|---|---|---|
| 0 | 99.606 | 99.963 | 99.963 | 100.000 |
| 1 | 99.691 | 99.980 | 99.949 | 99.997 |
| 2 | 99.888 | 99.986 | 99.888 | 100.000 |
| 3 | 99.684 | 99.959 | 99.946 | 100.000 |
| 4 | 99.113 | 99.990 | 99.963 | 99.997 |
| 5 | 95.143 | 99.993 | 99.898 | 99.997 |
| 6 | 99.966 | 99.997 | 99.868 | 99.986 |
| 7 | 98.689 | 99.997 | 99.963 | 100.000 |
| 8 | 99.925 | 99.868 | 99.983 | 99.993 |
| 9 | 40.231 | 99.976 | 99.864 | 100.000 |
| **Mean ± Std** | **93.194 ± 17.707** | **99.971 ± 0.037** | **99.928 ± 0.042** | **99.997 ± 0.004** |

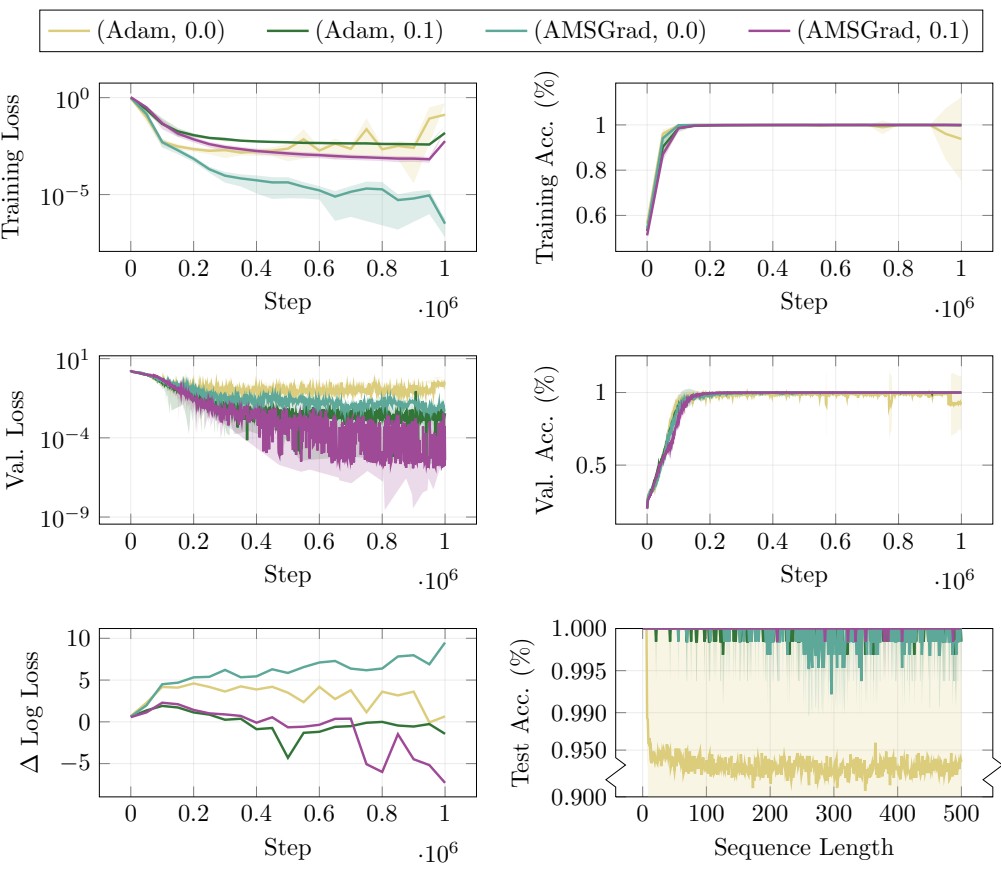

Figure 11: Optimizer ablation study comparing Adam and AMSGrad performance on LDRU models. Standard deviations are shown as the filled area around the mean values. The experiments show that dropout is necessary to enable generalization outside of the training distribution. It is notable that the AMSGrad optimizer without dropout achieves the lowest cross-entropy training loss but has worse performance outside the training distribution. Note the broken axis on the y-axis for the test accuracy (Test Acc.) plot to better highlight the performance differences between the highly performing models.

# F  ALL REGULAR TASK RESULTS

**All Baselines** We present the OOD accuracies of all baseline models on all tasks in Table 14 under the experimental settings for **RQ1** (main text). The difference between Table 14 and Table 2 is that the former includes all baselines, while the latter does not include S5 and SD-SSM. We use the same statistical significance notation as in Table 2.

Table 14: OOD accuracy results across 21 regular tasks. All models are trained on sequences with lengths 1–40 and evaluated on lengths 41–500. Results show mean $\pm$ standard deviation across seeds. $\dagger$ indicates statistical significance of LDRU improvements over baselines (i.e., $p < 0.05$).

| Task | RNN | LSTM | Transformer | TF (ALiBi) | TF ($\sim$RoPE) | S5 | SD-SSM | RegularGPT | LDRU |
|---|---|---|---|---|---|---|---|---|---|
| *1) Delétang et al. (2023)* | | | | | | | | | |
| Even Pairs | $77.4 \pm 12.2^{\ddagger}$ | $100.0 \pm 0.0$ | $50.3 \pm 0.0^{\dagger}$ | $61.1 \pm 5.1^{\dagger}$ | $91.5 \pm 7.9$ | $53.9 \pm 0.8^{\dagger}$ | $65.7 \pm 0.8^{\dagger}$ | $91.9 \pm 0.7^{\dagger}$ | $\mathbf{100.0 \pm 0.0}$ |
| Modular Arithmetic | $100.0 \pm 0.0$ | $100.0 \pm 0.0$ | $76.0 \pm 0.0^{\dagger}$ | $54.4 \pm 4.5^{\dagger}$ | $31.4 \pm 5.5^{\dagger}$ | $47.6 \pm 6.7^{\dagger}$ | $99.9 \pm 0.0^{\dagger}$ | $99.1 \pm 1.3$ | $\mathbf{100.0 \pm 0.0}$ |
| Parity Check | $100.0 \pm 0.0$ | $100.0 \pm 0.0$ | $50.1 \pm 0.0^{\dagger}$ | $49.9 \pm 0.0^{\dagger}$ | $49.9 \pm 0.0^{\dagger}$ | $50.1 \pm 0.1^{\dagger}$ | $71.4 \pm 1.4^{\dagger}$ | $99.8 \pm 0.7$ | $\mathbf{100.0 \pm 0.0}$ |
| Cycle Navigation | $100.0 \pm 0.0$ | $60.3 \pm 2.4^{\dagger}$ | $20.0 \pm 0.0^{\dagger}$ | $21.6 \pm 0.7^{\dagger}$ | $23.3 \pm 1.5^{\dagger}$ | $22.9 \pm 0.7^{\dagger}$ | $44.2 \pm 0.8^{\dagger}$ | $99.9 \pm 0.2$ | $\mathbf{100.0 \pm 0.0}$ |
| *2) Bhattamishra et al. (2020)* | | | | | | | | | |
| $D_2$ | $100.0 \pm 0.0$ | $100.0 \pm 0.0$ | $100.0 \pm 0.0^{\dagger}$ | $100.0 \pm 0.0$ | $98.2 \pm 1.7$ | $88.4 \pm 10.3$ | $100.0 \pm 0.0$ | $93.7 \pm 5.5$ | $\mathbf{100.0 \pm 0.0}$ |
| $D_3$ | $100.0 \pm 0.0$ | $97.7 \pm 4.0$ | $99.9 \pm 0.0^{\dagger}$ | $98.3 \pm 2.6$ | $92.5 \pm 10.2$ | $83.6 \pm 9.3$ | $96.3 \pm 3.9$ | $87.6 \pm 4.6^{\dagger}$ | $\mathbf{100.0 \pm 0.0}$ |
| $D_4$ | $95.0 \pm 8.7$ | $100.0 \pm 0.0$ | $99.5 \pm 0.0^{\dagger}$ | $96.0 \pm 4.7$ | $97.6 \pm 0.4^{\dagger}$ | $81.7 \pm 10.3$ | $97.3 \pm 2.3$ | $94.1 \pm 4.7$ | $\mathbf{100.0 \pm 0.0}$ |
| $D_6$ | $97.1 \pm 4.7$ | $96.5 \pm 0.0$ | — | $92.3 \pm 6.8$ | $98.4 \pm 1.0$ | $75.0 \pm 1.5^{\dagger}$ | $92.2 \pm 4.3$ | $87.9 \pm 3.7^{\dagger}$ | $\mathbf{98.5 \pm 2.4}$ |
| $D_8$ | $\mathbf{99.2 \pm 1.3}$ | $89.0 \pm 4.3^{\ddagger}$ | $89.0 \pm 0.2^{\dagger}$ | $89.7 \pm 7.7$ | $98.7 \pm 0.8$ | $73.7 \pm 0.5^{\dagger}$ | $87.2 \pm 2.9$ | $91.0 \pm 2.8^{\dagger}$ | $98.1 \pm 1.6$ |
| $D_{12}$ | $92.7 \pm 9.5$ | $82.1 \pm 1.7^{\dagger}$ | $81.3 \pm 0.3^{\dagger}$ | $84.4 \pm 9.6$ | $\mathbf{98.7 \pm 1.0}$ | $73.5 \pm 0.3^{\dagger}$ | $72.9 \pm 4.3^{\dagger}$ | $89.8 \pm 4.1^{\dagger}$ | $96.0 \pm 2.1^{\dagger}$ |
| Tomita 3 | $100.0 \pm 0.0$ | $100.0 \pm 0.0$ | $98.7 \pm 0.0^{\dagger}$ | $100.0 \pm 0.0^{\dagger}$ | $70.7 \pm 8.9^{\dagger}$ | $72.8 \pm 1.6^{\dagger}$ | $100.0 \pm 0.0$ | $93.7 \pm 3.3$ | $\mathbf{100.0 \pm 0.0}$ |
| Tomita 4 | $100.0 \pm 0.0$ | $100.0 \pm 0.0$ | $96.1 \pm 0.0^{\dagger}$ | $100.0 \pm 0.0$ | $85.5 \pm 2.1^{\dagger}$ | $72.3 \pm 12.8$ | $100.0 \pm 0.0$ | $98.4 \pm 1.5$ | $\mathbf{100.0 \pm 0.0}$ |
| Tomita 5 | $100.0 \pm 0.0$ | $100.0 \pm 0.0$ | $74.3 \pm 0.0^{\dagger}$ | $74.3 \pm 0.0^{\dagger}$ | $74.3 \pm 0.0^{\dagger}$ | $65.8 \pm 3.2^{\dagger}$ | $81.1 \pm 1.8^{\dagger}$ | $97.7 \pm 2.2$ | $\mathbf{100.0 \pm 0.0}$ |
| Tomita 6 | $100.0 \pm 0.0$ | $100.0 \pm 0.0$ | $50.2 \pm 0.0^{\dagger}$ | $50.0 \pm 0.0^{\dagger}$ | $50.0 \pm 0.0^{\dagger}$ | $49.9 \pm 0.1^{\dagger}$ | $85.6 \pm 3.7^{\dagger}$ | $92.2 \pm 4.7$ | $\mathbf{100.0 \pm 0.0}$ |
| Tomita 7 | $100.0 \pm 0.0$ | $100.0 \pm 0.0$ | $100.0 \pm 0.0$ | $100.0 \pm 0.0$ | $100.0 \pm 0.0$ | $98.7 \pm 2.2$ | $100.0 \pm 0.0$ | $100.0 \pm 0.0$ | $\mathbf{100.0 \pm 0.0}$ |
| *3) Prefix languages (Ours)* | | | | | | | | | |
| $P_{1,2}$ | $82.2 \pm 3.3^{\dagger}$ | $100.0 \pm 0.0$ | $50.3 \pm 0.0^{\dagger}$ | $56.0 \pm 2.3^{\dagger}$ | $89.2 \pm 12.7$ | $52.9 \pm 2.8^{\dagger}$ | $61.8 \pm 3.5^{\dagger}$ | $100.0 \pm 0.0$ | $\mathbf{100.0 \pm 0.0}$ |
| $P_{2,2}$ | $65.4 \pm 24.0^{\dagger}$ | $100.0 \pm 0.0$ | $25.2 \pm 0.0^{\dagger}$ | $93.0 \pm 5.6$ | $86.4 \pm 5.0^{\dagger}$ | $29.3 \pm 1.8^{\dagger}$ | $34.0 \pm 1.6^{\dagger}$ | $99.6 \pm 0.3$ | $\mathbf{100.0 \pm 0.0}$ |
| $P_{4,2}$ | $99.4 \pm 1.0$ | $100.0 \pm 0.0$ | $6.3 \pm 0.0^{\dagger}$ | $90.0 \pm 10.3$ | $58.5 \pm 11.1^{\dagger}$ | $12.1 \pm 2.1^{\dagger}$ | $99.8 \pm 0.2$ | $91.5 \pm 2.1^{\dagger}$ | $\mathbf{100.0 \pm 0.0}$ |
| $P_{1,4}$ | $61.4 \pm 15.7^{\dagger}$ | $100.0 \pm 0.0$ | $25.2 \pm 0.0^{\dagger}$ | $40.3 \pm 2.4^{\dagger}$ | $79.5 \pm 7.0^{\dagger}$ | $31.8 \pm 3.6^{\dagger}$ | $38.7 \pm 0.5^{\dagger}$ | $100.0 \pm 0.0$ | $\mathbf{100.0 \pm 0.0}$ |
| $P_{2,4}$ | $96.9 \pm 2.5$ | $100.0 \pm 0.0$ | $6.3 \pm 0.0^{\dagger}$ | $31.0 \pm 8.1^{\dagger}$ | $51.6 \pm 7.6^{\dagger}$ | $14.4 \pm 1.3^{\dagger}$ | $30.2 \pm 4.9^{\dagger}$ | $99.7 \pm 0.2$ | $\mathbf{100.0 \pm 0.0}$ |
| $P_{4,4}$ | $85.0 \pm 15.0^{\ddagger}$ | $97.3 \pm 2.2^{\ddagger}$ | $0.4 \pm 0.0^{\dagger}$ | $21.3 \pm 7.6^{\dagger}$ | $58.2 \pm 2.7^{\dagger}$ | $8.0 \pm 3.4^{\dagger}$ | $42.9 \pm 2.5^{\dagger}$ | $95.3 \pm 0.6^{\dagger}$ | $\mathbf{100.0 \pm 0.0}$ |

**Individual Seed Results for Main Tasks** We present the individual seed results for all 21 regular language tasks in Table 15. The models were trained on sequences up to length 40 and evaluated on sequences of lengths 41–500.

Table 15: Individual seed results for all 21 regular language tasks. Models trained on sequences up to length 40, evaluated on lengths 41–500.

| Task | RNN 0 | RNN 1 | RNN 2 | LSTM 0 | LSTM 1 | LSTM 2 | TF (NoPE) 0 | TF (NoPE) 1 | TF (NoPE) 2 | TF (ALiBi) 0 | TF (ALiBi) 1 | TF (ALiBi) 2 | TF ($\sim$RoPE) 0 | TF ($\sim$RoPE) 1 | TF ($\sim$RoPE) 2 | S5 0 | S5 1 | S5 2 | SD-SSM 0 | SD-SSM 1 | SD-SSM 2 | RegularGPT 0 | RegularGPT 1 | RegularGPT 2 | LDRU 0 | LDRU 1 | LDRU 2 |
|---|---|---|---|---|---|---|---|---|---|---|---|---|---|---|---|---|---|---|---|---|---|---|---|---|---|---|---|
| *1) Delétang et al. (2023)* | | | | | | | | | | | | | | | | | | | | | | | | | | | |
| Even Pairs | 82.4 | 86.3 | 63.4 | 100.0 | 100.0 | 100.0 | 50.3 | 50.3 | 50.3 | 66.0 | 55.9 | 61.5 | 95.0 | 82.4 | 97.0 | 53.9 | 53.2 | 54.7 | 66.5 | 65.7 | 64.8 | 92.2 | 91.1 | 92.2 | 100.0 | 100.0 | 100.0 |
| Modular Arithmetic | 100.0 | 100.0 | 100.0 | 100.0 | 100.0 | 100.0 | 76.0 | 76.0 | 76.0 | 55.4 | 58.3 | 49.5 | 33.3 | 35.8 | 25.2 | 51.5 | 51.4 | 39.9 | 99.9 | 99.9 | 99.9 | 97.5 | 99.9 | 99.8 | 100.0 | 100.0 | 100.0 |
| Parity Check | 100.0 | 100.0 | 100.0 | 100.0 | 100.0 | 100.0 | 50.0 | 50.1 | 50.1 | 49.9 | 49.9 | 49.9 | 49.9 | 49.9 | 50.0 | 50.2 | 50.1 | 49.9 | 69.8 | 72.3 | 71.9 | 99.4 | 100.0 | 100.0 | 100.0 | 100.0 | 100.0 |
| Cycle Navigation | 100.0 | 100.0 | 100.0 | 62.1 | 57.5 | 61.2 | 20.0 | 20.0 | 20.0 | 20.8 | 22.0 | 21.9 | 21.6 | 24.0 | 24.3 | 22.1 | 23.6 | 23.0 | 45.0 | 44.3 | 43.4 | 99.9 | 100.0 | 99.7 | 100.0 | 100.0 | 100.0 |
| *2) Bhattamishra et al. (2020)* | | | | | | | | | | | | | | | | | | | | | | | | | | | |
| $D_2$ | 100.0 | 100.0 | 100.0 | 100.0 | 100.0 | 100.0 | 100.0 | 100.0 | 100.0 | 100.0 | 100.0 | 100.0 | 99.5 | 96.3 | 98.8 | 85.0 | 100.0 | 80.2 | 100.0 | 100.0 | 100.0 | 100.0 | 89.7 | 91.5 | 100.0 | 100.0 | 100.0 |
| $D_3$ | 100.0 | 100.0 | 100.0 | 99.9 | 99.9 | 99.9 | 99.9 | 99.9 | 99.9 | 95.3 | 100.0 | 99.5 | 98.3 | 98.5 | 80.8 | 79.4 | 94.3 | 77.2 | 100.0 | 96.8 | 92.1 | 92.7 | 83.7 | 86.4 | 100.0 | 100.0 | 100.0 |
| $D_4$ | 85.0 | 100.0 | 100.0 | 100.0 | 100.0 | 100.0 | 99.5 | 99.5 | 99.5 | 100.0 | 90.7 | 97.2 | 97.4 | 97.4 | 98.1 | 75.5 | 93.6 | 76.1 | 100.0 | 96.3 | 95.7 | 89.1 | 94.6 | 98.5 | 100.0 | 95.7 | 99.9 |
| $D_6$ | 91.6 | 99.6 | 100.0 | 100.0 | 98.5 | 96.8 | 96.6 | 96.6 | 96.5 | 93.1 | 85.1 | 98.7 | 98.7 | 97.2 | 99.1 | 73.7 | 76.6 | 74.7 | 97.1 | 89.4 | 90.4 | 90.7 | 83.7 | 89.3 | 99.9 | 95.7 | 99.9 |
| $D_8$ | 100.0 | 97.7 | 99.9 | 85.4 | 93.8 | 87.8 | 89.0 | 89.3 | 88.8 | 97.6 | 89.4 | 82.1 | 99.3 | 97.7 | 99.0 | 74.2 | 73.2 | 73.6 | 84.5 | 86.6 | 90.4 | 93.6 | 88.1 | 91.4 | 99.8 | 98.2 | 96.6 |
| $D_{12}$ | 96.9 | 81.8 | 99.3 | 80.2 | 82.6 | 83.4 | 81.1 | 81.3 | 81.6 | 95.3 | 77.6 | 80.2 | 99.3 | 99.3 | 97.6 | 73.9 | 73.4 | 73.3 | 77.7 | 69.4 | 71.6 | 93.5 | 90.5 | 85.3 | 97.5 | 96.8 | 93.6 |
| Tomita 3 | 100.0 | 100.0 | 100.0 | 100.0 | 100.0 | 100.0 | 98.7 | 98.7 | 98.7 | 100.0 | 100.0 | 100.0 | 68.6 | 80.5 | 63.1 | 74.3 | 71.0 | 73.1 | 100.0 | 100.0 | 100.0 | 96.8 | 94.2 | 90.2 | 100.0 | 100.0 | 100.0 |
| Tomita 4 | 100.0 | 100.0 | 100.0 | 100.0 | 100.0 | 100.0 | 96.1 | 96.1 | 96.1 | 100.0 | 100.0 | 100.0 | 83.1 | 86.5 | 86.8 | 58.5 | 83.7 | 74.8 | 100.0 | 100.0 | 100.0 | 99.5 | 98.2 | 95.3 | 100.0 | 100.0 | 100.0 |
| Tomita 5 | 100.0 | 100.0 | 100.0 | 100.0 | 100.0 | 100.0 | 74.3 | 74.3 | 74.3 | 74.3 | 74.3 | 74.3 | 74.3 | 74.3 | 74.3 | 63.0 | 65.3 | 69.2 | 82.7 | 81.4 | 79.2 | 99.5 | 98.2 | 95.3 | 100.0 | 100.0 | 100.0 |
| Tomita 6 | 100.0 | 100.0 | 100.0 | 100.0 | 100.0 | 100.0 | 50.2 | 50.2 | 50.2 | 50.0 | 50.0 | 50.0 | 50.0 | 50.0 | 50.0 | 50.0 | 50.0 | 49.7 | 82.6 | 84.6 | 89.7 | 93.2 | 87.1 | 96.4 | 100.0 | 100.0 | 100.0 |
| Tomita 7 | 100.0 | 100.0 | 100.0 | 100.0 | 100.0 | 100.0 | 100.0 | 100.0 | 100.0 | 100.0 | 100.0 | 100.0 | 100.0 | 100.0 | 100.0 | 96.2 | 100.0 | 100.0 | 100.0 | 100.0 | 100.0 | 100.0 | 100.0 | 100.0 | 100.0 | 100.0 | 100.0 |
| *3) Prefix languages (Ours)* | | | | | | | | | | | | | | | | | | | | | | | | | | | |
| $P_{1,2}$ | 78.4 | 84.3 | 84.0 | 100.0 | 100.0 | 100.0 | 50.3 | 50.3 | 50.3 | 58.7 | 54.2 | 55.2 | 97.8 | 95.1 | 74.6 | 56.1 | 50.6 | 52.1 | 65.8 | 60.2 | 59.4 | 100.0 | 100.0 | 100.0 | 100.0 | 100.0 | 100.0 |
| $P_{2,2}$ | 43.4 | 61.8 | 91.0 | 100.0 | 100.0 | 100.0 | 25.2 | 25.2 | 25.2 | 86.5 | 97.0 | 95.5 | 87.1 | 91.0 | 81.2 | 30.6 | 27.2 | 30.0 | 32.2 | 34.7 | 35.2 | 99.3 | 99.8 | 99.9 | 100.0 | 100.0 | 100.0 |
| $P_{4,2}$ | 100.0 | 100.0 | 98.2 | 100.0 | 100.0 | 100.0 | 6.3 | 6.3 | 6.3 | 78.0 | 96.2 | 95.6 | 62.0 | 67.5 | 46.0 | 14.5 | 11.2 | 10.6 | 99.9 | 99.5 | 100.0 | 93.0 | 89.1 | 92.4 | 100.0 | 100.0 | 100.0 |
| $P_{1,4}$ | 78.4 | 57.0 | 47.8 | 100.0 | 100.0 | 100.0 | 25.2 | 25.2 | 25.2 | 38.8 | 39.0 | 43.1 | 86.8 | 72.9 | 78.9 | 28.6 | 35.7 | 31.1 | 38.3 | 39.3 | 38.5 | 100.0 | 100.0 | 100.0 | 100.0 | 100.0 | 100.0 |
| $P_{2,4}$ | 94.1 | 97.7 | 98.9 | 100.0 | 100.0 | 100.0 | 6.3 | 6.3 | 6.3 | 27.9 | 40.2 | 24.8 | 59.9 | 49.7 | 45.1 | 15.5 | 13.0 | 14.7 | 27.6 | 27.0 | 35.9 | 99.5 | 99.7 | 99.8 | 100.0 | 100.0 | 100.0 |
| $P_{4,4}$ | 68.0 | 90.0 | 96.8 | 98.4 | 94.7 | 98.8 | 0.4 | 0.4 | 0.4 | 30.1 | 17.4 | 16.5 | 60.0 | 59.4 | 55.1 | 5.8 | 11.9 | 6.2 | 43.3 | 40.2 | 45.2 | 95.1 | 95.9 | 94.8 | 100.0 | 100.0 | 100.0 |

**Extended Sequence Length Results** We present the individual seed results for the sequence length experiments on $D_n$ languages in Table 16. The models were trained on sequences up to length $x \in [40, 60, 100, 150]$ and evaluated on sequences of length from $x + 1$ to 500.

**Operator Choice Results** We present the individual seed results for the **RQ3** experiments in Table 17. The models were trained on sequences up to length 40 and evaluated on sequences of lengths 41–500. The results show that the MLP operator achieves 100% OOD accuracy on all tasks. The other operators show varying performance—but none achieve over 70% OOD accuracy on Modular Arithmetic.

Table 16: Individual seed results for sequence length experiments on $D_n$ languages.

| | | RNN | | | LDRU | | |
|---|---|---|---|---|---|---|---|
| Task | Max Length | 0 | 1 | 2 | 0 | 1 | 2 |
| $D_4$ | 40 | 85.0 | 100.0 | 100.0 | 100.0 | 100.0 | 100.0 |
| | 60 | 100.0 | 100.0 | 100.0 | 100.0 | 100.0 | 100.0 |
| | 100 | 100.0 | 100.0 | 100.0 | 100.0 | 100.0 | 100.0 |
| | 150 | 100.0 | 100.0 | 100.0 | 100.0 | 100.0 | 100.0 |
| $D_6$ | 40 | 91.6 | 99.6 | 100.0 | 99.9 | 95.7 | 99.9 |
| | 60 | 99.9 | 98.8 | 99.9 | 100.0 | 100.0 | 100.0 |
| | 100 | 100.0 | 100.0 | 100.0 | 100.0 | 100.0 | 100.0 |
| | 150 | 100.0 | 100.0 | 100.0 | 100.0 | 100.0 | 100.0 |
| $D_8$ | 40 | 100.0 | 97.7 | 99.9 | 99.8 | 98.2 | 96.6 |
| | 60 | 99.9 | 100.0 | 100.0 | 98.8 | 91.5 | 99.9 |
| | 100 | 100.0 | 100.0 | 100.0 | 100.0 | 100.0 | 100.0 |
| | 150 | 100.0 | 100.0 | 100.0 | 100.0 | 100.0 | 100.0 |
| $D_{12}$ | 40 | 96.9 | 81.8 | 99.3 | 97.5 | 96.8 | 93.6 |
| | 60 | 81.8 | 82.1 | 98.1 | 96.1 | 98.2 | 96.5 |
| | 100 | 91.3 | 77.8 | 92.6 | 99.9 | 99.9 | 99.9 |
| | 150 | 99.4 | 99.5 | 100.0 | 99.9 | 99.9 | 99.9 |

Table 17: Individual seed results for operator study on the regular Delétang et al. (2023) tasks. All models were trained with a max sequence length of 40.

| | MLP | | | Elem. Sum | | | Concat. Proj. | | | Gated Sum | | |
|---|---|---|---|---|---|---|---|---|---|---|---|---|
| Task | 0 | 1 | 2 | 0 | 1 | 2 | 0 | 1 | 2 | 0 | 1 | 2 |
| Even Pairs | 100.0 | 100.0 | 100.0 | 51.9 | 51.8 | 51.9 | 100.0 | 100.0 | 100.0 | 100.0 | 100.0 | 100.0 |
| Modular Arith. | 100.0 | 100.0 | 100.0 | 32.4 | 32.6 | 32.9 | 59.4 | 60.3 | 62.7 | 68.7 | 68.8 | 63.7 |
| Parity Check | 100.0 | 100.0 | 100.0 | 100.0 | 100.0 | 100.0 | 100.0 | 100.0 | 100.0 | 100.0 | 100.0 | 100.0 |
| Cycle Nav. | 100.0 | 100.0 | 100.0 | 100.0 | 100.0 | 100.0 | 100.0 | 100.0 | 100.0 | 100.0 | 100.0 | 100.0 |

# G  ASSOCIATIVITY REGULARIZATION

Our results on the regular tasks indicate that the LDRU can learn to approximate an associative operator $\odot_\theta$ without explicit regularization. However, we hypothesize that adding an associativity regularization term to the training loss could further improve the model's ability to learn an associative operator, especially in low-data regimes or on non-regular tasks. Associativity encourages the operator to behave as a recurrence, which we believe is beneficial for generalization. To test this hypothesis, we added an associativity regularization term to the training loss during experiments on the ListOps experiments and the natural language tasks described in Appendix H.

We compute this additional loss term by taking every locally valid triple $(h_a, h_b, h_c)$ computed during the forward pass of an LDRU layer and punish deviations from associativity. For each step $k$ in the LDRU layer's reduction, we partition the sequence of token embeddings into these triples and we evaluate the following expressions:

$$x = \odot_\theta(\odot_\theta(\mathbf{h}_a, \mathbf{h}_b), \mathbf{h}_c); y = \odot_\theta(\mathbf{h}_a, \odot_\theta(\mathbf{h}_b, \mathbf{h}_c)), \tag{10}$$

which enables us to compute the associativity loss as:

$$\ell_{assoc}(x, y) = (1 - \frac{x \cdot y}{|x||y| + \epsilon})^2. \tag{11}$$

Let $\tau_k$ be the set of valid triples evaluated at step $k$. The associativity loss for step $k$ is:

$$\mathcal{L}_{assoc}^{(k)} = \frac{1}{|\tau_k|} \sum_{(h_a, h_b, h_c) \in \tau_k} \ell_{assoc}(x, y), \tag{12}$$

and the total associativity loss across all reduction steps is:

$$\mathcal{L}_{assoc} = \frac{1}{K} \sum_{k=1}^{K} \mathcal{L}_{assoc}^{(k)}. \tag{13}$$

This loss term is then added to the standard training loss with a weighting factor $\lambda_{assoc}$:

$$\mathcal{L} = \mathcal{L}_{task} + \lambda_{assoc} \mathcal{L}_{assoc}. \tag{14}$$

# H  NON-REGULAR LANGUAGE EXPERIMENTS

To maintain coherence, we restrict our extended study to sequence classifications to maintain consistency with our regular language experiments. We provide additional details about the ListOps and natural language tasks below.

## H.1  LISTOPS

We generated a dataset of ListOps sequences following the procedure outlined in Nangia & Bowman (2018). Each sequence is a list beginning with an operation (MAX, MIN, MED, SUM (modulo 10)) followed by either integer or nested list arguments. The sequences are generated with varying lengths, maximum depths, and maximum numbers of arguments per operation. We created training datasets of sizes 100k, 500k, and 1M sequences, containing sequences with lengths ranging from 5 to 40. The test data comprises of multiple sets where each set contains 10k sequences sampled ith different characteristics of length (bucketed), max depth and max number of arguments. This enables a comprehensive evaluation of length, depth and argument generalization of the trained models.

We performed hyperparameter sweeps for each of the baseline models and the LDRU on the ListOps task (except for the BBT-GRC, where we used the default hyperparameters given in the codebase of Ray Chowdhury & Caragea (2023)). The hyperparameter sweeps included learning rate, dropout probability, and model hidden dimension, and in the case of the LDRU: the weight of the associativity regularization. We used a single seed (1) for this sweep and we used the Adam (Kingma & Ba, 2015) optimizer with a batch size of 128. We continued to use linear warmup for 20% of the

training steps initially set to 1e-8 but no $L_2$ regularization was used. We trained each configuration for 200k steps on the 500k dataset and evaluated performance on a validation set of 2048 sequences sampled from the same distribution. We selected the best hyperparameters based on the highest average accuracy on the validation data. The final hyperparameters used for each model are presented in Table 19.

Table 18: Hyperparameter sweep ranges for LSTM, Transformer, and LDRU for ListOps.

| Model | Embedding / Hidden Dim | Dropout | Learning Rate | Associativity Regularization |
|---|---|---|---|---|
| LSTM | {256, 512, **1024**} | NA | {1e-5, 5e-5, **1e-4**} | NA |
| Transformer | {64, **128**} | {**0**, 0.1} | {1e-5, **5e-5**, 1e-4} | NA |
| LDRU | {64, 128, **256**} | {0, 0.1, 0.2} | {1e-5, 5e-5, 1e-4} | {0, 0.1, **1.0**} |
| LDRU (additional) | {**256**, 512} | {0.025, **0.05**, 0.1} | {1e-4, **2.5e-4**, 5e-4} | {**1.0**} |

Table 19: Selected model hyperparameters for all baseline models, BBT–GRC, and LDRU for ListOps.

| Component | Parameter | LSTM | Transformer | LDRU | BBT-GRC |
|---|---|---|---|---|---|
| Embedding | Embedding dim | None | 128 | 256 | 128 |
| | Vocab size | 19 | 19 | 19 | 19 |
| | Initialization | – | $\mathcal{N}(0, 0.02)$ | $\mathcal{N}(0, 0.02)$ | – |
| Core Architecture | Layers/Blocks | 1 | 5 | 1 | 1 |
| | Hidden dim | 1024 | 128 | 256 | 128 |
| Residual Connections | Dimension | – | 512 | 1024 | – |
| Normalization | Layer norm | No | Pre-norm & Post-norm | Post-norm | Post-norm |
| LDRU Operator | MLP hidden dims | – | – | $512 \rightarrow 1024 \rightarrow 256$ | – |
| | Activation | – | – | SiLU | – |
| | MLP initialization | – | – | Glorot | – |
| | Projection initialization | – | – | Identity | – |
| Transformer | Attention heads | – | 8 | – | – |
| | Head dimension | – | 8 | – | – |
| | Positional encoding | – | ALiBi/NoPE/Sinusoidal | – | – |
| Output | Classifier type | Linear | Linear | Linear | 2-layer MLP |
| | Output classes | 10 | 10 | 10 | 10 |
| Regularization | Dropout locations | – | Attention & Residual | Post-norm | In & Out |
| | Associativity Regularization | – | – | Yes | – |
| **Total Parameters** | **Approx.** | **4.2M** | **1.0M** | **2.6M** | **430k** |

**Results** The heatmaps in Fig. 12 report accuracy and standard deviation for length buckets with four combinations of max depth and max number of arguments under three training-set sizes. BBT-GRC largely attains the highest accuracy across buckets and all dataset sizes. Increasing the training-set size systematically improves LDRU accuracy: the 500k and 1M models show uniform gains relative to the 100k model across both in-distribution and out-of-distribution buckets. Under the same change in training-set size, the Transformer with ALiBi exhibits smaller improvements. The LDRU surpasses the standard baselines (LSTM, Transformer) at 500k and 1M. A consistent pattern across all sweeps was that LDRU configurations with non-zero associativity regularization achieved higher validation accuracy than those without it, indicating that encouraging approximate associativity is empirically beneficial and functions as a useful inductive bias rather than a redundant constraint.

## H.2 NATURAL LANGUAGE TASKS

We evaluated the LDRU and Transformer baselines on a set of standard sequence-classification datasets. We report results on all GLUE classification tasks except RTE, AX, and WNLI, which we exclude due to their small size, high variance, and limited incremental diagnostic value relative to the larger, more stable benchmarks. We also report results on AG's News and DBPedia, text classification tasks outside of GLUE. We report performance on the validation data for the GLUE tasks (using the recommended metrics for each individual task) and accuracy on test data for the additional datasets. We use the BERT base (uncased) (Devlin et al., 2018) tokenizer for all tasks. For paired inputs u, v, the concatenation format was the conventional [CLS],u,[SEP],v,[SEP].

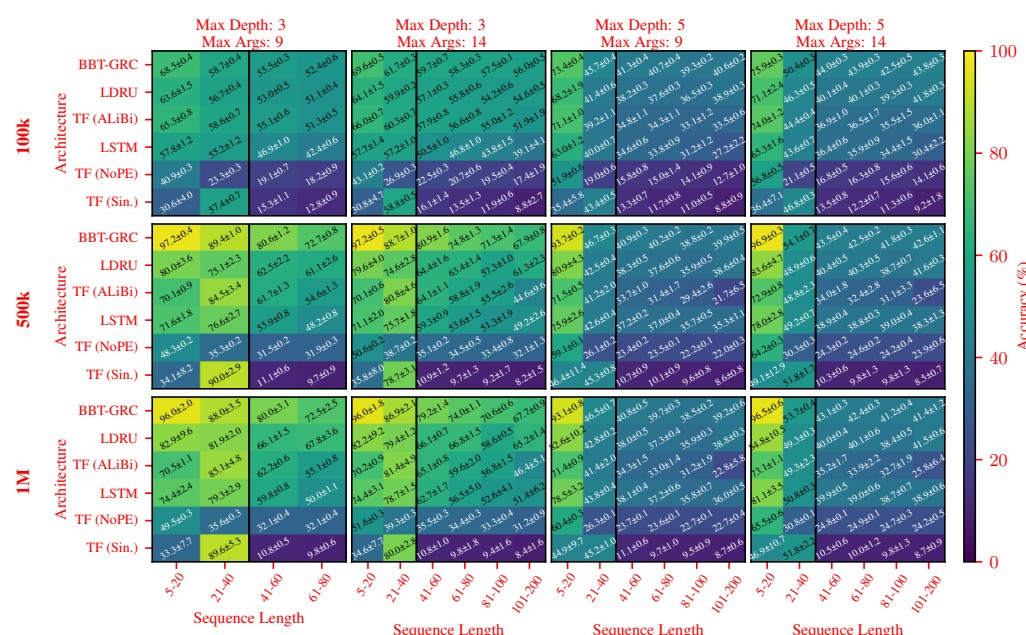

Figure 12: ListOps performance heatmaps for models trained with different dataset sizes (100k, 500k, and 1M sequences). Each cell presents the accuracy and its standard deviation of 10k generated sequences for that particular combination of sequence length, maximum depth, and maximum number of arguments. We bucket the sequence lengths for more efficient sampling. The black lines partitions the cells that are within the training distribution and the OOD cells that require generalization.

We present the truncation lengths, learning rates, batch sizes, and training steps for each task in Table 20.

Table 20: Sequence-length truncation thresholds, learning rates, batch sizes, and training steps for all evaluated NLP datasets.

| Dataset | Truncation Length | Learning Rate | Batch Size | Steps |
|---------|-------------------|---------------|------------|-------|
| CoLA | 64 | 1e-5 | 128 | 4000 |
| SST-2 | 64 | 1e-4 | 128 | 8000 |
| MRPC | 128 | 1e-5 | 32 | 4000 |
| QQP | 256 | 1e-4 | 32 | 50000 |
| MNLI | 384 | 1e-4 | 32 | 35000 |
| QNLI | 384 | 1e-4 | 32 | 20000 |
| AG's News | 512 | 1e-4 | 32 | 20000 |
| DBPedia | 256 | 1e-4 | 32 | 30000 |

We performed a hyperparameter sweep to find the best settings for the Transformer baselines (using ALiBi) and for the LDRU. We used the AG's News task to conduct the sweep, varying learning rate, dropout probability, and embedding/hidden dimension. For the LDRU we additionally swept the associativity-regularization weight. We used linear warmup over the first 20% of updates (initial learning rate $10^{-8}$), and no $L_2$ regularization. We give the sweep parameters and embolden the selected parameters in Table 21. Additional model details for the LDRU and Transformer are listed in Table 22.

**Results** The full results are shown in Table 23. Performance differences between the LDRU and the Transformer baselines are largely moderate across all GLUE tasks, with each architecture exhibiting strengths on different subsets of benchmarks. On QQP and MNLI (matched and mismatched), the

Table 21: Hyperparameter sweep ranges for all models on natural language tasks. We used ALiBi as the PE for the hyperparameter sweep.

| Model | Embedding Dim | Dropout | Learning Rate | Weight Decay | Assoc. Reg. |
|---|---|---|---|---|---|
| Transformer | {64, **128**} | {0, **0.1**} | {**1e-5**, 5e-5, 1e-4} | {0.0, 1e-4, **1e-5**} | NA |
| Transformer (Additional) | {256} | {**0.1**} | {**1e-5**} | {**1e-5**} | NA |
| LDRU | {64, 128, **256**} | {0, **0.01**, 0.025, 0.05} | {**1e-5**, 5e-5, 1e-4} | {0.0, 1e-4, **1e-5**} | {0, **0.1**, 1.0} |
| LDRU (Additional) | {256} | {**0.01**} | {**1e-5**, 5e-5} | {0.0, 1e-4, **1e-5**} | {0.5} |

Table 22: Model hyperparameters for all natural language tasks.

| Component | Parameter | Transformer | LDRU |
|---|---|---|---|
| Embedding | Embedding dim | 128 | 256 |
| | Vocab size | 30522 | 30522 |
| | Initialization | $\mathcal{N}(0, 0.02)$ | $\mathcal{N}(0, 0.02)$ |
| Core | Layers/Blocks | 6 | 1 |
| | Hidden dim | 128 | 256 |
| Attention | Heads | 8 | – |
| | Head dim | 16 | – |
| | Positional encoding | ALiBi/NoPE/Sinusoidal | – |
| LDRU Operator | MLP hidden dims | – | $256 \rightarrow 512 \rightarrow 256$ |
| | Activation | – | SiLU |
| | Assoc. Reg. | – | Yes |
| Output | Classifier | Linear | Linear |
| | Classes | task-dependent | task-dependent |
| Regularization | Dropout | 0.1 | 0.01 |
| **Total Parameters** | **Approx.** | **1.0M** | **2.6M** |

LDRU achieves the highest average scores across seeds. On CoLA, SST-2, and MRPC, the Transformer (Sinusoidal) obtains the strongest results. Transformer (NoPE) obtains the strongest results on QNLI. On the non-GLUE classification tasks, AG's News and DBPedia, the LDRU obtains the lowest accuracy on AG's News, but is similar to Transformer performance in absolute terms. The LDRU also achieves the highest accuracy on DBPedia (98.6%), exceeding all Transformer variants. Variances across seeds are uniformly small for both architectures.

Across the sweep on AG's News, LDRU configurations with non-zero associativity regularization consistently outperformed those with zero weight, and the selected configuration for every dataset used $\lambda_{assoc} = 0.1$. This result indicates that encouraging approximate associativity is beneficial outside of regular-language settings as well.

Table 23: Performance across GLUE benchmarks, AG's News, and DBPedia. Metrics follow GLUE conventions: CoLA (Matthew's Correlation Coefficient), SST-2 / MNLI / QNLI (Accuracy), MRPC / QQP (F1/Accuracy). AG's News and DBPedia report classification accuracy on test data. We give the metrics and their standard deviations over 5 seeds. Subscript M indicates matched and MM mismatched accuracy for MNLI.

| Architecture | GLUE Benchmarks | | | | | | | Additional Tasks | |
|---|---|---|---|---|---|---|---|---|---|
| | CoLA | SST-2 | MRPC | QQP | $MNLI_M$ | $MNLI_{MM}$ | QNLI | AG's News | DBPedia |
| TF (ALiBi) | $0.096 \pm 0.031$ | $79.3 \pm 1.0$ | $56.2 \pm 1.8 / 62.6 \pm 2.9$ | $74.0 \pm 1.9 / 78.3 \pm 0.9$ | $53.9 \pm 0.7$ | $52.8 \pm 0.7$ | $58.1 \pm 0.2$ | $\mathbf{89.8 \pm 0.4}$ | $98.2 \pm 0.1$ |
| TF (Sin.) | $\mathbf{0.124 \pm 0.027}$ | $\mathbf{81.1 \pm 0.9}$ | $\mathbf{59.1 \pm 1.4 / 67.5 \pm 3.6}$ | $70.7 \pm 0.7 / 75.8 \pm 0.2$ | $49.0 \pm 0.3$ | $50.5 \pm 0.5$ | $56.9 \pm 3.3$ | $89.1 \pm 0.9$ | $97.8 \pm 0.1$ |
| TF (NoPE) | $0.097 \pm 0.028$ | $79.1 \pm 1.2$ | $59.0 \pm 1.6 / 64.0 \pm 2.4$ | $72.8 \pm 1.6 / 77.8 \pm 0.8$ | $50.5 \pm 0.4$ | $51.1 \pm 0.6$ | $\mathbf{58.8 \pm 0.9}$ | $89.6 \pm 0.3$ | $97.8 \pm 0.1$ |
| LDRU | $0.085 \pm 0.036$ | $80.9 \pm 1.0$ | $57.0 \pm 3.6 / 63.2 \pm 5.7$ | $\mathbf{76.9 \pm 0.3 / 79.1 \pm 0.3}$ | $\mathbf{58.3 \pm 0.7}$ | $\mathbf{57.9 \pm 0.7}$ | $56.1 \pm 1.2$ | $89.0 \pm 0.5$ | $\mathbf{98.6 \pm 0.0}$ |

# I    ANALYSIS OF MONOID COMPOSITIONS

This section describes how we computed the equivalence class composition patterns shown in Fig. 6 from the main text and further explains why training sequence length directly impacts generalization performance on $D_n$ languages. Fig. 6 presents the composition patterns for $D_6$ over varying training

and test lengths. To illustrate the explicit structure of monoids for the $D_n$ languages, we present the complete monoid table for $D_2$ in Table 24, which recognizes the language $(0(01)^*1)^*$. We present $D_2$ because it has 15 equivalence classes, and $D_6$ is significantly more complex with 141 classes. In general, the monoid of $D_n$ has $1 + \frac{(n+1)(n+2)(2n+3)}{6}$ equivalence classes. The complete $D_2$ automaton has three states: state 0 (initial/accepting), state 1 (after reading '0'), state 2 (the bottom of the fixed stack), and state 3 (rejecting sink state).

However, we focus on the equivalence classes that can be components of positive examples to reduce the complexity of modeling the LDRU operator $\odot_\theta$. This means we are only interested in equivalence classes that contain even-length sequences. Despite only considering a subset of equivalence classes (73 for $D_6$), there can still be many classes, so we focus on the $D_6$ language instead of $D_8$ or $D_{12}$ for tractability and simpler visualizations.

Table 24: Monoid elements for $D_2$. Each equivalence class is characterized by its state mapping function and representative sequences. All other sequences in $\Sigma^*$ can be characterized as one of these equivalence classes. A representative sequence is shown in the third column, and the description of each class is provided in the fourth column. The symbol $\epsilon$ denotes the empty sequence.

| Element | State Mapping | Representative | Description |
|---|---|---|---|
| $e_0$ | $\{0 \mapsto 0, 1 \mapsto 1, 2 \mapsto 2\}$ | $\epsilon$ | Identity |
| $e_1$ | $\{0 \mapsto 1, 1 \mapsto 0, 2 \mapsto 2\}$ | 0 | Stack push |
| $e_2$ | $\{0 \mapsto 3, 1 \mapsto 0, 2 \mapsto 1\}$ | 1 | Stack pop |
| $e_3$ | $\{0 \mapsto 2, 1 \mapsto 3, 2 \mapsto 3\}$ | 00 | Double stack push |
| $e_4$ | $\{0 \mapsto 0, 1 \mapsto 1, 2 \mapsto 3\}$ | 01 | Partial identity |
| $e_5$ | $\{0 \mapsto 3, 1 \mapsto 1, 2 \mapsto 2\}$ | 10 | Partial identity |
| $e_6$ | $\{0 \mapsto 3, 1 \mapsto 3, 2 \mapsto 0\}$ | 11 | Double stack pop |
| $e_7$ | $\{0 \mapsto 3, 1 \mapsto 3, 2 \mapsto 3\}$ | 000 | Annihilation |
| $e_8$ | $\{0 \mapsto 1, 1 \mapsto 3, 2 \mapsto 3\}$ | 001 | Stack push |
| $e_9$ | $\{0 \mapsto 3, 1 \mapsto 0, 2 \mapsto 3\}$ | 011 | Stack pop |
| $e_{10}$ | $\{0 \mapsto 3, 1 \mapsto 2, 2 \mapsto 3\}$ | 100 | Stack push |
| $e_{11}$ | $\{0 \mapsto 3, 1 \mapsto 3, 2 \mapsto 1\}$ | 110 | Stack pop |
| $e_{12}$ | $\{0 \mapsto 0, 1 \mapsto 3, 2 \mapsto 3\}$ | 0011 | Partial identity |
| $e_{13}$ | $\{0 \mapsto 3, 1 \mapsto 1, 2 \mapsto 3\}$ | 0110 | Partial identity |
| $e_{14}$ | $\{0 \mapsto 3, 1 \mapsto 3, 2 \mapsto 2\}$ | 1100 | Partial identity |

**Monoid Computation** For $D_6$, we compute the monoid by first constructing the DFA recognizing the language (Fig. 13) and then extracting equivalence classes by generating all sequences up to a fixed length and grouping them by their induced state mappings in the DFA. To ensure that we cover all relevant equivalence classes, we generated all even-length sequences up to length 12 (i.e., $2n$ length is sufficient for $D_n$). This length is sufficient to capture all possible state mappings in the DFA due to its structure, including its sink state that absorbs malformed sequences. The equivalence classes that we do not discover are those that only contain odd-length sequences, meaning that we can characterize any even-length sequence by dividing it into smaller even-length sequences.

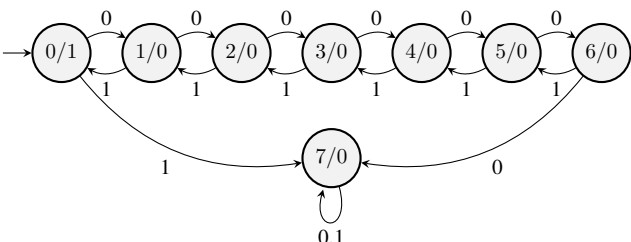

Figure 13: A complete DFA that recognizes $D_6$. We include the rejecting sink state (a state such that if a sequence reaches it, then it is rejected as it is malformed), unlike the $D_3$ figure in Table 7, to better illustrate the state mappings of the monoid.

**LDRU Processing Simulation** We simulate LDRU processing to estimate the frequency of monoid element compositions that occur during training and testing:

1. We determine the number of sequences to generate based on fixing the number of equivalence class compositions we want to observe. We fix the number of compositions to 1,000,000 within a range of sequence lengths and vary the number of samples per length within the range to ensure an equal number of equivalence class compositions per length. The ranges of sequence lengths we examined were 10–40 (step 2), 42–60 (step 2), 62–100 (step 4), 102–150 (step 4), and 480–500 (step 2).

2. We determine the equivalence classes of even-length subsequences that reflect increasing depth in the LDRU: the first are the 2-length, then 4-length, then 8-length subsequences, and so on until the subsequence is the entire sequence. Each of these subsequences is processed by the LDRU as the token embeddings are aggregated during $\odot_\theta$ steps.

3. During reduction simulation, we count all pairwise compositions $(e_i, e_j) \rightarrow e_k$ that occur when applying the monoid operator $\odot$. We do not count the compositions that result in the equivalence classes representing the 2-length subsequences because they are only 4 types of compositions (binary alphabet) and would dominate the composition counts.

4. We record composition frequencies to generate probability distributions over monoid element pairs.

The simulation replicates the exact tree structure of LDRU processing, ensuring that recorded compositions match those encountered during actual model training. The heatmaps in Fig. 6 reveal insights into why training sequence length influences generalization success:

**Analysis** Positive sequences of length 10–40 exhibit sparse composition patterns, with most equivalence class pairs showing low log probabilities (lighter regions). The concentration of high-probability compositions in the lower-index region (equivalence classes 0–15) occurs because these represent the most frequently encountered partial sequences in shorter training data. As training sequence length increases to 62–100 and beyond, additional composition patterns emerge. As equivalence classes are discovered dynamically from the generated sequences with increasing length, higher index equivalence classes tend to be rarer and sharper state mappings (i.e., an increasing number of annihilations, see Table 24).

Complete generalization requires some exposure to all possible monoid element compositions that can occur during testing. The difference in sparsity between training and testing heatmaps directly explains the empirical results: insufficient training sequence length fails to provide sufficient coverage of rare but necessary monoid compositions. Moreover, some languages are likely to be impossible to learn (i.e., generalize to) without a sufficient maximum training sequence length, as they require compositions between equivalence classes that are never encountered in shorter sequences.

