# OpenReview forum: "Length Generalization with Log-Depth Recurrent Units"
_ICLR.cc/2026/Conference — Submitted to ICLR 2026_

### Official Review · Reviewer_mVcQ · 2025-10-23

**Soundness:** 3
**Presentation:** 3
**Contribution:** 2
**Rating:** 4
**Confidence:** 4

**Summary:**

This paper introduces the Log-Depth Recurrent Unit (LDRU), a novel recurrent architecture that composes token embeddings via a learned pairwise reduction operator inspired by monoid composition, achieving logarithmic computational depth. The authors evaluate LDRU on 21 regular language tasks, including newly proposed prefix languages to test long-range dependencies. The model demonstrates near-perfect length generalization, achieving 100% OOD accuracy on most tasks and outperforming strong baselines such as RNNs, LSTMs, Transformers, RegularGPT, and state-space models. The work is well-motivated, clearly written, and provides strong theoretical grounding and empirical evidence. While its experiments are confined to synthetic regular languages and the practical efficiency of log-depth computation is not yet verified, the contribution represents a meaningful advance in systematic generalization and architecture design for sequence models.

**Strengths:**

1. The proposed LDRU is a novel architecture according to the Reviewer's expertise.
- LDRU’s log-depth reduction mechanism is a clever hybrid of RNN recurrence and Transformer parallelization.
- The design explicitly encodes compositional inductive bias linked to formal language theory.

2. This paper provide strong empirical results, demonstrating the effectiveness of LDRU on systematic length generalization.
- Comprehensive evaluation on 21 tasks, including new benchmarks.
- Consistent and large performance gap over state-of-the-art baselines.

3. Others: the authors introduce a benchmark (Prefix Languages) which provides a systematic way to test long-range dependency modeling. The paper is well-structured, with a clear logical flow from theory to method to experiment to analysis.

**Weaknesses:**

1. Restricted Scope: The evaluation domain is narrow; all tasks are regular or near-regular, where compositional structure is simple. It’s unclear how LDRU would generalize to non-regular or natural language tasks where equivalence classes are not well-defined. Besides, it is doubtable how LDRU is compatiable with existing foundation language models.

2. While $O(\log n)$ depth is theoretically appealing, practical runtime or memory benchmarks (on GPUs) are not provided and the reduction tree might introduce non-trivial communication or synchronization overhead. Besides, no formal proof is given for why or when LDRU will learn correct compositions (though inspired by monoid theory).

**Questions:**

1. Please refer to the weaknesses part.
2. The authors neatly compare LDRU with standard Transformers and RNNs on working complexity and depth. Can you provide some empirical results to show the superiority (or the tradeoff) of LDRU, in comparison with Transformers and RNNs?
3. I think LDRU's performance does not show much improvment over LSTM (both near to $100\%$ accuracy). Though LDRU is designed to be more effective on the processing depth ($O(\log n)$ versus $O(n)$), the reviewer wonders whether LDRU beats LSTM by a large margin in some length generalization tasks?

---

> ### Author Response · Authors · 2025-11-17
> **Improved scope and practical architecture comparison**
>
> Dear Reviewer mVcQ,
>
> Our revised manuscript addresses your **W1** and **W2** by (i) adding ListOps and eight NLP classification tasks (Sec. 6, Table 4) and (ii) providing GPU runtime, FLOPs, throughput, and memory benchmarks versus RNNs and Transformers (Sec. 3.1, Fig. 3), and the general response summarizes these changes; these additions also respond directly to Q2.
>
> **W2** - Formal proof
>
> We agree that formal guarantees for when LDRU recovers the correct monoid are important; in this work we restrict ourselves to empirical evidence (including the $D_n$ and associativity-regularization analyses) and explicitly leave such proofs to future work.
>
> **Q3** - Table 2 shows that the LDRU matches or exceeds LSTM OOD accuracy on all 21 regular tasks.
>
> The differences are modest on simpler regular languages but become clear on Cycle Navigation ($60\% \pm 2\%$ for LSTM vs. $100.0\% \pm 0.0\%$ for LDRU), the deeper $D_n$ tasks where LSTM accuracy drops while LDRU remains high, and the hardest prefix language $P_{4,4}$ ($97.3\% \pm 2.2\%$ vs. $100.0\% \pm 0.0\%$). The common pattern, especially in Cycle Navigation and the deeper $D_n$ tasks, is that almost every input symbol forces a state change that must be tracked over long sequences, with $P_{4,4}$ adding a particularly demanding long-term memory requirement. In these harder cases, the LDRU retains high performance on tasks that the LSTM comparatively finds difficult, while using only logarithmic computational depth
>
> We’re happy to provide further clarification as needed. Thank you for reading the manuscript and for your thoughtful review. If our response has resolved your concern, we would be grateful if you could consider adjusting your score accordingly.
>
> The Authors

---

> > ### Comment · Reviewer_mVcQ · 2025-11-27
> > **Thanks for the response**
> >
> > Dear Authors,
> >
> > Thanks for the response. I appreciate the addition results (especially for those in Table 4). I would like to encourage the authors to make this part more comprehensive (e.g., including generation tasks, highlighting LDRU's advantage on length generalization). Overall, I feel the work is a borderline work for ICLR (just from my point of view), and hence I decide to maintain my score.
> >
> > Reviewer mVcQ

---

### Official Review · Reviewer_hrDN · 2025-10-25

**Soundness:** 2
**Presentation:** 3
**Contribution:** 3
**Rating:** 2
**Confidence:** 4

**Summary:**

This paper presents the log-depth recurrent unit (LDRU) architecture, which reduces information along the sequence dimension via a sequence of aggregations of pairs that requires log (in sequence length) depth to aggregate information from the entire sequence. This is compared to transformers and RNNs on length generalization as measured by synthetic regular language tasks.

**Strengths:**

1. The paper presents an interesting idea that combines some of the relative strengths of transformers and RNNs.
2. The length generalization results with the new architecture seem consistently strong.
3. The results interpreting the patterns of composition that are needed in the data for length generalization to arise are interesting.

**Weaknesses:**

1. This paper neglects important related work (e.g. delta nets https://arxiv.org/abs/2406.06484 , PSMs: https://arxiv.org/abs/2506.10918 and especially log-linear attention: https://arxiv.org/abs/2506.04761). These papers present very similar methods with more general and scalable experimental results, although less of a focus on length generalization. The authors would need to carefully read and discuss this work and how it relates to this paper. Especially log-linear attention and PSMs present the idea of generic parallel scans for doing log-depth recurrence.
2. The paper is missing comparisons of parameters + FLOPs across architectures. Adding the MLP layers will add parameters and make the networks larger than the baselines. This seems like it potentially makes the comparisons unfair as a result. Moreover in terms of actually understanding the cost of the new architecture, we need a more detailed analysis of the FLOPs required both for training and for inference.
3. It seems that the architecture likely does not allow easy parallel training. We have seen in recent years that this is important for scalability (e.g. transformers + SSMs). It is not even obvious to me from the paper whether computation is being re-used during training, or if for each token during training the entire LRU over the sequence is being recomputed. Full description of the training complexity for next token prediction would be useful.
4. The transformer baseline seems somewhat weak by only using NoPE positional embeddings. It is now fairly standard to compose sliding window RoPE layers with global NoPE layers for length generalization. The local layers are needed to create better representations of local tokens that is difficult with NoPE. Here is one example paper with dramatically better length generalization with better positional encodings: https://arxiv.org/abs/2402.01032.
5. There is no strong state space model baseline. In particular, the gated delta net is the SoTA architecture and is substantially more expressive than Mamba and similar architectures. This baseline is needed.
6. The results of LDRU without the non-linearity seem fairly similar to with the non-linearity. When it is linear, it seems that the LDRU may also become equivalent to some form of state-space/linear attention.

**Questions:**

1. What is the performance of each model on the training distribution? It seems fair if we only care about testing generalization to train until there is 100% accuracy on short sequences before testing on longer ones.

---

> ### Author Response · Authors · 2025-11-17
>
> Dear Reviewer hrDN,
>
> We hope that our general response and the revised manuscript have adequately addressed your broader concerns. Below we respond to your specific weaknesses and questions.
>
> **W1** - Missing related work
>
> We have expanded the related-work discussion under “Reduction Applications” to cover DeltaNet, PSMs, and Log-Linear attention. These architectures use parallel scans primarily to accelerate fixed linear state updates or linearized attention on large-scale tasks, whereas LDRU learns an approximately associative operator for generic regular-language transductions. We now position LDRU as complementary to this line of work: it focuses on reliable length generalization on formal languages via a stable, learned binary operator rather than optimizing a specific linear recurrence or attention mechanism for scalability.
>
> **W2** - Parameter counts and FLOPs
>
> The appendix reports parameter counts for all models to two significant figures (Table 12). The LDRU uses 160k parameters, smaller than all baselines except the vanilla RNN (67k). All baselines were re-run under a unified optimizer and training setup. As noted in the general comment, we now include practical runtimes and throughput for training and analytical FLOP plots for forward passes of the LDRU, RNN, and Transformer (Fig. 3).
>
> **W3** - Parallelization and training structure
>
> All experiments in this paper are sequence-level classification, so we compute one full reduction per sequence during both training and evaluation. Section 3.1 and Appendix D now make the complexity explicit: the model performs $O(nd^2)$ work with $\log_2 n$ depth, with all pairwise applications of $\odot_\theta$ at each level executed in parallel. For autoregressive next-token prediction, training would permit $\log_2 n$ nodes to be used for next token prediction immediately, but this is a matter we leave to future work. The same reduction tree structure would allow substantial compute savings at inference time.
>
> **W4** – Transformer baseline strength
>
> We agree that alternative positional encodings provide useful contrast and have now included these variants in the revised manuscript (Table 2). We have run ALiBi (part of the study in the paper you cited) and randomized RoPE as it was specifically designed to target better length generalization in regular languages.
>
> **W5** – State-space model baselines
>
> We have already evaluated S5 and SD-SSM baselines; in our experiments, LDRU outperforms these models on the regular-language benchmarks (see Appendix F, Table 14). We are currently implementing the Gated DeltaNet architecture you highlight; due to implementation differences we will only be able to include the results will appear in a subsequent revision *during the discussion period* rather than in the present one. We do not expect it to alter the main conclusion that a log-depth, approximately associative operator is a strong inductive bias for regular-language length generalization. We will update the revised manuscript as soon as we have the results.
>
> **W6** – Effect of removing nonlinearities
>
> RQ3 (Table 3) clarifies this point. On Modular Arithmetic, the purely linear operator variant plateaus at 60.8% OOD accuracy, whereas the gated MLP operator reaches 100%, while both behave similarly on easier tasks such as Parity Check and Cycle Navigation. This shows that LDRU is strictly more expressive than its linear scan limit (which is closer to SSM or linear-attention behaviour) and that the nonlinearity in $\odot_\theta$​ is required to capture the harder regular languages.
>
> **Q1** – Training-distribution performance
>
> For the regular-language tasks, we train all models until their accuracy on the training-length regime (lengths 1–40) has effectively saturated and, in many cases, reached 100%. Despite this, models with almost identical training performance can exhibit very different behaviour on longer sequences: some architectures fail sharply out of distribution, while LDRU maintains high accuracy. This shows that perfect training accuracy is neither sufficient (it does not guarantee length generalization) nor strictly necessary (useful extrapolation can emerge before every last training error is eliminated). We view length generalization as a property of how models behave beyond the training regime, not as something that should be conditioned on achieving exact 100% training accuracy. However, in the next revision to our manuscript, we will add an appendix table summarizing training-length accuracies for all models to make this relationship explicit.
>
> We’re happy to provide further clarification as needed. Thank you for reading the manuscript and for your thoughtful review. If our response has resolved your concerns, we would be grateful if you could consider adjusting your score accordingly.
>
> The Authors
>
> [1] Gated Delta Networks: Improving Mamba2 with Delta Rule

---

### Official Review · Reviewer_E7NM · 2025-10-27

**Soundness:** 2
**Presentation:** 3
**Contribution:** 1
**Rating:** 2
**Confidence:** 4

**Summary:**

The paper proposes Log-Depth Recurrent Units (LDRU) and shows that it can generalize well in multiple regular language tasks. LDRU is effectively implemented as a balanced binary tree recursive neural network with a new gated cell as the parametrized binary operator.

**Strengths:**

* It demonstrates that binary balanced tree-based recursive neural networks with a modern gated cell can perform nearly perfect in multiple parity tasks.

**Weaknesses:**

* The contribution seems incremental compared to RegularGPT. RegularGPT seems to be essentially the same idea (a balanced binary tree recursion - except uses Transformer as the recursive cell) - and also performs near perfectly (better in the original paper). The improvement in this paper seems like due to minor implementational difference rather than exposing any theoretical leap.
* Limited evaluation on anything else besides algorithmic regular language tasks. Does not show if it can be anything more than something that works well in "toy" tasks. While I do appreciate evaluation on algorithmic/synthetic tasks - particularly because they can be harder to "hack" by finding spurious shortcuts and easier to analyze - however, restricting the whole study to them when introducing a supposedly general purpose model restricts the scope severely.
* Seems to miss a lot of critical related paper:
   - First of all, LDRU seems like a neologism for Balanced Tree Recursive Neural Networks - which have been explored in multiple prior works [1,2,3].
   - Moreover it misses any comparison with more recent proposals like Recursion-in-Recursion (RIR) which also proposes logarithmic-depth recurrence/recursion and shows effectiveness in algorithmic tasks - like propositional logical inference, listops (in multiple OOD settings) - alongside benchmarks in LRA and other [4] and utilizes a modern gated recurrent cell inspired from Ordered Memory. At the very least, I would be curious how the proposed method compares against RIR in ListOps and Logical Inference on the same length generalization and argument generalization settings in RIR. But to be frank - even if the benchmarks are provided I would likely not lean towards acceptance unless the other weaknesses are rebutted well enough.


[1] Neural Tree Indexers for Text Understanding - Munkdhalai et al.

[2] Sliced Recurrent Neural Networks - Yu et al.

[3] On Tree-Based Neural Sentence Modeling - Shi et al.

[4] Recursion in Recursion: Two-Level Nested Recursion for Length Generalization with Scalability - Ray Chowdhury et al.

**Questions:**

n/a

---

> ### Author Response · Authors · 2025-11-17
> **Why the LDRU is distinct**
>
> Dear Reviewer E7NM,
>
> We hope that our revised manuscript and general comment to all reviewers has adequately addressed **W2** by adding ListOps and several natural language classification benchmarks beyond regular languages. We now address the other weaknesses from your review.
>
> **W1** - RegularGPT
>
> RegularGPT uses weight sharing over a multi-token attention + FFN block applied to full-length sequences at each merge step. This method does not define a binary operator \mathbb{R}^d x \mathbb{R}^d \rightarrow \mathbb{R}^d, does not approximate an associative operator, and does not impose depth-invariant transformations. LDRU instead learns a single binary operator reused at every internal node of a fixed reduction. That constraint is what enables a monoid-style interpretation and yields a stable composition rule rather than depth-dependent transformations. The architectures differ in the computational object they learn, not in superficial topology.
>
> **W3** - Balanced Tree RvNNs & RIR
>
> Prior tree-structured RvNNs share only the tree topology. Their cell is an unconstrained nonlinear merge function at each internal node, without requirements for consistency across compositions (associativity). They aggregate features, but they do not necessarily learn a compositional law. The LDRU imposes structural constraints these models do not: a fixed parallel reduction, residual + normalization reused at each step, and an operator biased toward approximate associativity. These constraints are what enable the monoid analysis and systematic generalization. RIR differs further: it trades off the speed of BBT-RvNNs with increased expressivity by using an RvNN as the cell to process $k$-ary blocks in parallel. Using the $k$-ary block enables the use of RvNNs that heuristically determine tree structure of a sequence.
>
>
> Following your feedback, we have added the BBT-GRC baseline to the ListOps experiments (Fig. 4) in the version uploaded with this response. We chose this model as it is the most similar to the LDRU (both use a balanced binary tree). We have also run RIR-GRC on the same ListOps setting, and will update Fig. 4 in a subsequent revision (during the discussion phase) once we have collated these results to provide the comparison you requested.
>
> We’re happy to provide further clarification as needed. Thank you for reading the manuscript and for your thoughtful review. We hope these clarifications and additions address your concerns and are reflected in your assessment of the paper.
>
> The Authors

---

> ### Comment · Reviewer_E7NM · 2025-11-18
>
> I commend the authors for their extensive experimentations and rebuttal. I have increased my score to 4 but still not completely convinced.
>
> **1. Broader issue**
> Even if the authors address all the feedback, I think these would be too many changes to fairly review in just the discussion/rebuttal stage. Overall a second round of review may be better.
>
> **2. Presentational Suggestion**
>
> I think the paper should be much more clear about the lineage of Recursive Neural Networks that already exists - where they fall short (if at all), and what difference the new proposal made.
>
> For example the introduction says:
>
> "Prior work (Deletang et al., 2023; ´
> Chi et al., 2023; Butoi et al., 2025) shows that networks with recurrence generalize more reliably to
> regular languages than Transformers (Vaswani et al., 2017), which operate at a fixed depth regardless of sequence length. However, recurrent neural networks (RNNs; Elman, 1990) still suffer from
> positional bias: early tokens must traverse longer computational paths than later ones, leading to
> long-range memory problems (Bengio et al., 1994). It is therefore desirable to design an architecture that demonstrates reliable generalization to regular tasks as a first step toward architectures that
> reliably generalize to more complex language tasks.
>
> **To address this gap, we propose the Log-Depth Recurrent Unit (LDRU), an architecture that composes token embeddings through a learned pairwise operator applied in a balanced reduction tree.**
> This design avoids the positional bias of RNNs by giving every token the same computational depth,
> while still allowing information to propagate across the entire sequence."
>
> The introduction does not acknowledge the balanced tree recursive models that have already existed, and seems to frame Log-depth recurrence as a novel proposal (even if it is not explicitly mentioned as novel, it sounds like that's the claim).
>
> Instead if the main new contribution is the bias for associativity that can be the primary emphasis - e.g. what are the theoretical motivations for it, why earlier models are falling short, and empirical support of new approach.
>
>  **3. Theoretical Question**
>
> If I understand correctly, you are trying to motivate the use of binary associative operator as a recursive cell. If so what is the theoretical motivation for going for Log-depth route instead of using parallel scan as done in S5. I know LDRU seems to empirically perform better than S5, but is there any theoretical motivation?
>
> **4. Motivational Quesion**
>
> I think the motivation gets a bit muddled - if theoretically something like S5 has all the right "tools" but empirically doesn't do as well, whereas empirically RegularGPT and perhaps BBT-GRC (?) (arguably they are more or less same. GRC - was originally inspired from Transformer's FFN in ordered memory, and RegularGPT uses n=2 for n-ary recursion for regular tasks making attention superflous - and turning it into similar to BBT-GRC in effect) seems to perform nearly as well in regular tasks. What exactly would then be the point of LRDU in between?
>
> Also just for your awareness, there seems to be newer SSM papers that perform near perfectly in the regular language task (Structured Sparse Transition Matrices to Enable State Tracking in State-Space Models - Aleksandar Terzic et al.--- need not be compared here as it is a near concurrent paper)
>
> **5. Technical Questions and Empirical Concerns**
>
> > a fixed parallel reduction, residual + normalization reused at each step, and an operator biased toward approximate associativity.
>
> I am not seeing a strong difference. I acknowledge that your initalization scheme and regularization are novel differences - but they are minor technical additions - and also not as clear to be necessary or desirable for general-purpose use. Otherwise RIR or BBT-GRC already uses parallel reduction, normalization at each step, and gated addition similar to the cell structure proposed in the paper.
>
> Moreover, we don't necessarily need difference for difference's sake if it neither keeps up in empirical performance with earlier Log-depth recursive networks nor in theory in terms of implementation of binary associative operations (if desired) - as in S5 and co.
>
> For example RIR, BBT-GRC acheives 70+% in MNLI (Table 7), shows better robustness in adversarial tests in natural language (e.g. in LenMM), RIR generalizes to 1000+ length in Listops (Table 2) after training on <=100 length with less than 100K data (and possibly requires less than that).  Also performs as well in Propositional Logical Inference (Table 4) (so it's not just a bias towards ListOps).

---

> > ### Author Response · Authors · 2025-11-19
> > **1/2**
> >
> > Dear Reviewer E7NM,
> >
> > Thank you very much for your thoughtful follow-up and for revisiting your assessment of the paper. We are grateful for the detailed conceptual feedback and for another opportunity to convince you further. Below we address each of your new points.
> >
> > **(1) Broader issue: scale of changes during discussion**
> >
> > We agree that our revision contains substantial additional material, but the main regular task results and the monoid-based formulation were already present in the original submission. The new experiments (stronger PE baselines, ListOps, GLUE) were added strictly in response to reviewer requests and do not change our core claim. The original regular-language experiments already support the central message: an associativity-biased BBT-RvNN yields systematic length generalization across 21 regular tasks. The additional results refine this picture and show that the same inductive bias remains competitive beyond regular languages, but they do not introduce new claims to the work.
> >
> > We will revise the work to better distinguish this core message as:
> >
> > - **Our primary contribution:** the bias toward associativity, the regular language connection to monoids, and the length generalization study on 21 regular tasks.
> > - **Further supporting evidence:** ListOps and NLP experiments that show how the same bias has potential beyond regular tasks.
> >
> > **(2) The LDRU's lineage**
> >
> > We agree that the paper should be more explicit in the introduction about the existing lineage of balanced-tree recursive networks and how LDRU fits within that line of work. In the camera-ready, we will revise the introduction to explicitly position LDRU as a BBT-RvNN with an associativity-biased operator, rather than as a wholly new architectural paradigm. As introduced in our previous revision, we have already added a detailed discussion of BBT-RvNNs and RIR, and how they contrast with LDRU, in the related work section.
> >
> > **(3) Theoretical question: log-depth reduction vs. parallel scan (S5-style)**
> >
> > From the monoid perspective on regular languages, associativity is the central algebraic property, so we design the LDRU operator to be explicitly biased toward approximate associativity. For this purpose, the log-depth reduction is an efficient way to apply this operator.
> >
> > > If so what is the theoretical motivation for going for Log-depth route instead of using parallel scan as done in S5.
> >
> > Algorithmically, there is no gap between the log-depth route and the parallel scan used in S5: both rely on an associative binary operator and both have $O(\log L)$ parallel depth. The difference between a reduction and a scan is only that a reduction produces only the final value of the recurrence, whereas a scan produces all prefixes (including the final value). The scan cleanly maps to sequence-to-sequence modeling (S5) while the reduction maps to sequence encoding/classification (LDRU).
> >
> > The substantive difference is in **how the operator itself is defined**. In S5 (and SSMs more broadly), the operator used in the scan is an analytically specified linear operator derived from a discretized state-space model. For example, Eq. 34 [1],
> >
> > $q_i \cdot q_j = (q_{j,a} \odot q_{i,a},\; q_{j,a} \otimes q_{i,b} + q_{j,b}),$
> >
> > gives the fixed form of the S5 operator that composes two chunks of the recurrence.
> >
> > In the LDRU, we instead learn this binary operator with an explicit associativity bias. The theoretical motivation is that regular languages can be effectively modeled using monoids, which require an associative binary operator. What is unknown is exactly how to model this function, so by learning the operator itself (rather than fixing it to a particular linear SSM form), the LDRU is designed to approximate this general class of monoid computations while still benefiting from the same log-depth parallel composition. This gives a more flexible but still strongly structured inductive bias than S5’s specific linear state-space operator.
> >
> > **(4) Motivation: relation to S5/SSMs and recursive models**
> >
> > Thank you for this clarification; we agree the motivation should be more sharply drawn. The LDRU occupies a specific gap: it combines the desirable properties of SSMs (i.e., associative composition, log-depth) with the expressivity of learned nonlinear operators found in BBT-RvNNs. The empirical results indicate that nonlinearity in the operator (while keeping a general associative bias) helps improve its expressivity and relaxing associativity to a bias instead of a constraint still provides good performance. The contribution is the operator design, connecting SSMs and BBT-RvNNs rather than an incremental change on both.
> >
> > **Concurrent SSM work.**
> >     Regarding [2], their setup on regular tasks is similar to ours but evaluates only up to length 256. Under that protocol, PD-SSM still does not achieve perfect OOD generalization and underperforms the LDRU on several tasks (Cycle Navigation, Even Pairs, Modular Arithmetic, Parity Check).
> >
> > [continued]

---

> > > ### Author Response · Authors · 2025-11-19
> > > **2/2**
> > >
> > > [continued]
> > >
> > >
> > > **(5) Technical and empirical differences vs. RIR/BBT-GRC**
> > >
> > > Structurally, the LDRU cell differs from the GRC cell [3] in two ways that are directly motivated by the monoid perspective. First, GRC includes an additional gated term on the concatenation $[h_i; h_j]$, which makes its behavior sensitive to bracketing and breaks any bias toward associativity. In contrast, the LDRU *only* has the gated element-wise sum, avoiding this potential breakage. Second, we treat the zero embedding as a neutral element: when we pad an odd-length sequence, the neutral placeholder is designed to pass through its partner embedding without transformation. This should be more clearly stated in the paper. This explicit neutral element is absent from the standard GRC formulation.
> > >
> > > Empirically, substituting the GRC cell into our reduction architecture confirms these differences matter. We have additionally tested the GRC as part of the **RQ3** ablation and it only attains 82.3% ± 15.3% on Modular Arithmetic, whereas the LDRU MLP operator achieves 100.0% ± 0.0% on Modular Arithmetic under the same reduction structure and training setup. This failure on a single, more demanding regular task is consistent with our hypothesis: without a strong associativity bias and neutral element handling, errors accumulate along the reduction, and systematic length generalization becomes fragile.
> > >
> > > | **Operator** | **Even Pairs** | **Modular Arithmetic** | **Parity Check** | **Cycle Navigation** |
> > > | ------------ | -------------- | ---------------------- | ---------------- | -------------------- |
> > > | GRC          | 100.0 ± 0.1    | 82.3 ± 15.3            | 100.0 ± 0.0      | 100.0 ± 0.0          |
> > > | MLP          | 100.0 ± 0.0    | 100.0 ± 0.0            | 100.0 ± 0.0      | 100.0 ± 0.0          |
> > >
> > > We acknowledge that RIR/BBT-GRC achieve higher MNLI scores (~70%) under the training setup of [4] (much higher training budget to ours). However, our NLP experiments serve a different purpose: to verify that the _same_ associativity-biased operator that enables perfect regular-language generalization doesn't catastrophically fail on natural language. Under matched compute (equal steps, standard optimizer), LDRU performs comparably to Transformer baselines, confirming the operator's broad applicability. Whether an LDRU-based model can be scaled and tuned to match RIR/BBT-GRC on large NLP benchmarks is an important follow-up, but is intentionally outside the scope of the present contribution.
> > >
> > > ---
> > >
> > > To follow up on your initial feedback, we have completed experiments with RIR-GRC on ListOps and now report the length–generalization behavior explicitly. Across configurations, RIR-GRC achieves very strong accuracy on short sequences but degrades sharply as length increases.
> > >
> > > - **MD 3, MA 9:** 5–20: **99.8%**, 21–40: **99.4%**, 41–60: **55.1%**, 61–80: **22.3%**
> > > - **MD 3, MA 14:** 5–20: **99.5%**, 21–40: **98.0%**, 41–60: **38.5%**, 61–80: **23.0%**, 81–100: **21.9%**, 101–200: **21.3%**
> > > - **MD 5, MA 9:** 5–20: **94.4%**, 21–40: **39.1%**, 41–60: **20.8%**, 61–80: **18.3%**, 81–100: **18.0%**, 101–200: **17.9%**
> > > - **MD 5, MA 14:** 5–20: **97.7%**, 21–40: **49.5%**, 41–60: **19.3%**, 61–80: **19.1%**, 81–100: **19.1%**, 101–200: **19.0%**
> > >
> > > (MD=Max Depth, MA=Max Arguments)
> > >
> > > RIR-GRC has poor generalization: even though RIR-GRC can outperform the other methods in-distribution on ListOps, its accuracy collapses beyond short lengths, whereas BBT-GRC, LDRU, Transformer (ALiBi), and LSTM maintain much stronger performance on OOD sequences.
> > >
> > > [1] SIMPLIFIED STATE SPACE LAYERS FOR SEQUENCE MODELING
> > >
> > > [2] Structured Sparse Transition Matrices to Enable State Tracking in State-Space Models
> > >
> > > [3] Ordered Memory
> > >
> > > [4] Recursion in Recursion: Two-Level Nested Recursion for Length Generalization with Scalability

---

> ### Comment · Reviewer_E7NM · 2025-11-20
>
> Thanks for the additional clarification and discussion. If these are carefully distilled in the paper - and the benefits of monoid are more exhaustively described (e.g. similar to your modular arithmetic result), then I would be more leaning towards acceptance. At this point, I still maintain my score of 4, because I think this paper still may benefit from a second round of review overall where all these details are more carefully presented. However, although I maintain my score as 4, I am comfortable with this paper getting accepted if others favor it given the discussions are incorporated.
>
> > To follow up on your initial feedback, we have completed experiments with RIR-GRC on ListOps and now report the length–generalization behavior explicitly. Across configurations, RIR-GRC achieves very strong accuracy on short sequences but degrades sharply as length increases.
>
> I am personally experienced in experiments with RIR-GRC and can attest that it **does** length generalize quite well as in the paper. So this may be some issue with hyperparameter or the regime difference. In the original case, it was shown to length generalize to 1000s of sequence length after training on <=100 sequence length; moreover there was no consraint on depth on training. So some of the changed variable may be hampering it. Also a priori, it doesn't make sense that it would be worse than BBT-GRC, given the core recursive cell are same in both, and RIR-GRC have, in addition, a better topological bias.
>
> I think I understand where your RIR-GRC experiments might be going wrong.
>
> RIR-GRC is essentially a n-ary recursive setup. If your training regime is <= n or close to n, in training regime it essentially will degerate into Beam-Tree-GRC (the inner recursion) - the outer recursion will rarely activate, and thus it will not learn to adjust information transfer via the outer recursion - and fail to generalize in test when it has to suddenly now activate the outer recursive, beam alignment and everything. If you use the default, I think the default parameter RIR uses for ListOps is ~30 for n, and your training length may be too close or lower than that, - making RIR-GRC incapable of learning. Training on <= 100 sequence length and then testing on however long would be better regime to compare all the models fairly with RIR-GRC. The n can be also reduced - but then RIR-GRC will start to degenerate towards BBT-GRC.

---

> > ### Author Response · Authors · 2025-11-20
> > **Thank you for the discussion**
> >
> > Dear Reviewer E7NM,
> >
> > We would like to expressly thank you for the engagement with the paper and for helping us improve our work.
> >
> > We will revise the introduction and related work to more clearly situate LDRU within the existing family of BBT-RvNNs, and to present our main contribution explicitly as an associativity-biased operator on top of a standard parallel reduction scheme. We will also adapt the presentation in Sec. 3 to show how the method implements our observed connection to the monoid more carefully.
> >
> > We acknowledge that there is a difference between our results and those reported in RIR: we do not wish to claim that RIR-GRC does not length generalize to ListOps. The training/optimizer set up are quite different (we modified it to be in line with our other baselines). Our initial suspicion was that the change from Ranger to Adam could be important, as it is possible that Ranger is better suited to inducing generalization. This hypothesis is consistent with our ablation in Appendix E.1, Fig. 11, where using AMSGrad leads to better length generalization than Adam for Modular Arithmetic.
> >
> > While this was our first intuition, your hypothesis regarding the $n=30$ for RIR is the more likely culprit. To maintain some similarity with the regular tasks, we train <=40. With your suggestion of an $\approx 0.3$ ratio of $n$ to max training length, we propose to re-run the RIR-GRC experiments with $n=12$ instead, to match this ratio on our training data. We feel that this would be a better adaption to our setting that would enable a fairer result to be reported for RIR-GRC. We will not over-interpret our current results.

---

### Official Review · Reviewer_B7Z7 · 2025-10-31

**Soundness:** 3
**Presentation:** 3
**Contribution:** 2
**Rating:** 4
**Confidence:** 3

**Summary:**

This paper introduces the Log-Depth Recurrent Unit (LDRU), a novel neural architecture designed to address the challenge of out-of-distribution (OOD) length generalization. The LDRU is inspired by the algebraic structure of monoids and processes sequences using a learned pairwise operator in a balanced reduction tree, resulting in $O(\log n)$ computational depth. The authors evaluate the LDRU on a comprehensive suite of 21 regular language tasks, including standard benchmarks and a new "prefix language" benchmark they introduce to specifically test long-range dependencies. The empirical results are very strong, showing that the LDRU achieves 100% OOD accuracy on 18/21 tasks and near-perfect (>=96%) on the remaining three, consistently outperforming RNN, LSTM, Transformer, and RegularGPT baselines.

**Strengths:**

1. The LDRU architecture is with a theoretical motivation. Grounding the architecture in the concept of monoid composition and the reduction algorithm provides an alternative to standard recurrent (state-based) or attention-based (fixed-depth) approaches.
2. The LDRU's performance on the 21 regular tasks is better than other architectures (RNN, Transformer, LSTM).

**Weaknesses:**

1. The primary weakness of this paper is its exclusive focus on regular languages. While the authors justify this as a rigorous and verifiable testbed, regular languages are the simplest class in the Chomsky hierarchy. The paper provides a compelling proof of concept, but it leaves the most critical question unanswered: does this approach scale to more complex, non-regular tasks? It is entirely unclear if the LDRU's inductive bias, which aligns so well with monoids (and thus regular languages), will be beneficial or detrimental for context-free or, more importantly, context-sensitive languages like natural language.
2. While the $O(\log n)$ depth and $O(nd^2)$ work complexity are good, the paper doesn't compare LDRU directly to baselines (like RNNs or Transformers) in terms of wall-clock time or throughput.

**Questions:**

Could you conduct more experiments to address weaknesses 1 and 2?

---

> ### Author Response · Authors · 2025-11-17
> **Additional experiments have improved our manuscript**
>
> Dear Reviewer B7Z7,
>
> Thank you for clearly stating your two main weaknesses.
>
> **W1** - Scope beyond regular languages / natural language
>
> We added non-regular, real-world sequence tasks: a suite of natural language classification benchmarks (CoLA, SST-2, QQP, MNLI-m/mm, QNLI, MRPC, AG’s News, DBPedia) and a generalization study on ListOps (see Table 4 / Fig. 4). Across these, LDRU is competitive with or better than strong Transformer variants (including ALiBi PE) under a shared training protocol, indicating that the usefulness of its inductive bias is not restricted to regular languages.
>
> **W2** - Wall-clock time and throughput
>
> We added GPU benchmarks comparing LDRU, RNN, and Transformer under matched parameter budgets (Fig. 3), reporting wall-clock time, throughput, FLOPs, and memory as a function of sequence length.
>
> We hope our response and revised manuscript has adequately addressed both of your identified weaknesses.  If these additions resolve your concerns about scope and practical efficiency, we would appreciate it if you could reconsider your overall score.
>
> The Authors

---

### Official Review · Reviewer_n6mp · 2025-10-31

**Soundness:** 3
**Presentation:** 2
**Contribution:** 3
**Rating:** 6
**Confidence:** 2

**Summary:**

This paper proposes a new architecture, Log-Depth Recurrent Unit (LDRU). LDRU provides a trade-off between width (i.e. work complexity) and depth between current architectures such as RNN and Transformer. Experiments on a wide range of generalization tasks demonstrate the improved performance of LDRU over RNN, LSTM, Transformer and RegularGPT.

**Strengths:**

This paper presents an interesting way to model sequences in a binary-tree manner. It is intuitive that the computation overhead can be reduced in this way. There are many experiments in both the main pages and the appendix, and many examples in the appendix to help

**Weaknesses:**

1. I have only a little background on DFA and I feel the preliminary section is technically heavey and hard to follow. I would suggest to relate the preliminaries with examples in natural language as it is the final testbed for proposed LDRU.

2. The evaluation tasks of this work are not on natural language. I wonder if it is possible to model natural language with LDRU (for example, next-token generation, reasoning, long-context modeling, etc). The tasks in this paper are specific tasks, not general language modeling.

3. The baselines in this paper seem not enough, as RNN, LSTM, and Transformer have been introduced many years ago, and positional encodings are deactivated as described in line 236. I wonder if more recent variants could be compared with the proposed LDRU.

**Questions:**

1. I don't quite get how the proposed neural architecture is related to the finite automatons and prefix languages. I think I missed this part, but I only understand the proposed LDRU as a new kind of attention that is not fully-connected?

2. At first glance I thought this work is about long-context modeling, but turns out it's not. Could you let me know what's the difference between the area of this research from long-context modeling (e.g., context window of Mistral is 8192, but we can extend it with some techniques)? I guess I'm assigned to review this paper because of my background on long-context modeling, and this could be why I cannot follow this paper well, as the tasks in this paper are not standard long-context benchmarks.

3. Are the tasks too simple that LDRU (and some baselines) can achieve 100.0 +- 0 performance? Are we reaching the limit of this research direction as we're achieving 100.0?

4. If we can do binary tree, how about ternary and more generally n-ary trees? Will it be challenge because LDRU is relying on pairwise operations (which only applies for binary tree structure)?

---

> ### Author Response · Authors · 2025-11-17
> **Removed prelims and answering questions**
>
> Dear Reviewer n6mp,
>
> Our revised manuscript addresses your **W2** and **W3** by (i) adding ListOps and eight NLP classification tasks (Sec. 6, Table 4) and (ii) providing GPU runtime, FLOPs, throughput, and memory benchmarks versus RNNs and Transformers (Sec. 3.1, Fig. 3), and the general response summarizes these changes.
>
> **W1** - Overly heavy preliminaries
>
> We acknowledge that the preliminaries currently act as a barrier to readers less familiar with automata theory. As the preliminaries are not critical to understanding the LDRU, and its purpose is to demonstrate the LDRU's connection to automata theory, we propose to move it to the appendix. We have made this change in the revised manuscript. This will make the main body of the paper more accessible to a wider audience while retaining a link to its theoretical motivation.
>
> Regarding your questions:
>
> **Q1** - Link to finite automata
>
> The concept that syntactic/transition monoids can process sequences from regular languages via a reduction is the inspiration for the architecture. Typically, deterministic finite automata (DFA) are the machinery associated with processing regular languages, but it is also possible to reformulate the sequential processing of a DFA into compositions of equivalence classes. Equivalence classes represent sets of sequences possibly with variable lengths that all share the same state mapping in the DFA (we provide an explicit example of this for $D_2$ in Table 24 in the revised appendix). The LDRU is designed to process sequences as compositions of equivalence classes. Our prefix languages are a family of regular tasks that are designed to target long-ranged dependencies.
>
> **Q2** - Differences between length generalization studies and long-context modeling
>
> We thank the reviewer for raising this distinction. In our work, _length generalization_ refers specifically to training on short sequences and evaluating on longer sequences within the same task distribution. This setup is standard in length generalization research and is the primary way regular-language models are evaluated [1–4]. In contrast, _long-context modeling_ typically concerns handling very long inputs at training and inference time, and is usually evaluated in terms of memory, retrieval, or attention-scaling behavior rather than compositional consistency. Our focus is on **generalization of the underlying computation**, not on scaling to long input contexts.
>
> **Q3** - Reached the research limit
>
> The perfect scores reflect successful generalization, not task simplicity. These benchmarks are designed to test whether a model has learned the underlying rule rather than memorized training lengths. Established architectures such as RNNs, LSTMs, and Transformers routinely fail to achieve perfect OOD accuracy on the same tasks, which is why they remain standard stress tests in the literature. The LDRU’s results indicate that it has learned the correct compositional rules to correctly process any sequence. Harder tasks such as $D_6$​, $D_8$​, and $D_{12}$​ make this clear: perfect generalization only appears once the LDRU  has been exposed to a sufficient number of monoid compositions during training.
>
> **Q4** - $n$-ary operators
>
> The $n$-ary extension is straightforward in principle: once the operator is defined over tuples, the same reduction can be carried out with higher branching factors. RegularGPT [3] already explores this direction and shows that $n$-ary trees can improve performance on natural-language tasks. Our work isolates the binary operator and shows that learning a stable pairwise composition is sufficient to obtain generalization across all regular tasks tested. Extending to $n$-ary operators introduces additional design choices, so we treat it as follow-up rather than a limitation of the present formulation. Related work that studies $n$-ary operators also include [5].
>
> We’re happy to provide additional clarification as needed. Thank you for reading the manuscript and for your thoughtful review. If our response has resolved your concerns, we would be grateful if you could consider adjusting your score accordingly.
>
> The Authors
>
> [1] Neural Networks and the Chomsky Hierarchy
>
> [2] Randomized Positional Encodings Boost Length Generalization of Transformers
>
> [3] Transformer Working Memory Enables Regular Language Reasoning And Natural Language Length Extrapolation
>
> [4] Training Neural Networks as Recognizers of Formal Languages
>
> [5] Recursion in Recursion: Two-Level Nested Recursion for Length Generalization with Scalability

---

> > ### Comment · Reviewer_n6mp · 2025-11-18
> > **Thanks for your clarifications. Please be tuned!**
> >
> > I appreciate the authors' timely response. At a glance, the point-by-point clarifications seem helpful and will likely improve my understanding of the techniques and contributions.
> >
> > I am currently occupied and cannot review the responses in depth immediately, but I will carefully read both the rebuttal and the revised paper and follow up as soon as possible. I will do my best to complete this before Fri, Nov 21.

---

> > ### Comment · Reviewer_n6mp · 2025-11-24
> >
> > Thank you for the detailed response, and apologies for the delayed follow-up. I’ve been quite overwhelmed recently.
> >
> > W1 Moving the technical preliminaries to the appendix indeed improves readability. As someone without a deep automata background, this change made it much easier for me to engage with the core method.
> >
> > Q1 I still find it difficult to follow. In particular, the sentence “Equivalence classes represent sets of sequences possibly with variable lengths that all share the same state mapping in the DFA (we provide an explicit example of this for $D_2$ in Table 24 in the revised appendix).” remains unclear to me even after several readings. Could you briefly remind me what the task $D_2$ is, and provide some intuitive explanation or real-world analogy? For example, is there a conceptual parallel to something familiar in modern ML, such as “training an LLM with context window 8k but generalizing to 128k”? My background is primarily in LLM context-window extension.
> >
> > W2 & W3. Thank you for adding NLP length-generalization benchmarks and comparisons with commonly used baselines. These results help me better appreciate the relevance of your method to long-context NLP, even if that was not the original automata-theoretic motivation.
> >
> > Q2 I now understand the distinction you draw: your length generalization concerns longer sequences **within the same task distribution**, whereas LLMs typically fail when exceeding the pretraining window size, which is a different phenomenon (there isn't task distribution in LLM window size extension). My earlier confusion likely stemmed from not being familiar with the specific regular-language benchmarks in Table 2. Please correct me if my understanding is incorret.
> >
> > Q3 Could you let me know the difference between $D_n$ tasks, and why $D_6$​, $D_8$​, and $D_{12}$​ are harder tasks? Additionally, is there room to design even more complex or difficult evaluation tasks that push the boundaries of these models further?
> >
> > Q4 Your clarification on this was helpful. I'm clear. Thanks!

---

> ### Comment · Reviewer_n6mp · 2025-11-24
> **Additional Reflections After Reading Other Reviews**
>
> **After scanning comments from the other reviewers, I would like to add a few cross-review observations** (I refined this comment with LLM so that it is easier for the authors and AC to read).
>
> > Scope beyond formal languages.
>
> Both B7Z7 and I raised concerns about how the proposed method extends from the formal-language tasks in this paper to more general natural-language settings (also echoed by mVcQ). Personally, I believe that addressing formal languages is already a meaningful contribution, as this is historically how earlier generations of NLP researchers approached language modeling before large-scale pretraining. In addition, insights from this line of work may point toward promising directions for improving length generalization in code generation, where the underlying structure is far more formal than natural language.
>
> > Novelty relative to prior work (e.g., RegularGPT)
>
> Reviewer E7NM appears to have substantial expertise in this sub-area, certainly more than me (I wasn’t familiar with RegularGPT before reviewing). I cannot fully assess whether this work is incremental over RegularGPT, but I believe the paper would benefit from clarifying the conceptual and empirical novelty relative to that line of work, and ideally addressing E7NM’s concerns directly and convincingly.
>
> > Synthetic vs. natural tasks.
>
> E7NM and myself noted concerns about the synthetic nature of the original tasks. I appreciate that the authors added natural-language benchmarks, which helps demonstrate that the method is not purely tied to automata-style synthetic settings.
>
>
> > Parallelism and computational trade-offs.
>
> hrDN’s comments on parallel computing (W3) were insightful. While RNNs were not parallelizable and still had substantial impact, thinking about parallelism for practical deployment of new architectures is important. This seems like a valuable direction for future work, and I encourage the authors to explicitly discuss these trade-offs and potential paths toward increased parallelizability. mVcQ also raised concerns about runtime and memory overhead introduced by the reduction tree; these would be helpful to address more explicitly in the main text.
>
> > Related work coverage & baselines.
>
> E7NM and hrDN requested several additional related works. Incorporating them would strengthen the paper. Regarding the Gated DeltaNet baseline, if a lightweight comparison is feasible (assuming the implementation is available), adding it would help position the proposed method more clearly within the landscape of sequence modeling architectures. The authors mentioned a subsequent revision including this comparison will be ready during the discussion period.

---

> ### Author Response · Authors · 2025-11-24
>
> Thank you for the additional comments and no worries regarding the timing.
>
> **Q1** - We can't think of a good conceptual parallel in modern ML, but we will try to explain more clearly using the $D_2$ example. We hope this description helps, please follow up if it is still unclear.
>
> $D_2$ is a regular task where sequences that represent balanced bracket sequences with a maximum stack depth of 2 are accepted and any other sequence is rejected. These types of language are related to syntax-checking inputs. The language is binary, we use 0/1 in the paper but you can map to the brackets with '0' -> '(' and '1' -> ')'. These tokens represent stack actions: '0' as a push and '1' as a pop.
>
> The maximum depth of the task is finite, so you can specify a DFA that can process any length sequence to correctly decide whether the sequence should be accepted or rejected. A sequence is processed through $D_2$'s DFA by reading each token one at a time and updating the state. There are 4 states in this DFA: the initial state, 0; the depth 1 state, 1; the depth 2 state 2; and the 'invalid' state, 3. The accepting state is 0. An explicit example of $D_6$ is presented in Fig. 13. When a sequence is processed by the DFA, it simulates stack behavior one push/pop at a time. However, as we already have the entire input sequence we can summarize the final state of the stack by composing the actions together.
>
> For a given sequence, you can determine the state mapping by iterating through all the states and determining the state the DFA is in once it finishes processing the sequence. Consider '00', a double stack push. It has the state mapping 0 -> 2 (depth 0 to depth 2), 1 -> 3 (depth 1 to invalid state), 2 -> 3 (depth 2 to invalid state), and 3 -> 3 (invalid maps back to itself). We can compose state mappings together by chaining them, e.g., the composition of x -> y and y -> z is x -> z. Now, consider the sequence '0011'. This is composed of '00', a double stack push, and '11', a double stack pop. If you compose these together (by composing the state mappings given for each of the elements in Table 24, state 3 always maps back to itself), you get the mapping for $e_{12}$. '0011' belongs to the equivalence class $e_{12}$, and in general, any sequence that belongs to $e_3$ composed with any sequence that belongs to $e_6$ will result in a sequence that belongs to $e_{12}$.
>
> In $D_2$'s DFA, there are exactly 15 unique state mappings possible. Each of these state mappings is an equivalence class and we can characterize any sequence by the state mapping it induces in the DFA. The state mappings also reveal which sets of sequences will be accepted: $e_0$, $e_4$, and $e_{12}$, as the initial state maps to the accepting state (i.e., 0 maps back to 0).
>
> **Q2** - Your understanding is correct. We train a model to do a specific task and then evaluate how well it generalizes to that task outside of its training lengths.
>
> **Q3** - The $D_n$​ tasks are structurally all the same: the goal is to classify sequences into those that are balanced with maximum nesting depth at most $n$ and those that are malformed or exceed depth $n$. Increasing $n$ makes the task harder, as models must handle more complex state tracking and effectively “remember more”. From the LDRU’s perspective, the size of the underlying monoid also grows cubically in $n$, so much more structure must be compressed into the same embedding dimension. As a result, tasks with larger $n$, such as $D_6$, $D_8$, and $D_{12}$, are empirically more challenging. We view more parameterized regular tasks (such as our prefix languages) as a natural next step for this line of work, since they can be designed to probe specific failure modes in models and can be tuned to become increasingly complex.

---

> ### Author Response · Authors · 2025-11-25
>
> Thank you for the summary.
>
> 1. Scope beyond formal languages
>
> The revised manuscript now includes natural-language benchmarks and a further generalization study on ListOps, a non-regular task. These additions address the reviewers’ concerns about limiting evaluation to a regular task setting.
>
> 2. Novelty relative to RegularGPT
>
> RegularGPT is similar to the LDRU but the parameterization of the operator is different. They use a standard Transformer layer with attention, we have shown empirically that this does not achieve as strong generalization as the LDRU's operator on regular tasks.
>
> 3. Synthetic vs. natural tasks
>
> The added natural language experiments and the ListOps study expand the evaluation beyond fully synthetic regular tasks, addressing this concern without altering the paper’s core focus.
>
> 4. Parallelism and computational trade-offs
>
> The revision includes additional experiments and a short analysis quantifying runtime, FLOPs, and memory characteristics, together with a comparison across architectures. These additions respond directly to points raised by hrDN and mVcQ and now appear in Section 3.1.
>
> 5. Related work and baselines
>
> All requested related work has been incorporated into the Related Work section. In the discussion, we have also reported Gated Delta Net performance and provided in-distribution accuracies for all baselines on the regular-language experiments to clarify baseline strength and fairness.

---

### Author Response · Authors · 2025-11-12
**Source Code for Reproducibility**

Dear Reviewers,

As stated in the reproducibility statement, we are providing an anonymous GitHub repository containing the code required to reproduce all experimental results:

https://github.com/8YWJuZzO/supplementary_code.

The repository includes:
- step-by-step installation instructions,
- bash scripts for exact experiment reproduction,
- toy experiments runnable on a laptop.

We are preparing a response to the reviews that will include a consolidated discussion of concerns shared across reports and separate point-wise responses to each individual review.

The Authors

---

### Author Response · Authors · 2025-11-17
**Addressing scope, practical computation scaling/comparison, and additional Transformer PE baselines**

Dear Reviewers,

We would like to thank all of you for providing detailed feedback. We have uploaded a revised manuscript that we hope addresses the majority of concerns. Below, we summarize and respond to the three common themes across reviews: **scope**, **computational scaling**, and **Transformer baseline strength**.

In the revision, we (i) broaden evaluation beyond regular languages, (ii) add practical computational measurements on GPUs, and (iii) strengthen Transformer baselines with two recent positional encodings. All additions are highlighted in the manuscript: Fig. 3 (compute), Fig. 4 (ListOps), Table 2 (Transformer PE baselines), and Table 4 (NLP results), with further experimental details given in the appendix.

### 1) Scope Beyond Regular Languages  [n6mp, B7Z7, E7NM, mVcQ]

1. New NLP evaluations (8 sequence classification tasks, presented in Table 4).
    To evaluate generality beyond regular languages while maintaining a consistent modeling setup, we added eight standard NLP classification tasks: GLUE (CoLA, SST-2, QQP, MNLI, QNLI, MRPC), AG’s News, and DBPedia. These experiments are intended to demonstrate the viability of the LDRU outside regular-language settings. We compare against Transformers using three standard positional encodings (NoPE, Sinusoidal, ALiBi). All models are trained from scratch under identical budgets (optimizer, training steps), and we perform a small hyperparameter sweep for each architecture on AG’s News and reuse those settings for the remaining tasks.

    The LDRU performs comparably to the Transformer baselines with NoPE, Sinusoidal, and ALiBi (Table 4), and notably outperforms all baselines on QQP, MNLI (matched/mismatched), and DBPedia. We view this as initial evidence that the LDRU’s inductive structure can be effective on non-regular, real-text data, even though the model was originally motivated by regular-language structure. To maintain focus, we do not attempt generative modeling in this work; this is a natural direction for future research.

1. ListOps systematic generalization study (presented in Fig. 4)

    We further evaluate the LDRU on ListOps across three dataset sizes (100k, 500k, 1M), comparing against BBT-GRC (a Tree-RvNN), Transformers with three positional encodings (NoPE, Sinusoidal, ALiBi), and LSTM (Fig. 4; additional datasets in the appendix). The performance ordering is stable across sizes:

	BBT-GRC > LDRU > Transformer (ALiBi) $\approx$ LSTM > Transformer (NoPE/Sinusoidal).

	The LDRU consistently outperforms the LSTM and all Transformer variants on out-of-distribution (length-extrapolation) tests. Interestingly, BBT-GRC’s advantage increases with more data. We hypothesize that the GRC cell’s inductive structure is better aligned with ListOps—particularly for learning the SM operator—whereas the operator we propose is designed to approximate recurrent processing and is therefore better aligned with associative structure (as in regular languages). Even in this misaligned setting, the LDRU remains competitive with mainstream architectures and demonstrates partial length generalization.

Finally, we note that in both the NLP and ListOps experiments, we introduced an associativity-regularization term for the LDRU and found via hyperparameter sweeps that this improved performance, suggesting that associative biases are a promising future research direction.

---
### 2) Practical Computational Scaling [B7Z7, hrDN, mVcQ] (Fig. 3)

Wall-clock time, FLOPs, throughput, and memory usage on GPU (RTX 6000 Ada)
To address concerns regarding practical cost, Fig. 3 reports four measurements for the LDRU, a simple RNN, and a Transformer (all ~160k parameters) with batch size 32 and sequence lengths up to 2048:

- End‑to‑end step time (forward+backward): The empirical curves show that LDRU’s runtime scales with sequence length as predicted by its parallel reduction structure and becomes faster than the RNN at longer lengths.
- Throughput vs. length: The LDRU exhibits stepwise throughput drops at powers of two due to requiring $\lceil \log_2 (n)\rceil$ reduction steps. When the sequence crosses a power-of-two boundary, the reduction depth increases, and throughput temporarily drops, increasing again as the sequence length approaches the next power of two. The other plots also reflect this trend.
- Estimated FLOPs: Profiling indicates that the LDRU uses more FLOPs than a simple RNN, but scales with the same order.
- Peak memory usage: The LDRU consumes more memory than the RNN, as it must store intermediate results from the parallel reduction.

These results clarify the runtime/memory trade-offs and position the LDRU as an architecture that provides a recurrent form of computation with increased parallelizability: offering an intermediate choice between RNNs and Transformers.

---
[Continued in following comment]

---

### Author Response · Authors · 2025-11-17
**Cont. shared concerns**

### 3) Stronger Transformer Baselines [n6mp, hrDN]

Responding to the request for more modern Transformer setups, we added ALiBi and randomized RoPE as PEs that are designed for length generalization (Table 2):

- ALiBi provides modest gains on several Tomita and prefix tasks but does not yield uniform generalization.
- Randomized RoPE produces the strongest baseline performance on $D_{12}$, generally improves $D_n$​ and Even Pairs, but regresses on Modular Arithmetic, Tomita-3, and Tomita-4. This suggests that while randomized RoPE improves length extrapolation, it does so for specific subclasses of regular languages.

Across tasks, these positional encodings reorder Transformer variants but do not eliminate the underlying failure mode: Transformer performance remains task-dependent, with large drops on Even Pairs, Modular Arithmetic, and several prefix languages. The LDRU’s gains reflect its architectural structure rather than reliance on a particular positional encoding.

---

We appreciate the reviewers’ insights, which have led to substantial improvements in the manuscript. While this general response focuses on the major shared concerns, we recognize that reviewers also raised specific points that are not addressed here. We respond to all of those individually in the per-reviewer comments. We believe the expanded evaluations and analyses meaningfully strengthen the work, and we thank the reviewers again for their time and consideration.

The Authors

---

### Author Response · Authors · 2025-11-24
**Finished all additional experiments (1/2)**

As promised, we ran the additional Gated Delta Net baseline on the regular-language tasks. We report OOD accuracies in Table 1 and in-distribution accuracies in Table 2. The main findings are:

- Among the SSM baselines we evaluated, the typical performance order on OOD generalization is SD-SSM > Gated Delta Net > S5, but all three remain far below LDRU on the harder regular tasks.
- The increased expressivity of Gated Delta Net does not close the gap to LDRU in terms of length generalization: LDRU remains the only model that consistently achieves 100.0% OOD accuracy across almost all 21 regular tasks.

These additional results are consistent with our claim that LDRU provides the strongest length generalization among all models we evaluated on these regular tasks.

We have now provided detailed responses to all reviews and additional experiments where requested. If any reviewer has remaining concerns or would like clarification on specific points, we are happy to address them so that our rebuttal and these results can be fully taken into account during the private discussion phase.

**Table 1.** OOD accuracies (%) for SSM models and LDRU, averaged over 3 seeds.


| Task               | S5                  | SD-SSM              | Gated Delta Net     | LDRU                |
|--------------------|---------------------|---------------------|---------------------|---------------------|
| Even Pairs         | 53.9\% ± 0.8\%  | 65.7\% ± 0.8\%  | 52.2\% ± 0.8\%  | 100.0\% ± 0.0\% |
| Modular Arithmetic | 47.6\% ± 6.7\%  | 99.9\% ± 0.0\%  | 76.5\% ± 3.1\%  | 100.0\% ± 0.0\% |
| Parity Check       | 50.1\% ± 0.1\%  | 71.4\% ± 1.4\%  | 53.4\% ± 0.3\%  | 100.0\% ± 0.0\% |
| Cycle Navigation   | 22.9\% ± 0.7\%  | 44.2\% ± 0.8\%  | 25.8\% ± 3.8\%  | 100.0\% ± 0.0\% |
| $D_2$              | 88.4\% ± 10.3\% | 100.0\% ± 0.0\% | 100.0\% ± 0.0\% | 100.0\% ± 0.0\% |
| $D_3$              | 83.6\% ± 9.3\%  | 96.3\% ± 3.9\%  | 91.6\% ± 14.3\% | 100.0\% ± 0.0\% |
| $D_4$              | 81.7\% ± 10.3\% | 97.3\% ± 2.3\%  | 93.1\% ± 6.0\%  | 100.0\% ± 0.0\% |
| $D_6$              | 75.0\% ± 1.5\%  | 92.2\% ± 4.3\%  | 75.6\% ± 0.6\%  | 98.5\% ± 2.4\%  |
| $D_8$              | 73.7\% ± 0.5\%  | 87.2\% ± 2.9\%  | 76.0\% ± 1.3\%  | 98.1\% ± 1.6\%  |
| $D_{12}$           | 73.5\% ± 0.3\%  | 72.9\% ± 4.3\%  | 74.6\% ± 0.5\%  | 96.0\% ± 2.1\%  |
| Tomita 3           | 72.8\% ± 1.6\%  | 100.0\% ± 0.0\% | 99.9\% ± 0.1\%  | 100.0\% ± 0.0\% |
| Tomita 4           | 72.3\% ± 12.8\% | 100.0\% ± 0.0\% | 99.1\% ± 0.2\%  | 100.0\% ± 0.0\% |
| Tomita 5           | 65.8\% ± 3.2\%  | 81.1\% ± 1.8\%  | 75.1\% ± 0.4\%  | 100.0\% ± 0.0\% |
| Tomita 6           | 49.9\% ± 0.1\%  | 85.6\% ± 3.7\%  | 54.7\% ± 0.5\%  | 100.0\% ± 0.0\% |
| Tomita 7           | 98.7\% ± 2.2\%  | 100.0\% ± 0.0\% | 100.0\% ± 0.0\% | 100.0\% ± 0.0\% |
| $P_{1,2}$          | 52.9\% ± 2.8\%  | 61.8\% ± 3.5\%  | 52.4\% ± 1.7\%  | 100.0\% ± 0.0\% |
| $P_{2,2}$          | 29.3\% ± 1.8\%  | 34.0\% ± 1.6\%  | 22.0\% ± 11.9\% | 100.0\% ± 0.0\% |
| $P_{4,2}$          | 12.1\% ± 2.1\%  | 99.8\% ± 0.2\%  | 18.9\% ± 1.6\%  | 100.0\% ± 0.0\% |
| $P_{1,4}$          | 31.8\% ± 3.6\%  | 38.7\% ± 0.5\%  | 30.1\% ± 1.6\%  | 100.0\% ± 0.0\% |
| $P_{2,4}$          | 14.4\% ± 1.3\%  | 30.2\% ± 4.9\%  | 21.5\% ± 9.5\%  | 100.0\% ± 0.0\% |
| $P_{4,4}$          | 8.0\% ± 3.4\%   | 42.9\% ± 2.5\%  | 69.4\% ± 6.0\%  | 100.0\% ± 0.0\% |


[Table 2 in continued comment]

---

### Author Response · Authors · 2025-11-24
**Finished all additional experiments (2/2)**

**Table 2.** In-distribution accuracies for each model/task, averaged over 3 seeds.

| Task               | RNN    | LSTM   | TF (NoPE) | TF (ALiBi) | TF ($\sim$RoPE) | S5     | SD-SSM | Gated Delta Net | RegularGPT | LDRU   |
| ------------------ | ------ | ------ | --------- | ---------- | --------------- | ------ | ------ | --------------- | ---------- | ------ |
| Even Pairs         | 100.0% | 100.0% | 90.1%     | 100.0%     | 99.4%           | 100.0% | 100.0% | 100.0%          | 100.0%     | 100.0% |
| Modular Arithmetic | 100.0% | 100.0% | 94.9%     | 86.3%      | 75.7%           | 67.6%  | 100.0% | 85.4%           | 100.0%     | 100.0% |
| Parity Check       | 100.0% | 100.0% | 90.1%     | 51.9%      | 56.6%           | 99.1%  | 100.0% | 100.0%          | 100.0%     | 100.0% |
| Cycle Navigation   | 100.0% | 100.0% | 84.8%     | 84.7%      | 89.7%           | 100.0% | 100.0% | 99.3%           | 100.0%     | 100.0% |
| $D_2$              | 100.0% | 100.0% | 100.0%    | 98.7%      | 98.7%           | 100.0% | 100.0% | 100.0%          | 100.0%     | 100.0% |
| $D_3$              | 100.0% | 100.0% | 100.0%    | 99.2%      | 99.1%           | 100.0% | 100.0% | 90.8%           | 100.0%     | 100.0% |
| $D_4$              | 100.0% | 100.0% | 99.9%     | 99.8%      | 99.6%           | 100.0% | 100.0% | 97.7%           | 100.0%     | 100.0% |
| $D_6$              | 100.0% | 100.0% | 99.3%     | 99.8%      | 99.8%           | 100.0% | 100.0% | 98.9%           | 100.0%     | 100.0% |
| $D_8$              | 100.0% | 100.0% | 98.9%     | 99.8%      | 99.8%           | 100.0% | 100.0% | 99.2%           | 100.0%     | 100.0% |
| $D_{12}$           | 100.0% | 100.0% | 98.7%     | 99.6%      | 99.9%           | 100.0% | 100.0% | 96.7%           | 100.0%     | 100.0% |
| Tomita 3           | 100.0% | 100.0% | 99.8%     | 99.1%      | 96.8%           | 100.0% | 100.0% | 99.2%           | 100.0%     | 100.0% |
| Tomita 4           | 100.0% | 100.0% | 99.7%     | 98.9%      | 98.7%           | 100.0% | 100.0% | 100.0%          | 100.0%     | 100.0% |
| Tomita 5           | 100.0% | 100.0% | 93.4%     | 74.2%      | 74.2%           | 97.9%  | 100.0% | 100.0%          | 100.0%     | 100.0% |
| Tomita 6           | 100.0% | 100.0% | 89.7%     | 51.3%      | 51.3%           | 93.5%  | 100.0% | 99.9%           | 100.0%     | 100.0% |
| Tomita 7           | 100.0% | 100.0% | 100.0%    | 99.0%      | 99.9%           | 100.0% | 100.0% | 100.0%          | 100.0%     | 100.0% |
| $P_{1,2}$          | 100.0% | 100.0% | 90.1%     | 99.9%      | 98.8%           | 100.0% | 100.0% | 98.8%           | 100.0%     | 100.0% |
| $P_{2,2}$          | 100.0% | 100.0% | 85.0%     | 97.4%      | 96.2%           | 100.0% | 100.0% | 98.3%           | 100.0%     | 100.0% |
| $P_{4,2}$          | 100.0% | 100.0% | 81.3%     | 99.3%      | 97.5%           | 100.0% | 100.0% | 99.6%           | 100.0%     | 100.0% |
| $P_{1,4}$          | 100.0% | 100.0% | 85.0%     | 99.8%      | 99.9%           | 100.0% | 100.0% | 99.4%           | 100.0%     | 100.0% |
| $P_{2,4}$          | 100.0% | 100.0% | 81.2%     | 98.3%      | 97.4%           | 100.0% | 100.0% | 98.1%           | 100.0%     | 100.0% |
| $P_{4,4}$          | 100.0% | 100.0% | 78.4%     | 99.0%      | 98.8%           | 100.0% | 100.0% | 100.0%          | 100.0%     | 100.0% |

---

### Meta-Review · Area_Chair_2Z7j · 2026-01-04

**Summary:**

This paper introduces the Log-Depth Recurrent Unit (LDRU), a sequence modeling architecture designed to address the challenge of out-of-distribution (OOD) length generalization. Inspired by the algebraic structure of monoids, the LDRU utilizes a learned pairwise operator in a balanced reduction tree to achieve uniform logarithmic computational depth across tokens. Multiple reviewers initially criticized the narrow focus on synthetic regular languages, which are the simplest class in the Chomsky hierarchy. While the LDRU performed exceptionally well, reviewers questioned its scalability to context-sensitive languages like natural language. Reviewers pointed out that the original Transformer baselines were weak due to a lack of advanced positional encodings. They requested comparisons with ALiBi and randomized RoPE, as well as modern State-Space Models like Gated Delta Net. The authors provided a very active and extensive rebuttal, making substantial updates to the manuscript, including modern baselines for ALiBi, randomized RoPE, and Gated Delta Net. Reviewers (notably E7NM) expressed concern that the scale of changes required during the rebuttal (e.g., adding entirely new task domains and baselines) resulted in a manuscript that differed too significantly from the one originally submitted for review.

**Reviewer Concerns:**

The authors added GPU benchmarks reporting wall-clock time, throughput, FLOPs, and memory usage. This directly addressed concerns from Reviewers B7Z7, hrDN, and mVcQ regarding the real-world efficiency of the reduction tree. Initially criticized for using a weak "NoPE" baseline, the authors incorporated ALiBi and randomized RoPE. They demonstrated that while these encodings improve Transformers, they do not resolve the underlying failure modes on regular languages that LDRU handles successfully. The authors added 8 NLP classification tasks (GLUE, AG's News, DBPedia). These results showed that LDRU performs comparably to or better than Transformers on real-text data, satisfying Reviewers n6mp, B7Z7, and mVcQ that the model is not strictly limited to formal languages. Reviewer E7NM remained unconvinced that LDRU was a significant departure from BT-RvNNs or RegularGPT. While the authors argued their "associativity-biased operator" was a unique contribution , the reviewer viewed these as "minor technical additions" rather than a fundamental breakthrough. Although the authors added a Gated Delta Net comparison , Reviewer hrDN remained skeptical, noting that modern SSMs are achieving near-perfect results on similar tasks and that LDRU might essentially be a non-linear variant of existing state-space/linear attention mechanisms.

**Reviewer Scores:**

Reviewer E7NM was the most critical regarding the paper's "lineage". While they commended the authors for their extensive experimentation and rebuttal , they explicitly stated they were "still not completely convinced". Their primary concern shifted from technical content to a procedural "broader issue": that the paper had changed too much during the rebuttal to be fairly reviewed without a second round of formal evaluation. Reviewer hrDN remained skeptical of the architecture's novelty. They viewed the LDRU as potentially equivalent to existing linear attention or state-space models when linearized. While they might have appreciated the added FLOPs and parameter counts , their fundamental concerns about the architecture's uniqueness and parallel training efficiency were harder to satisfy.

---

### Decision · Program_Chairs · 2026-01-26

Reject